# REWARDING PROGRESS: SCALING AUTOMATED PROCESS VERIFIERS FOR LLM REASONING

**Amrith Setlur**[*,1]**, Chirag Nagpal**[*,2]**, Adam Fisch**[3]**, Xinyang Geng**[3]**, Jacob Eisenstein**[3]**,
**Rishabh Agarwal**[3]**, Alekh Agarwal**[2]**, Jonathan Berant**[3,†]**, Aviral Kumar**[1,3,†]
[1]CMU, [2]Google Research, [3]Google DeepMind

## ABSTRACT

A promising approach for improving reasoning in large language models is to use process reward models (PRMs). PRMs provide feedback at each step of a multi-step reasoning trace, improving credit assignment over outcome reward models (ORMs) that only provide feedback at the final step. However, collecting dense, per-step human labels is not scalable, and training PRMs from automatically-labeled data has thus far led to limited gains. With the goal of using PRMs to improve a *base* policy via test-time search and reinforcement learning (RL), we ask: "How should we design process rewards?" Our key insight is that, to be effective, the process reward for a step should measure *progress*: a change in the likelihood of producing a correct response in the future, before and after taking the step, as measured under a *prover* policy distinct from the base policy. Such progress values can distinguish good and bad steps generated by the base policy, even though the base policy itself cannot. Theoretically, we show that even weaker provers can improve the base policy, as long as they distinguish steps without being too misaligned with the base policy. Our results show that process rewards defined as progress under such provers improve the efficiency of exploration during test-time search and online RL. We empirically validate our claims by training *process advantage verifiers (PAVs)* to measure progress under such provers and show that compared to ORM, they are $> 8\%$ more accurate, and $1.5 - 5\times$ more compute-efficient. Equipped with these insights, our PAVs enable **one of the first results** showing a $6\times$ gain in sample efficiency for a policy trained using online RL with PRMs vs. ORMs.

## 1 INTRODUCTION

Trained reward models or *verifiers* are often used to improve math reasoning in large language models, either by re-ranking solutions at test-time (Collins, 2000) or during reinforcement learning (RL) (Uesato et al., 2022). Typically, verifiers are trained to predict the outcome of an entire reasoning trace, often referred to as *outcome* reward models (ORM) (Cobbe et al., 2021b; Hosseini et al., 2024). However, ORMs only provide a sparse signal of correctness, which is hard to learn from and inefficient to search against. This challenge is alleviated by fine-grained supervision, in theory. For reasoning, prior works train *process* reward models (PRMs) that assign intermediate rewards after each step of search (Snell et al., 2024) or during RL. While Lightman et al. (2023) obtains PRM annotations from human raters, this approach is not scalable. More recent works (Wang et al., 2024; Luo et al., 2024) train PRMs to predict automatically-generated annotations that estimate future success of solving the problem, akin to value functions in RL. So far, automated PRMs, especially as dense rewards in RL, only improve by 1-2% over ORMs (Shao et al., 2024), raising doubts about their utility.

To resolve these uncertainties, in this paper, our goal is to train automated PRMs that can improve a *base* policy, via compute efficient test-time search, and sample-efficient online RL with PRMs as dense rewards. For this, we first ask: **(i)** what should the *per-step* process rewards measure, and **(ii)** what kind of automated data collection strategy should we use to train PRMs that predict this measure. For **(i)**, conventional belief (Lightman et al., 2023; Uesato et al., 2022) has been to measure mathematical correctness or relevance of steps. But, it is unclear if this supervision yields the most improvement in the base policy (e.g., a policy may need to generate simpler, repetitive, and even incorrect steps to explore and discover the final answer during test-time search and RL). Our key insight is that per-step, process rewards that measure notion of *progress*: change in the likelihood of arriving at a correct final answer before and after taking the step, are effective, for both test-time

---

[*]Equal contribution, [†]Equal advising. Please send correspondence to: *asetlur@andrew.cmu.edu*.

**Figure 1:** *Process advantage verifier (PAV):* Process reward for a step is defined as progress (advantage) under the prover policy, *i.e.*, change in prover policy's success rate before and after the step. **(a):** The base policy samples both correct ❶ and incorrect ❷ steps but struggles to succeed from either. A strong prover completes the solution from any step, making no progress on either ❶ or ❷ (both scored 0.0). Conversely, a complementary prover policy distinguishes ❶, ❷ more prominently (only succeeds from ❶). **(b,c):** Compared to ORMs, PAVs are 5x more compute efficient, 10% more accurate in test-time search, and 6x more sample efficient for RL.

beam search and online RL. Reinforcing promising steps that make progress regardless of whether they appear in a correct or incorrect trace results in overall ***better exploration*** of possible answers at initial steps, which is crucial when the approach to solve a problem is not clear. Formally, such rewards correspond to per-step *advantages* of steps from the RL literature (Sutton & Barto, 2018). We empirically show that using advantages in conjunction with ORM rewards outperforms using future probabilities of success or $Q$-values (Wang et al., 2024) for both search and RL. This is because, in the combinatorial space of responses, $Q$-values mainly "exploit" states whereas advantages also "explore" steps that make the most progress towards the final answer.

To answer **(ii)** above, we first note that advantages under a poor base policy are $\approx 0$ on most steps, and thus will not be informative for test-time search. More generally, we argue that regardless of the strength of the base policy, using its own per-step advantages as process rewards in RL will result in updates equivalent to *only* using outcome rewards. Hence, we propose to use advantages estimated via rollouts under a different *"prover" policy* as process rewards (Fig. 1). How should we choose this prover policy? A natural guess would be to use a very capable prover policy, akin to how strong reward models are found to be effective for RLHF (Gao et al., 2023). However, we show advantages under an overly capable prover, that can succeed from any step, fail to distinguish between good and bad steps. A similar argument holds for very weak provers. We formalize this intuition to define good provers as policies that are *complementary* to the base policy, *i.e.*, policies that can distinguish steps, while producing advantages correlated with the base policy. For example, when restricted to the special case of Best-of-K policies corresponding to a base policy, the set of good provers consists of policies for $K > 1$ with not too large values of $K$. Contrary to intuition, in general, this set also contains policies that are worse than the base policy, but can still improve it. We show that dense verifiers trained to predict advantages of these prover policies, which we call ***process advantage verifiers (PAVs)***, accelerate sample and compute efficiency of RL and search.

With the conceptual design of PAVs in place, we prescribe practical workflows for training PAVs and demonstrate their efficacy on a series of 2B, 9B, and 27B Gemma2 models (Gemma Team et al., 2024). PAV training data is gathered by sampling "seed" solution traces from the prover and partial rollouts from the same to estimate the $Q$-value at each prefix of the seed trace. Our workflow prescribes favorable ratios for seed and partial rollouts. Our first set of empirical results show that for an equal budget of test-time compute, combining beam search with trained PAVs is $> 8\%$ better in accuracy, and **1.5 − 5×** more compute efficient compared to best-of-N re-ranking against an ORM. Dense rewards from PAVs improve the efficiency of step-level exploration during search by pruning the combinatorial space of solutions aggressively and honing in on a diverse set of possible sequences. Finally, we demonstrate ***for the first time***, that using PAVs as dense rewards in RL scales up data efficiency by **6×** compared to only using outcome rewards. Moreover, base policies trained with PAVs also achieve **8×** better Pass@N performance (chance of sampling the correct solution in $N$ attempts), and consequently afford a higher ceiling on the performance of any test-time re-ranker.

## 2    PRELIMINARIES, DEFINITIONS, AND NOTATION

Following protocols from Uesato et al. (2022); Lightman et al. (2023), a reasoning trace from an LLM consists of multiple logical steps separated by a demarcation token. An outcome reward model (ORM) is a trained verifier that assigns a scalar numerical score after the last step of the trace, and a process reward model (PRM) is a trained verifier that scores each step of the trace individually.

**Problem setup and notation.** Given a math problem $x \in X$, our goal is to improve a *base policy* $\pi$ that samples a response $y \sim \pi(\cdot \mid x)$ in the set $\mathcal{Y}$. A response $y$ consists of multiple reasoning steps (maximum $H$), separated by a delimiter ('next line' in our case), *i.e.*, $y = (a_1, a_2, \ldots, a_H)$. Since sampling is auto-regressive, we can view each step as an action taken by the agent $\pi$ in a Markov decision process with deterministic dynamics. Specifically, we treat the prefix $(x, a_1, \ldots, a_{h-1})$ as the current *state* $s_h$ and next step $a_h \sim \pi(\cdot \mid x)$ as the *action* taken by $\pi$ at $s_h$, resulting in the next state $s_{h+1}$. For problem $x$, with ground-truth response $y_x^\star$, we can evaluate the accuracy of $\pi$ by running a regular expression match on the final answer (Hendrycks et al., 2021): $\text{Rex}(y, y_x^\star) \mapsto \{0, 1\}$, *i.e.*, accuracy is given by $\mathbb{E}_{y \sim \pi(\cdot \mid x)} \left[ \text{Rex}(y, y_x^\star) \right]$. Now, given a dataset $\mathcal{D} = \{(x_i, y_{x_i}^\star)\}_i$ of problem-solution pairs, the main goal is to learn a good base policy by optimizing accuracy on $\mathcal{D}$. Next, we see how we can leverage the final answer verifier Rex available on $\mathcal{D}$ to train ORMs and PRMs. **Outcome reward model (ORM).** Given a response $y$, an ORM estimates the ground-truth correctness $\text{Rex}(y, y_x^\star)$. To train such a model we first take problems in $\mathcal{D}$, and collect training data of the form $\{(x, y \sim \pi(\cdot \mid x), \text{Rex}(y, y_x^\star)\}$. Then we train an ORM that takes as input a problem-response pair $(x, y)$ and predicts $\text{Rex}(y, y_x^\star)$. At test time, when $y_x^\star$ is unknown, the ORM is used to score candidate solutions revealed by test-time search. Given a base $\pi$, a *best-of-K* policy $\text{BoK}(\pi)$ is a policy that samples $K$ responses from $\pi$, scores them against ORM, and returns the one with the highest score. Note that, whenever ORM matches Rex, and the likelihood of $\pi$ solving problem $x$ is $p_x$, then for $\text{BoK}(\pi)$ this likelihood is $1 - (1 - p_x)^K$, making it a policy stronger than $\pi$.

**Typical process reward models (PRMs).** A PRM scores every step $a_h$ in a multi-step response $y \sim \pi$ (*e.g.*, in Lightman et al. (2023) PRMs are trained to score correct steps over incorrect and irrelevant ones). But, unlike ORMs, which only require Rex for data collection, PRM training data requires expensive step-level human annotations. Prior work (Wang et al., 2024; Setlur et al., 2024) attempted to scale process rewards automatically by sampling from the model to provide a heuristic understanding of when a step is actually correct. In particular, they evaluate a prefix by computing the expected future accuracy of multiple completions sampled from $\pi$, after conditioning on the prefix, *i.e.*, value function $Q^\pi$ (Eq. 1) from RL. Luo et al. (2024) use $Q^\pi$ as the PRM that assigns a score of $Q^\pi(s_h, a_h)$ to the action $a_h$, at state $s_h$. Similarly, $V^\pi(s_h) := \mathbb{E}_{a_h \sim \pi(\cdot \mid s_h)} Q^\pi(s_h, a_h)$.

$$Q^\pi(\underbrace{(x, a_1, \ldots, a_{h-1})}_{\text{state } s_h}, \underbrace{a_h}_{\text{action } a_h}) = \underbrace{\mathbb{E}_{a_{h+1}, \ldots, a_H \sim \pi(\cdot \mid s_h, a_h)} \left[ \text{Rex}\left((a_1, \ldots, a_H), y_x^\star\right) \right]}_{\text{likelihood of future success}}, \qquad (1)$$

**Using PRMs for beam search at test-time.** Given a PRM, a natural way to spend test-time compute is to use it as a reranker within a beam search procedure (Snell et al., 2024). For each problem, at step 0, a beam of maximum width $B$, is initialized with a single state consisting of just the problem. At a given step $h$, a beam contains partial responses unrolled till a set of states $\{s_i\}_{i=1}^B$. From each state $s_i$ in this set, $C$ independent actions or steps $\{a_{i,j}\}_{j=1}^C$ are sampled from $\pi(\cdot \mid s_i)$, each of which corresponds to a new state. Process rewards assign a score to every new state $(s_i, a_{i,j})$, and only the states corresponding to the top-$B$ values are retained in the beam for the next step.

## 3 HOW SHOULD WE DEFINE PROCESS REWARDS AND WHY?

Ultimately, we are interested in test-time search and RL methods that can most efficiently and reliably discover solution traces with the correct final answer, thus maximizing Rex. To this end, process rewards should serve as step-level supervision to maximize outcome-level Rex. Our position contrasts with conventional belief that process rewards should mainly evaluate mathematical correctness or relevance of individual steps (Lightman et al., 2023; Uesato et al., 2022), since LLMs might need to generate trivial or repetitive intermediate steps in order to discover a trace with the correct final answer. With this insight, in this section we approach the design of dense automated step rewards as a form of supervision to be used in conjunction with sparse outcome rewards in order to learn good policies.

In an MDP, a starting point to design state-level dense feedback that is eventually meant to optimize a sparse outcome reward is the notion of a ***reward-shaping function*** (Ng et al., 1999): in our case, a function that summarizes some statistic of the outcome reward attained in the future, under some policy rolled out from that state. Under this framework, in Sec. 3.1, we evaluate two different choices of the reward-shaping functions and show that advantages – not value functions (Wang et al., 2024; Luo et al., 2024) – that measure a notion of "progress" at each new step are more appropriate for exploration during search and RL. Then in Secs. 3.3 and 3.4, we show that progress is measured best under a policy $\mu$, different from the base policy $\pi$. We call this policy $\mu$, a ***prover policy***.

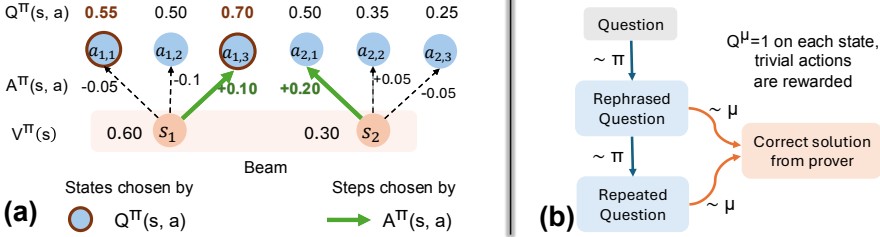

**Figure 2:** *Issues with using Q-values as process rewards*: **(a):** Unlike $A^\pi$, $Q^\pi$ mixes action evaluation with the $Q$-value of the previous state. Beam search with $Q^\pi$ exploits high-likelihood states, while adding $A^\pi$ (*e.g.*, $Q^\pi + \alpha A^\pi$ in Eq. 5) aids in exploring states reached by making actions that induce progress, *i.e.*, increase likelihood of success. **(b):** $Q^\mu$ from a strong prover $\mu$ can assign unmerited bonuses to trivial actions.

## 3.1 PROCESS REWARDS SHOULD BE ADVANTAGES, NOT VALUE FUNCTIONS

Setting aside how prover $\mu$ informs the **reward-shaping function**, we study a simplified setting of test-time beam search, and present some challenges with the reward design of Snell et al. (2024), which uses $Q^\pi$ of the base policy $\pi$ to assign rewards for a state-action pair $(s_h, a_h)$. Consider the example in Fig. 2(a), where from the 2 states in the beam, we sample 3 actions. If we only ***exploit*** the next states with highest values of $Q^\pi$, we would be comparing steps sampled from different states (*e.g.*, $a_{1,1}$ vs. $a_{2,1}$). Clearly, a reduction in $Q^\pi(s_1, a_{1,1}) - V^\pi(s_1)$, means that $a_{1,1}$ by itself has a negative effect of $-0.05$ on the probability of success from the $s_1$, whereas $a_{2,1}$ has a positive effect of $+0.20$ from $s_2$. However, expanding the beam based on absolute values of $Q^\pi$ retains the action that makes negative progress, and removes state $s_2$ from the beam (as the size of the beam is 2). In other words, $Q^\pi$ makes it hard to decouple the "evaluation" of an action (step), from the "promise" shown by the previous state. This will not be an issue for every problem, and particularly not when the beam capacity is unbounded, but under finite computational and sampling constraints, using $Q^\pi$ might retain states with potentially unfavorable steps that hurt the overall likelihood of success. On the other hand, when deciding what to retain in the beam, if we could also also ***explore*** the progress made by the previous step along with the likelihood of success $Q^\pi$, then we can address this tradeoff.

**How can we measure the "progress" made by a step?** One approach is to consider the relative increase/decrease in the likelihood of success, before and after the step. This notion is formalized by the advantage (Eq. 2) of a step under policy $\pi$. Furthermore, since advantages can attach either positive or negative values to a step, the base policy is trained not only when it generates a step that makes progress (where $A^\pi > 0$), but also when it fails to produce one, employing a "negative gradient" that speeds up RL training (Tajwar et al., 2024).

$$A^\pi(s_h, a_h) := Q^\pi(s_h, a_h) - V^\pi(s_h) = Q^\pi(s_h, a_h) - Q^\pi(s_{h-1}, a_{h-1}). \tag{2}$$

Since we view process rewards as reward-shaping functions in the MDP, they can be computed under any policy $\mu$, which can be the base policy. However, based on the example in Figure 2a, the reasons why $Q^\pi$ is a sub-optimal choice for process rewards also apply to $Q^\mu$. Nevertheless, we can possibly use advantage under $\mu$: $A^\mu$, which measures the progress made by a step to improve the likelihood of success under $\mu$. In that case, how should we choose this policy $\mu$, which we call the *prover policy*, and should it be necessarily different from base $\pi$? Before diving into the choice of $\mu$, we discuss: how should we use $A^\mu$ in conjunction with outcome rewards for improving $\pi$?

## 3.2 OUR APPROACH: PROCESS ADVANTAGE VERIFIERS (PAV)

For building an approach that uses process rewards $A^\mu$ together with the outcome reward Rex to improve the base policy $\pi$, we situate ourselves in the context of improving $\pi$ with online RL. If all we had was access to Rex on $\mathcal{D}$, the standard RL objective is given by:

$$\ell_{\text{ORM-RL}}(\pi) := \mathbb{E}_{x \sim \mathcal{D}, (a_1, \ldots, a_H) \sim \pi(\cdot | x)} \left[ \text{Rex} \left( (x, a_1, \ldots, a_H), y_x^\star \right) \right]. \tag{3}$$

Inspired by how reward bonuses are additive in RL (Ng et al., 1999; Bellemare et al., 2016), one way to use process rewards $A^\mu$ is to additively combine it with the standard RL objective in this way:

$$\ell_{\text{PAV-RL}}^{\pi'}(\pi) := \ell_{\text{ORM-RL}}(\pi) + \alpha \cdot \sum_{h=1}^{H} \mathbb{E}_{s_h \sim d_h^{\pi'}} \mathbb{E}_{a_h \sim \pi(\cdot | s_h)} \left[ A^\mu(s_h, a_h) \right] \tag{4}$$

The term in red is the difference in likelihoods of success of the prover $\mu$, summed over consecutive steps (a notion of ***progress***). Here, $d_h^{\pi'}$ denotes the distribution over states at step $h$, visited by the old

policy $\pi'$ (policy at previous iterate). Following policy gradient derivations (Sutton et al., 1999):

$$\nabla_\pi \ell_{\text{PAV−RL}}^{\pi'}(\pi)\Big|_{\pi'=\pi} = \sum_{h=1}^{H} \nabla_\pi \log \pi(a_h \mid s_h) \cdot \underbrace{(Q^\pi(s_h, a_h) + \alpha \cdot A^\mu(s_h, a_h))}_{\text{effective reward}} \tag{5}$$

At a glance, we can view $Q^\pi(s_h, a_h) + \alpha A^\mu(s_h, a_h)$ as the effective reward for step $a_h$ when scored against a combination of the outcome evaluation Rex, i.e., $Q^\pi$, and process rewards $A^\mu$. Thus, we can optimize Eq. 4 indirectly via **(a)** running beam-search against the effective reward; or **(b)** online RL where the policy gradients are given by Eq. 5. For either of these, we need access to verifiers that are trained to predict the advantage $A^\mu(s_h, a_h)$ under the prover. We refer to these verifiers as ***process advantage verifiers (PAVs)***. In Sec. D we describe how to train PAVs, but now we use the above formulation to reason about how to choose prover $\mu$ that is most effective at improving base $\pi$.

We also remark that the term in red resembles prior work on imitation learning via policy optimization (Ross & Bagnell, 2014; Sun et al., 2017), where the main aim is to learn a policy $\pi$ that imitates the prover $\mu$, or to improve upon it minimally. Of course, this is limiting since our goal is to not just take actions that perform at a similar level as $\mu$, but to improve the base policy even further, and using a combination of $Q^\pi$ **(exploitation)** and $A^\mu$ **(exploration)** is critical towards optimizing the goal.

**How should we choose the prover $\mu$?** Perhaps a natural starting point is to set the prover to be identical to the base policy, *i.e.*, $\mu = \pi$, which produces process rewards that prior works have considered Shao et al. (2024). However, setting $A^\mu = A^\pi$ in Eq. 5 results in exactly the same policy gradient update as only optimizing outcome evaluation Rex. Moreover, for a poor base policy $\pi$, where $Q^\pi \approx 0$ on most states, the term $A^\pi$ would also be $\approx 0$, and hence running beam search with the effective rewards would not be informative at all. Hence, ***a better approach is to use a different prover policy***, but a very weak prover $\mu$ will likely run into similar issues as a poor base policy. We could instead use a very capable prover $\mu$, but unfortunately even this may not be any better than optimizing (exploiting) only the outcome rewards, $Q^\pi$ either. To see why, consider a scenario where $\pi$'s response contains an intermediate step that does not help make progress towards the solution (*e.g.*, $\pi$ simply restates the question, see Fig. 2(b)). Here, $Q^\mu$ for a capable prover before and after this irrelevant step will be identical since $\mu$ can succeed from either step. This means that $\mu$ fails to distinguish steps, resulting in $A^\mu \approx 0$ in most cases. Training with this process reward during RL will then lead to gradients that are equivalent to those observed when purely optimizing $\ell_{\text{ORM−RL}}$. In fact, empirically, we observe that online RL with $Q^\mu$ from strong provers leads to polices that only produce re-phrasings of the question (App. G) and do not succeed at solving the question. Clearly, *any* policy different from the base policy cannot serve as a prover. So, how do we identify a set of good provers? Can they indeed be weaker than the base policy? We answer in the next sections.

> **Takeaway: What should process rewards measure during test-time search and online RL?**
>
> - Process rewards should correspond to progress, or **advantage**, as opposed to absolute $Q$-values, for a better explore-exploit tradeoff during beam search and online RL.
> - Advantages should be computed using a **prover** policy, different from the base policy.

### 3.3 ANALYSIS IN A DIDACTIC SETTING: LEARNING A PLANTED SUB-SEQUENCE

In this section, we aim to characterize prover policies that are effective in improving the base policy. To do so, we first introduce a didactic example, representative of real reasoning scenarios to illustrate the main intuition. Then, we will formalize these intuitions in the form of theoretical results.

**Didactic example setup.** Given an unknown sub-sequence $y^\star$ consisting of tokens from vocabulary $\mathcal{V} := \{1, 2, \ldots, 15\}$, we train a policy $\pi$ to produce a response which contains this sub-sequence. The task completion reward is terminal and sparse, *i.e.*, $r(y, y^\star) = 1$ for a $y$ if and only if $y^\star$ appears in $y$. By design, the reward $r(y, y^\star)$ resembles outcome reward Rex$(y, y_x^\star)$ in Sec. 2. The prover policy $\mu$ is a procedural policy, parameterized by a scalar $\gamma > 0$ (details in App. B). As $\gamma$ increases, the performance of $\mu$ improves and $\to 1$ as $\gamma \to \infty$. For simplicity, we assume oracle access to ground-truth $A^\mu$ and $Q^\pi$, and alleviate errors from learned verifiers approximating these values.

**(1) RL with effective reward $Q^\pi + \alpha A^\mu$ is $10\times$ more sample-efficient than only outcome reward.** In Fig. 3(a), we first note that training $\pi$ with this effective reward under a prover $\mu$ with strength $\gamma = 10$, produces optimal performance (100% accuracy) in $< 1$k iterations, despite starting from a mediocre initialization for $\pi$ ($\gamma = 5.0$). Training with only outcome reward is ineffective and requires

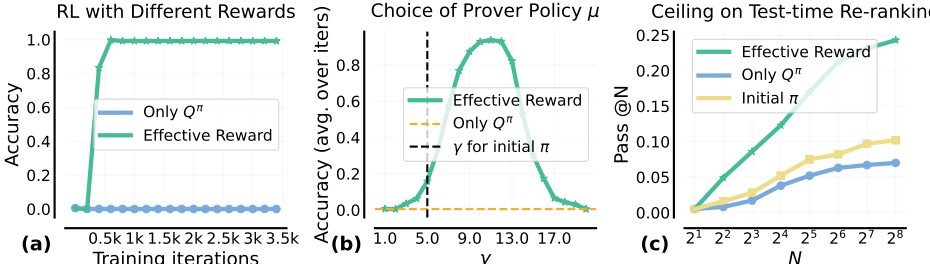

**Figure 3:** *Results for our didactic analysis:* **(a):** We train base policy via RL with either the effective reward $Q^\pi + \alpha A^\mu$, or the more typical $Q^\pi$. **(b):** We vary the strength $\gamma$ of the prover policy $\mu$ used to compute advantages $A^\mu$ in the effective reward, and plot the base policy accuracy averaged over the RL run. **(c):** We plot the max score out of $N$ responses (Pass @N) sampled *i.i.d.* from an undertrained base policy (iter 100) .

>10k iterations. More importantly, in Fig. 3(b), we note that effective rewards only help for a set of provers, in $\gamma \in [8.0, 15.0]$. Outside this range, we observed $A^\mu \approx 0$ on most states, either because $\mu$ was poor (small $\gamma$) and was unable to generate $y^\star$ even when $\pi$ got the sequence partially correct, or because $\mu$ was strong (large $\gamma$) that it generated $y^*$ with almost equal likelihood from all prefixes.

**(2) Effective reward improves Pass @N by** $5\times$ **over only outcome reward.** We report the "Pass @N" performance in Fig. 3(c), which measures if any out of $N$ *i.i.d.* samples from $\pi$ have reward 1 and hence, presents the ceiling on the performance of any test-time search method that picks a single response from multiple draws (*e.g.*, as in Best-of-N). For a policy trained with the effective reward for 100 iterations, the Pass @N performance grows $5\times$ faster with $N$, compared to the policy trained with only the outcome reward. Due to only sparse feedback, the latter policy does not learn to sample partially correct $y^\star$, whereas a policy trained with the effective reward produces partially correct $y^\star$, and is able to sample the complete $y^\star$ with higher likelihood during Pass @N.

> **Takeaway: Online RL with process rewards from different prover policies.**
>
> Effective rewards $Q^\pi + \alpha A^\mu$ from prover $\mu$: (i) improve sample efficiency of online RL, and (ii) yield policies with better Pass @N performance, over using only outcome rewards. But, advantages of very capable or poor $\mu$ do not improve base policy beyond outcome rewards.

### 3.4 Theory: Provers Complementary to Base Policy Boost Improvement

From our didactic analysis, it is clear that process rewards $A^\mu$ under different provers $\mu$ disparately affect the base policy that optimizes $Q^\pi + \alpha A^\mu$ via online RL. We now present a formal analysis of why this happens and characterize a class of provers that can guarantee non-trivial improvements to the base policy. For simplicity, we assume oracle access to $Q^\pi$, $A^\mu$ at every state-action pair $(s_h, a_h)$ and prove our result in the tabular RL setting, where the policy class is parameterized using the softmax parameterization in Agarwal et al. (2021). Proofs for this section are in App. F.

**Main intuitions.** We expect a prover $\mu$ to improve a base policy $\pi$ only when $\mu$ **is able to** *distinguish* **different actions taken by** $\pi$, by attaining sufficiently varying advantage values $A^\mu(s_h, a)$ for actions $a$ at state $s_h$. This can be formalized under the notion of sufficiently large variance across actions, $\mathbb{V}_{a \sim \pi}[A^\mu(s_h, a)]$. In that case, can we simply use a policy with large advantage variance under any measure? No, because when the prover $\mu$ ranks actions at a given state very differently compared to the base policy $\pi$ (e.g., if $A^\mu$ and $A^\pi$ are opposite), then effective rewards $Q^\mu + \alpha A^\pi$ will be less reliable due to conflicting learning signals. Thus, we want $\mathbb{E}_\pi[\langle A^\mu, A^\pi \rangle]$ to not be too negative, so that $\mu$ **and** $\pi$ **are reasonably** *aligned* on their assessment of steps from $\pi$.

In Theorem 3.1, we present our result on policy improvement where the base policy is updated with natural policy gradient (Kakade, 2001): $\pi_{t+1}(a \mid s_h) \propto \exp(\gamma \cdot (A^\pi(s_h, a) + A^\mu(s_h, a)))$. We note that in this idealized update rule, swapping $Q$ values (of $\mu$ or $\pi$) with advantages does not affect the update, since we only subtract an action-independent offset $V(s_h)$. Consequently, the analysis only uncovers good choices for the prover policy $\mu$ for process reward $A^\mu$, and is orthogonal to the design consideration of advantages or $Q$-values as process rewards. Theorem 3.1 formalizes our intuition by showing that policy improvement at iteration $t$, grows as the variance in $A^\mu$ values increases (higher distinguishability) and reduces when $A^\mu$ and $A^\pi$ become extremely misaligned. This will then allow us to discuss a special case for the case of Best-of-K policies as provers as an immediate corollary.

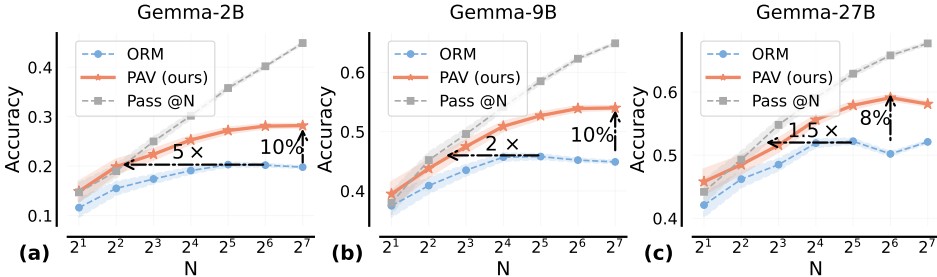

**Figure 4:** *For test-time search, PAVs are* $8 - 10\%$ *more accurate and* $1.5 - 5\times$ *more compute efficient over ORMs:* On samples from **(a)** Gemma-2B , **(b)** 9B , and **(c)** 27B SFT policies, we run test-time beam search with the effective reward $Q^\pi + \alpha A^\mu$ (PAV), where $\mu$ is the Bo4($\pi$) policy. We compare beam search performance with best-of-N, re-ranking with a trained outcome verifier (ORM), or the oracle Rex (Pass @N).

**Theorem 3.1** (Lower bound on policy improvement). *For base policy iterate $\pi_t$, after one update step with learning rate $\gamma \ll 1$, $\exists$ constant $C > 0$ s.t. over state distribution $\rho$ the following holds:*

$$\mathbb{E}_{s \sim \rho} \left[ V^{\pi_{t+1}}(s) - V^{\pi_t}(s) \right] \geq C\gamma \cdot \underbrace{\mathbb{E}_{s \sim \rho} \mathbb{V}_{a \sim \pi_t} \left[ A^\mu(s,a) \right]}_{\text{distinguishability from } \mu} + C\gamma \cdot \underbrace{\mathbb{E}_{s \sim \rho} \mathbb{E}_{a \sim \pi_t} \left[ A^\mu(s,a) A^{\pi_t}(s,a) \right]}_{\text{alignment between } \pi_t \text{ and } \mu} \quad (6)$$

It may seem that the base policy $\pi$ can only learn from an improved prover $\mu$, but our result shows that a **weak prover can also amplify a stronger base policy**, since a weak prover $\mu$ may have a lower average of $Q^\mu$ under its own measure, but still have higher variance across $Q^\mu$ (compared to $Q^\pi$) when evaluated under $\pi$ (see Proposition F.1 in App. F.5 for formal discussion). This tells us that rewarding progress under a prover is different from typical knowledge distillation or imitation learning algorithms (Hinton, 2015; Rusu et al., 2015) that in most cases remain upper bounded by the performance of the stronger teacher. So provers cannot be characterized purely by strength, what is a class of provers that is a reasonable starting point if we were to improve any base policy $\pi$?

**The policy class of "Best-of-K" (computed over base policies) contain complementary provers.** A good starting point to identify good provers for a base policy $\pi$, is the class of Best-of-K policies or BoK($\pi$). Recall from Sec. 2 that the performance of BoK($\pi$) increases monotonically with $K$. Thus, we can vary $K$ to obtain a class of policies with increasing strengths, starting from $\pi$. Applying Theorem 3.1 to this class, we arrive at Remark 3.1 that recommends using BoK($\pi$) with $K > 1$ as a prover policy for a poor base policy $\pi$. Conversely , when $Q^\pi(s,a) \approx 1$ , then increasing $K$ too much can hurt variance on those states (see Appendix F.4). In the next section, we empirically note that the policies in the class of BoK($\pi$) indeed induce different performance gains when used as prover policies, and we find Bo4 to be a good choice for test-time search over most base policies.

**Remark 3.1.** *When $Q^\pi(s,a) = O(1/K), \forall s, a$, using BoK($\pi$) as a prover for base $\pi$ improves distinguishability (and improvement) by $\Omega(K^2)$, and makes alignment worse at most by $O(K)$.*

> **Takeaway: Formal characterization of good prover policies that improve the base policy.**
>
> Provers with advantages that can **distinguish** actions taken by the base policy (more strongly than the base policy itself) but are **not too misaligned** from the base, boost improvements on each update of the base policy. We call such policies *complementary provers*. BoK($\pi$) for any base policy $\pi$ for $K > 1$ can provide a good starting choice of prover policies.

## 4 TEST-TIME: EFFICIENTLY SCALING TEST-TIME COMPUTE WITH PAVS

Now, we study how process verifiers can scale up test-time compute. While our derivations from Sec. 3.2 were with RL, we can also use the *effective reward* $Q^\pi(s_h, a_h) + \alpha \cdot A^\mu(s_h, a_h)$ for running beam search over intermediate steps sampled from base policy $\pi$. In the end, we select the final response in the beam using an ORM. In practice, we need to train PRMs that predict $Q^\pi$ (we call these PQVs) and PAVs that predict $A^\mu$, procedures that we discuss in App. D. For now, we assume access to trained ORMs, PQVs and PAVs and discuss test-time search results. For clarity, we abuse notation and refer to the estimated effective reward (ORM + $\alpha$ PAV) as PAV directly.

**Setup**. We finetune Gemma 2B, 9B, and 27B (Gemma Team et al., 2024) on MATH (Hendrycks et al., 2021) via supervised fine-tuning (SFT) to get three base policies. The set of provers consists of the three base SFT policies themselves as well as their best-of-K policies for different values of $K \in \{2^0, \ldots, 2^5\}$. Additional details for the experiments in this section are in App. C.

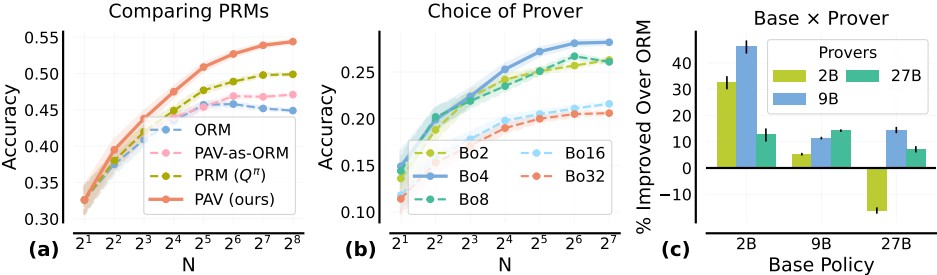

**Figure 5:** *Comparing PAVs with search baselines and ablating over the prover policy:* **(a):** We compare beam search over Gemma 9B SFT, using either effective reward (PAV), or $Q^\pi$ (Snell et al., 2024), and report best-of-N performance where the re-ranker is either the ORM or PAV-as-ORM. **(b):** For base $\pi$ Gemma 2B SFT, we run beam search with the effective reward where the prover is $\text{BoK}(\pi)$ for different values of $K$. In both $(a)$, $(b)$ the x-axis scales the size of the beam or $N$ for best-of-N. **(c):** For each base policy in the set: Gemma 2B, 9B, 27B SFTs, we run beam search with PAVs (beam size of 16) where the prover is another policy from the same set.

## 4.1 PAVS SCALE TEST-TIME COMPUTE BY $5 - 10\times$ OVER ORMS

**Result 1: PAVs are more compute efficient than ORMs.** In Fig. 4, we plot the performance of beam search with PAVs for different sizes of the beam $N$, and compare it with best-of-$N$ using ORMs, *i.e.*, sampling $N$ complete solutions from the base policy and returning the one with the highest ORM score. Our comparisons report *compute efficiency* of PAVs over ORMs: ratio of total compute needed by PAVs to obtain the same performance as running best-of-128 with ORM. Accounting for the additional compute needed for beam search (since each element in the beam samples $C = 3$ next steps, before scoring and pruning the beam), PAVs scale the compute efficiency by **10×** over ORMs for Gemma-2B, 9B base policies, and by 5× for Gemma-27B. For all base policies $\pi$, we use $\text{BoK}(\pi)$ with $K = 4$ as the prover policy. We also compare performance with beam search using process verifiers that only predict $Q^\pi$, and best-of-N where the ORM is replaced with PAV (PAV-as-ORM). At $N = 128$, similar to Luo et al. (2024), we note a similar gain of 4% for "PAV-as-ORM" Fig. 5(a) over only ORMs, for base Gemma-9B $\pi$. When comparing beam search with $Q^\pi$ (Snell et al., 2024), we find that PAVs scale compute efficiency by 8×. Evidently, advantages from the prover in the effective reward positively impact the beam search. Why does $A^\mu$ help, and for what choice of the prover $\mu$?

**Result 2: Beam search with too weak/strong provers is sub-optimal.** In Fig. 5(b), for the setting when the base policy $\pi$ is a Gemma-2B SFT model, we compare beam search with PAVs where the provers are given by $\text{BoK}(\pi)$, for different values of $K$. Recall that as $K$ increases, $\text{BoK}(\pi)$ becomes stronger. Corroborating our analysis in Sec. 3.4, our results show that neither too weak (Bo2) or too strong (Bo32) provers perform best. Instead, across all values of $N$, we find Bo4 to be dominant. The advantage values $A^\mu \approx 0$ on all steps for very large $K$, since $Q^\mu(s_h, a_h) = 1 - (1 - Q^\pi(s_h, a_h))^K \to 1$ on all steps, as we increase $K$. Hence, ***in order to succeed we need an intermediate-level prover policy***. We make similar observations in Figure 5(c) where we take the three base policies (Gemma 2B/9B/27B) and evaluate beam search with PAVs at $N = 16$ with the Bo4 over the SFT policies acting as provers. We find that for the 2B and 9B base models, the 9B and 27B provers are most effective respectively, whereas for the 27B model, *surprisingly a weaker 9B policy is more effective than the stronger 27B model.* The weaker model presumably offers a complementary signal that distinguishes between different actions taken by 27B, aligning with our theoretical observations in Sec. 3.4.

**Result 3: Advantages from the prover policy enable exploration.** As discussed in Sec. 3.1, advantage $A^\mu$ measures the progress made by an action agnostic of the value of the previous state, where as $Q^\pi$ measures the promise of a particular state. Given a finite capacity beam, our effective reward (Eq. 5), which linearly combines $Q^\pi$ and $A^\mu$ *induces a better tradeoff between exploring new prefixes (states) from where progress can be made and exploiting currently known prefixes with high Q-values.* Exploration at initial steps is critical to ensure that the beam at later steps covers diverse partial rollouts each with a high likelihood of producing the correct answer. Thus over-committing to the beam with actions from the same state, regardless of the progress made by each can prove to be sub-optimal over a selection strategy that balances rewarding previous actions $A^\mu$ and current states

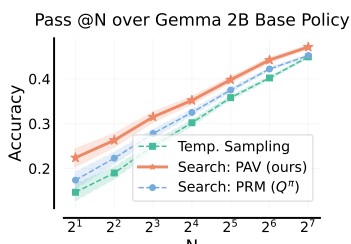

**Figure 6:** Beam search with PAVs improves Pass@N over typical PRMs.

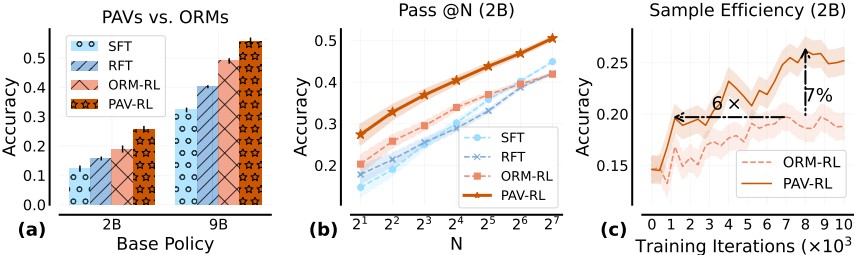

**Figure 7:** *PAVs as dense rewards in RL is* 6× *more sample efficient than ORMs, and gains* +7% *on accuracy:*
**(a)** We report the performance of a base policy trained using RL with effective rewards (PAV-RL), or only outcome rewards (ORM-RL), and baselines SFT, RFT. **(b):** For the policies trained in (a) we report the best-of-N performance where the oracle reward Rex is used to rank $N$ candidates sampled from the base policy (Pass @N). **(c):** Across training iterations, we report the test performance of policies trained with PAV-RL and ORM-RL.

$Q^\pi$. Indeed, we observe in Fig. 6, beam search with PAV enhances pass@N performance of the base Gemma 9B model over beam search with $Q^\pi$ and *i.i.d.* sampling.

> **Takeaways: Scaling test-time compute with process advantage verifiers.**
>
> - Beam search with PAVs boosts accuracy by >8% & compute efficiency by 1.5-5x over ORMs.
> - Utilizing Best-of-K policies (corresponding to the base policy) as provers improves the performance of test-time beam search over the base policy. Optimal provers appear at $K > 1$.

## 5 TRAINING-TIME: SCALING RL SAMPLE-EFFICIENCY WITH PAVS

We can also use PAVs to train policies via online reinforcement learning (RL), by using the effective reward $Q^\pi + \alpha A^\mu$ as dense, per-step rewards. In this section, we compare the sample efficiency of PAV-RL (i.e., $\ell_{\text{PAV-RL}}$ in Eq. 4) with standard ORM-RL (i.e., $\ell_{\text{ORM-RL}}$ in Eq. 3) on Gemma 2B and Gemma 9B SFT models, which are further optimized via rejection finetuning (RFT) (Yuan et al., 2023), before using them to initialize RL. To our knowledge, no prior work has successfully demonstrated the use of dense per-step feedback with a process reward model for RL, and we present the first significant set of results establishing the efficacy of this approach. We show that PAV-RL is much more sample-efficient, and enjoys a higher ceiling on the performance of any test-time re-ranker. Additional details for the experiments are in App. E.

**Result 1: PAV-RL is > 7% better than ORM-RL in test accuracy, and 6× sample efficient**. In Fig. 7(a), we report the test accuracies of Gemma 2B and 9B models trained with SFT, RFT, ORM-RL and PAV-RL. PAV-RL improves the RFT policy by 11% for 2B, and 15% for 9B, with > 7% gain over ORM-RL in both cases. Not only do the effective rewards from PAV improve the raw accuracy after RL, this accuracy is attained 6× faster (see Fig. 7(c)) for the 2B run and similarly for the 9B RL run (see App. E). For both 2B and 9B, RL runs, we experiment with two options for the prover policy: **(i)** 2B SFT policy; and **(ii)** 9B SFT policy. While both of these provers rapidly become weaker than the base policy within a few gradient steps of RL, a fixed PAV trained with each of these provers is able to still

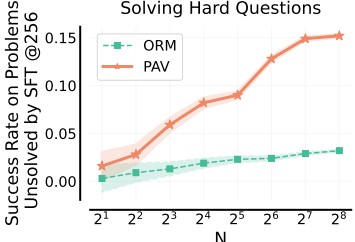

**Figure 8:** Amongst hard problems that remain unsolved by Best-of-256 over SFT, we check how many are solved by Best-of-N over PAV-RL or ORM-RL.

sustain performance gains in RL. More interestingly, we find that the 2B SFT policy serves as the best choice of the prover for both 2B and 9B policies. This observation that a weak prover can still improve the base policy corroborates our results in the didactic setup and our analysis in Sec. 3.4.

**Result 2: PAV-RL achieves higher performance ceiling on test-time re-ranking.** In Fig. 7(b), for the Gemma 2B model, we plot the Pass @N performance for each method, and find (i) Pass @N is higher (> 7%) for PAV-RL, compared to ORM-RL, for any $N \leq 128$; and (ii) the rate at which Pass @N improves for PAV-RL is higher than ORM-RL. Both these trends are consistent with our observations in the didactic analysis in Sec. 3.3. Notably, for $N \geq 64$, ORM-RL is worse than the SFT policy, perhaps due to lower entropy over the distribution at the next step resulting in non-diverse candidates. So, why does PAV-RL deliver a policy that can generate more diverse candidates, and does not suffer from the low entropy problem in ORM-RL? We answer this with a key insight on how *the primary benefit of PAVs is to promote efficient exploration.*

**Result 3: PAVs improve exploration and discover correct solutions to novel problems.** The ORM rewards downweight all steps in an incorrect rollout equally during RL, whereas the effective reward in PAVs, up-weights steps that make progress under the prover, even when the complete rollout is incorrect. This increases the coverage over individual steps that can be proposed at a given prefix. This mechanism for exploration is analogous to test-time search in Sec. 4.1. Hence, the directed supervision from PAVs improves sample-efficiency throughout the course of training (Fig. 7(c)). In fact, we also find that combining the PAV-RL policy with test-time beam search is able to solve a substantially larger number of *new* problems within smaller compute budgets ($N$=16,32) that the SFT policy cannot solve with a much larger budget $N$=256 (Fig. 8).

---

**Takeaway: RL with process advantage verifiers (PAVs) as dense rewards**

- Using trained PAVs as dense rewards in RL boosts scales sample efficiency by 6×, compared to only using sparse ORM rewards, and results in policies with a higher Pass @N performance.
- Advantages from a complementary prover policy improves the sample efficiency of exploration in RL, and produces policies that can discover solutions to hard novel questions.

---

## 6 RELATED WORKS

Outcome reward models (ORMs) (Cobbe et al., 2021b; Uesato et al., 2022) are used for best-of-$N$ search where ORMs are trained to assess correctness using binary classification (Cobbe et al., 2021a; Yu et al., 2023), preference optimization (Hosseini et al., 2024), or next-token prediction (Zhang et al., 2024). In contrast, we propose process reward models (PRMs), that assign scores to intermediate solutions to improve performance with beam-search and enable online RL with dense rewards.

To address issues of sparse feedback in ORMs, recent works (Lightman et al., 2023; Uesato et al., 2022) trained process reward models (PRMs) to densely predict incorrect steps in a multi-step reasoning trace. Since human data collection for process labels is not scalable enough, recent work (Wang et al., 2024; Luo et al., 2024) used automated supervision to annotate steps with $Q$ values under the base policy, *i.e.*, the PRMs score a step with the likelihood of future success, when continuing to sample from the step. While $Q$-value PRMs in Lightman et al. (2023); Luo et al. (2024) were mainly used as verifiers for re-ranking, Snell et al. (2024) used them for test-time beam search. Shao et al. (2024) uses PRMs for RL but found a gain of only $1-2\%$ with PRMs. In our work, we question solely relying on $Q$-values or advantages of the base policy, and find that measuring progress (i.e., advantages) under a different prover policy can amplify exploration, thus boosting test-time search and RL. Our data collection strategy is similar to (Setlur et al., 2024; Hwang et al., 2024) (i.e., identify "first pits" in traces), but these works only use it to collect preference pairs. Beyond all these, we also characterize which policy to use for computing advantages. Trained outcome or process verifiers can optimize policies similarly to RLHF (Ouyang et al., 2022). Prior works (Uesato et al., 2022; Havrilla et al., 2024; Shao et al., 2024) showed limited gains from PRMs compared to ORMs or ground-truth rewards (Rex). Notably, Havrilla et al. (2024) found expert iteration (Anthony et al., 2017) superior for exploration. In contrast, our work demonstrates that PRMs, combined with outcome rewards in online RL, improve performance by +7% and boost sample efficiency by 6×.

Finally, the idea of improving a sub-optimal expert by mixing policies, as in structured prediction (Chang et al., 2015), or using a "guide" policy for rollouts (Chang et al., 2023), relates to our work. However, we show that even weak prover policies can enhance a base policy by distinguishing its steps through advantages, without needing to surpass the base directly.

## 7 CONCLUSION

We explore process rewards to improve outcome-level correctness of a base policy, and show that defining them as advantages of a distinct prover policy enhances exploration and accuracy in math reasoning tasks. We formally characterize the set of good (complementary) prover policies and train process advantage verifiers (PAVs) to predict their advantages. For the base policy, PAVs improved search compute-efficiency by 1.5–5× and accuracy by 8%, while dense online RL with PAV rewards boosted sample efficiency by 5–6× and accuracy by 6%. Challenges remain in designing optimal prover policies and addressing PAV fitting errors, highlighting avenues for future research, such as joint optimization of prover and base policies or alternative advantage estimation methods.

## 8 REPRODUCIBILITY STATEMENT

For our experiments on the didactic setup in Section 3.3, we provide details in Appendix B. For experiments on training process advantage verifiers (PAVs) we provide details in Appendix D. Details for experiments that use the trained PAVs for test-time search in Section 4, are provided in Appendix C. Finally, for our experiments on training the base policies with PAVs as dense rewards in reinforcement learning (Section 5), we provide details in Appendix E. All proofs for our theoretical claims can be found in Appendix F. The pretrained model checkpoints for Gemma-2B, 9B, and 27B used in this work are available publicly at https://huggingface.co/google, and the MATH dataset we use from Hendrycks et al. (2021) is also public here: https://github.com/hendrycks/math.

## ACKNOWLEDGEMENTS

The authors would like to thank Charlie Snell, Yi Su, Katherine Heller, and Virginia Smith for feedback on an earlier version of this paper. We also thank Ahmad Beirami, Sergey Levine, Victor Veitch, Idan Shenfeld, Arian Hosseini, Stephen Pfohl, Xiangyu Qi, Tianhe Yu, and Christina Baek for technical discussions. AS and CN also thank Preston Robinette, Sho Kannan, Tianze Shi, Diana Mincu, Hritik Bansal, and Liangchen Luo for code, infrastructure and data analytics support.

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

# Appendices

## A   ADDITIONAL RELATED WORK

In this section, we expand on the related works we discuss in Section 6. Our work is closely related to three categories of works that aim to improve the math reasoning capabilities of large language models. First, we look at works that train verifiers to provide outcome level feedback (Cobbe et al., 2021b; Hosseini et al., 2024; Zelikman et al., 2022; Singh et al., 2023b) on the correctness of the full response (ORM). Next, we look at works that alleviate issues with sparse feedback in ORMs, and instead train process reward models, that can perform credit assignment. They are trained either through human annotations (Lightman et al., 2023; Uesato et al., 2022), or automated forms of supervision (Snell et al., 2024; Wang et al., 2024; Luo et al., 2024). While most PRMs and ORMs have been used to collect supervised finetuning data or for test-time verfication, a selected set of recent works also experiment with using them for training LLMs with online RL (Wang et al., 2024; Uesato et al., 2022; Shao et al., 2024). Below, we discuss each of these categories separately.

**Outcome reward models** (ORMs) or verifiers (Cobbe et al., 2021b; Uesato et al., 2022) are commonly used to improve the test-time performance using Best-of-N, where we generate multiple candidate solutions from the base policy (LLM), rank them using verifier, and pick the best one. ORMs are trained to assess correctness of a solution either using binary classification (Cobbe et al., 2021a; Yu et al., 2023), preference optimization using DPO (Hosseini et al., 2024), or next-token prediction (Zhang et al., 2024). Furthermore, prior works train LLMs on self-generated data using ground-truth outcome rewards (Rex), either via supervised fine-tuning (Zelikman et al., 2022; Singh et al., 2023a; Yuan et al., 2023), or online RL (Bi et al., 2024). In contrast to these approaches, our work focuses on process reward models (PRMs) for improving performance with beam-search at test time as well as online RL where we jointly maximize the effective reward in equation 5 which linearly combines both Rex (outcome supervision) and process supervision in the form of advantages under a prover policy, distinct from the base policy.

**PRMs and credit assignment.** Several works focus on training step-level PRMs on math reasoning tasks, either using human labels (Lightman et al., 2023) or automated LLM-generated data to estimate value functions (Wang et al., 2024; Luo et al., 2024). Our work also focus on automated data collection for PRMs but empirically argues for additionally using the advantage function as step-level rewards, with a conceptual explanation focused in Section 3.1. Several prior works have explored step-level search algorithms with PRMs, such as beam search (Snell et al., 2024), heuristic greedy search (Ma et al., 2023), and reward-balanced tree search (Wu et al., 2024). Hwang et al. (2024); Setlur et al. (2024) use advantages to identify the "first pit" in an incorrect reasoning trace. Specifically, they collect data by computing advantages at each step using Monte Carlo rollouts. Then in an incorrect trace, they identify the step with the least advantage, and use the prefix of that step to construct preference pairs for offline direct preference optimization (Rafailov et al., 2023). In contrast, our work computes advantages under a prover policy, that we formally characterize, and use the computed advantages for improving test-time search and efficiency of online reinforcement learning.

**Online RL for math reasoning.** Once we have a trained outcome or process verifier, it is natural update a policy by optimizing it against the learned signal, similar to how learned reward models are optimized in RLHF (Ouyang et al., 2022). In the context of math reasoning, Uesato et al. (2022); Havrilla et al. (2024); Shao et al. (2024) trained policies with RL, experimenting with both dense and sparse rewards. In all three instances, the gains observed by using PRMs trained to predict step-level correctness (similar to Lightman et al. (2023)) is quite small, compared to simply using trained ORMs, or the ground-truth outcome supervision Rex. In fact, Havrilla et al. (2024) states that the only algorithm that does well is a form of expert iteration Anthony et al. (2017), which does not inhibit exploration as severely as some other approaches they compare with. Our work presents one of the first results, where trained PRMs, used in conjunction with the outcome rewards during online RL, result in policies with substantially higher (+7%) performance, than the one trained only with outcome supervision. Our results also indicate a 6× sample efficiency boost for online RL, with our trained PRM (PAV).

**Connections to imitation learning through RL.** Finally, in addition to the above three sets of papers, we note that the idea of mixing potential functions from different policies $\mu$ and $\pi$, in order

Planted sub-sequence $y^\star$ : [3 6 0 3 5]

Samples from policy corresponding to $\gamma = 15$        Samples from policy corresponding to $\gamma = 100$

```
sample=[ 3  6  0  3  5 14 14 14 14 14]   reward=1.0        sample=[ 3  6  0  3  5 14 14 14 14 14]   reward=1.0
sample=[ 3  6  0  4  0  3  1 11  7  3]   reward=0.0        sample=[ 3  6  0  3  5 14 14 14 14 14]   reward=1.0
sample=[ 3  6  0  3  5 14 13  3  2 11]   reward=1.0        sample=[ 3  6  0  3  5 14 14 14 14 14]   reward=1.0
sample=[ 3  6  0 13  5  3  1  3 14  1]   reward=0.0        sample=[ 3  6  0  3  5 14 14 14 14 14]   reward=1.0
sample=[12  8  3  6 14  3  6  0  3  5]   reward=0.0        sample=[ 3  6  0  3  5 14 14 14 14 14]   reward=1.0
```

**Figure 9:** *Pictorial description of our planted sub-sequence didactic setup*: An example showing five samples drawn *i.i.d.* from a very strong $\gamma = 100$, and relatively weaker $\gamma = 15$ policies in our didactic setup.

to improve upon a sub-optimal expert $\mu$ appears in Chang et al. (2015), but this work considers the structured prediction problem which is vastly different from our setting. Related to this, is the work by Chang et al. (2023), which uses a "guide" policy to rollout from prefixes generated by a base policy. The base policy can now imitate the guide by cloning those rollouts, and eventually surpass. Our work also uses a prover policy which can complete rollouts from states where the base policy fails. But, we also show that weak provers in many cases are able to improve the base policy, or search over its responses, better than a stronger prover policy. We tie this observation to the insight that the main goal of the prover policy is to distinguish steps taken by the base policy, as measured by advantages under the prover. Thus, we do not require the prover policy to be something better than the base, that the base can rely on to clone rollouts that lead to correct final answers.

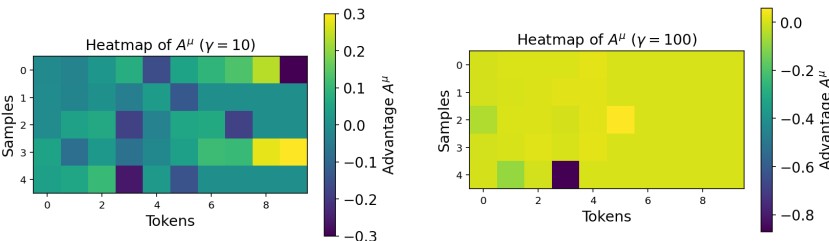

**Figure 10:** *Comparing heat map of advantages from weak ($\gamma = 10$) and strong provers ($\gamma = 100$):*. When the strength of the prover policy increases, the magnitude of step-level rewards reduces (since $A^\mu \approx 0$), as noted in the differences between the left (weak prover) and right plots (strong prover).

**Magnitudes of step-level rewards for weak vs. strong prover policies.** For 5 random samples drawn from the base policy, each with 10 tokens, we plot the values of the advantage $A^\mu$, at each token of each sample. In Figure 10, we plot heatmaps of the advantages for two different choices of the prover policy $\mu$, one with $\gamma = 10$ (weak prover), and the other with $\gamma = 100$ (strong prover). For the weak prover, the advantage magnitudes are much higher, since the weaker prover policy is more likely to complete a partially correct generation over an incorrect one, *i.e.*, a generation where the last set of tokens in the prefix partially matches the planted sub-sequence. On the other hand, when the strength of the prover policy increases, the magnitude of token-level rewards reduces ($A^\mu \approx 0$) since it can complete the solution with nearly equal likelihood from both correct and incorrect prefixes sampled from the base policy. Thus, RL training with the effective rewards (equation 5) computed using strong provers performs similarly to RL training with only outcome rewards, as noted in Figure 3b.

## B    DIDACTIC ANALYSIS

We consider sequences of length 10 from a 15-token vocabulary $\mathcal{V} := \{0, 1, 2, \ldots, 14\}$, where the end-of-sequence token is given by 14. Given an unknown planted sequence $y^\star$ (in Figure 9), we train a policy $\pi$ with on-policy policy gradient, where the reward is terminal and sparse, *i.e.*, for $y \sim \pi$ we have $r(y, y^\star) = 1$ if and only if $y^\star$ appears in $y$, and 0 otherwise (Figure 9). The policy $\pi$ in our experiments is represented by a multi-layer neural network, similar to the MADE architecture (Germain et al., 2015). The prover policy $\mu$ is parameterized by a scalar $\gamma > 0$. In particular, at any state $s$, where the last $k$ tokens leading up to $s_h$ match first $k$ tokens of $y^\star$, then:

$$\mu(y^\star_{k+1} \mid s) \propto \gamma,$$

and uniform on all other tokens. Thus, as $\gamma$ increases, the performance of $\mu$ improves and $\rightarrow 1$ as $\gamma \rightarrow \infty$. For our experiments, we assume oracle access to ground-truth advantage and Q-values, thus mitigating any confounding issues due to usage of a learned verifier.

**Training details.** We use effective rewards with $\alpha = 1$, and use the gradient in equation 5 to update the policy via policy gradient iterations. For the ORM runs, where we only use the outcome reward $r$, the policy gradient is equivalent to the case where $\alpha = 0$ in equation 5. We train for 10,000 iterations in both cases, with a batch size of 64, and a constant learning rate of $1e - 3$ for the Adam optimizer. The RL runs are initialized with a supervised finetuned policy. For this we take a randomly initialized network, based on the MADE architecture (Germain et al., 2015), with 3 layers, and 128 hidden units in each. Then we train it with supervised next-token prediction loss for 50 iterations on a dataset of 3200 samples from a weak policy ($\gamma = 5.0$).

## C ADDITIONAL DETAILS: EXPERIMENTS ON TEST-TIME SEARCH WITH PAVS

**Implementation details.** For our experiments in Sec. 4, we consider three base policies: Gemma 2B, 9B and 27B. We finetune each of these on the MATH (Hendrycks et al., 2021) dataset. The finetuning is done for 5000 iterations, with a batchsize of 32, with a maximum learning rate of $5e - 6$ for 2B, 9B and $5e - 7$ for 27B policy. We used the Adam optimizer, with a linear warm up and cosine decay learning rate schedule. The linear warm up is done for the first 500 iterations only. Given the SFT checkpoints, we next train PAVs using the procedure in Sec. D. We do this for a class of provers, which include the base policies themselves. As we discuss in Sec. 4, the prover class also includes the best-of-K policy for $K$ in $\{2, 4, 8, 16, 32\}$.

We used a hold out validation set to ascertain the value of $\alpha$ in the effective reward. For each base policy we run beam search with a beam size of 16 on this hold out validation set, and using the base policy itself as a prover, we evaluate the value of $\alpha$ that works best in the effective reward. We find that $\alpha = 0.5$ worked best for Gemma 2B and 9B base policies, while a lower value of $\alpha = 0.2$ was optimal for Gemma 27B. To tune $\alpha$ we ran a grid search over the range $[0.0, 1.0]$, evaluating at an interval of 0.1. We observe that the choice of $\alpha$ is a relatively robust one, since for all three base policies, we saw improvements (over only $Q^\pi$ as the reward) for values in the range of $[0.2, 0.6]$.

**Experiment: Is the effective reward able to predict the final outcome better than $Q^\pi$?** In Fig. 11, we describe an experiment where for both the effective reward $Q^\pi + \alpha A^\mu$ and just $Q^\pi$, we compute the error of the classifier that makes a prediction on the final outcome by thresholding on either reward value at each step of the rollout. This threshold is computed using a validation set, and is separate for each step and reward combination. The figure tells us that the outcome prediction error drops for both rewards, as the base policy is rolled out more, the effective reward dominates $Q^\pi$ across all steps. Thus, the effective reward is a more informative signal (lower classification error) for the problem of predicting the success of a partial rollout, especially in the earlier steps of the rollout. This helps to explain the better performance of beam search with a finite capacity beam that re-ranks partial rollouts with the effective reward.

**Experiment: Explicitly comparing test-time search using our PAVs, with PRMs from prior works.** Several works on automated PRMs, use the Q-function as the step-level reward Wang et al. (2024); Luo et al. (2024); Snell et al. (2024). Each of these use the step-level rewards in different ways at test-time, and we compare with all of these approaches in Section 4. In Figure 4a, we compare the performance of PAVs with $Q^\pi$ for test-time beam search (see the line for "PRM $Q^\pi$"), as done in Snell et al. (2024). In the same figure, the line for "PAV-as-ORM" corresponds to using the trained PAVs as outcome reward models, *i.e.*, to score the full sequence. The sequence is scored by computing the minimum over the step-level scores from the PAV model. This is similar to how Wang et al. (2024); Luo et al. (2024) use their trained PRMs during test-time search. In Figure 12, we make these comparisons explicit. In addition, we run exactly the same procedure for test-time best-of-$N$ search with PRM $Q^\pi$ as done in Wang et al. (2024); Luo et al. (2024).

**Confidence intervals:** We compute a 95% confidence interval over the true mean of the reported test-time search performance, for each algorithm in Figure 4, Figure 5 and Figure 6. This mean is computed over the randomness in the test data, and the randomness in the search algorithm. To account for both, we run each algorithm 5 times, and then use the 500 examples in the test set, to compute the confidence interval of the true performance of each independent run of the algorithm.

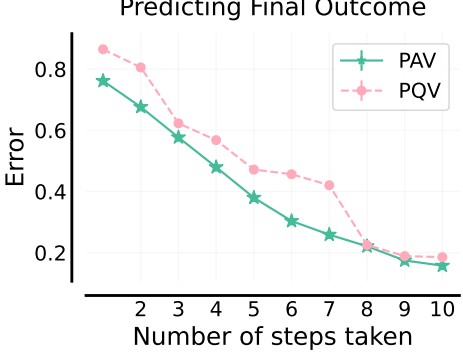

**Figure 11:** *Effective rewards at any step are able to predict the outcome rewards at the final step, better than* $Q^\pi$**:** For both the effective reward $Q^\pi + \alpha A^\mu$ and just $Q^\pi$, we compute the error of the classifier that makes a prediction on the final outcome by thresholding on either reward value at each step of the rollout. This threshold is computed using a validation set, and is separate for each step and reward combination. The figure tells us that the outcome prediction error drops for both rewards, as the base policy is rolled out more, the effective reward dominates $Q^\pi$ across all steps.

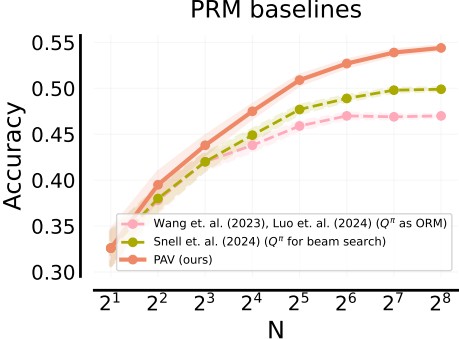

**Figure 12:** *Explicit comparison with PRM baselines in prior works:* We plot the performance of beam search with PRM $Q^\pi$ run by Snell et al. (2024), and best-of-N search with the same PRM run by Wang et al. (2024); Luo et al. (2024). We compare both with beam search run using PAVs. Note that the search mechanism for PAVs and the one in Snell et al. (2024) is the same, with the only difference being in the re-ranking function at each step of the beam search.

Then, we average the means of the confidence intervals across the 5 runs to compute the final 95% confidence interval.

**Exploration-exploitation tradeoff in Result 3 from Section 4.** For process rewards defined as our effective reward $Q^\pi + \alpha A^\mu$, as $\alpha \to 0$, the process rewards are purely exploitative, i.e. only upweight steps that are already likely to each the correct solution, under the current base policy $\pi$. On the other hand, as we increase $\alpha > 0$, then we also upweight steps that are preferred under the prover policy $\mu$. When the prover policy and the base policy are not too misaligned i.e., $E_\pi A^\mu A^\pi$ is sufficiently high, then in expectation, we expect for the the effective reward to explore new solution traces that might help the discovery of the correct solution, when sampling solutions from the base policy $\pi$. This is because, it is possible that some solution trace (set of steps) has low probability under $\pi$, but because it is preferred (high $A^\mu$ for each step in the trace) by a complementary prover policy $\mu$, the trace ends up getting a positive reward, and gets up-weighted. This aids in the discovery of the correct solution.

## D   DETAILS ON COLLECTING DATA AND TRAINING PAVS

**Collecting data.** We predict $Q^\pi$ for a policy $\pi$ by fine-tuning LLMs with cross-entropy loss on triplets $(s, a, Q^\pi_{\mathrm{mc}}(s, a))$. Data is collected by sampling $n_{\mathrm{cov}}$ seed rollouts from the base or prover policy (ORM or PAV) for coverage, followed by $n_{\mathrm{mc}}$ additional rollouts per prefix to estimate $Q^\pi$ via

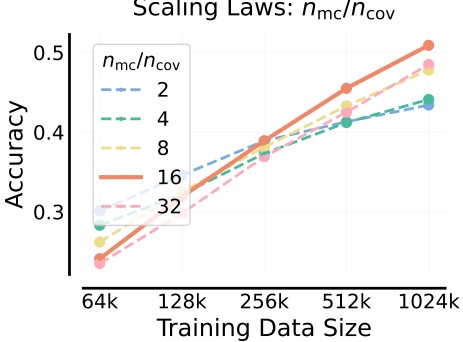

**Figure 13:** Performance of beam search over Gemma 9B SFT for PAVs trained on datasets with different $n_{mc}/n_{cov}$, where $n_{cov}$ is the number of initial seed rollouts, and $n_{mc}$ is the number of rollouts used to compute the Monte Carlo $Q$-value estimate for each state in the seed rollout.

Monte Carlo. As shown in Fig. 13, under low budgets, prioritizing coverage ($n_{cov} > n_{mc}$) improves performance, while higher budgets benefit from reducing label noise ($n_{mc} > n_{cov}$).

We now describe the procedure for training outcome verifiers and PAVs. We can learn to predict $Q^\pi$ for a policy $\pi$ (similar for $Q^\mu$) by finetuning LLMs with a cross-entropy loss on the following data with triplets $(s, a, Q_{mc}^\pi(s, a))$. To collect this data, we first sample $n_{cov}$ "seed" rollouts from the base or prover policy respectively for ORM and PAVs, to promote coverage over prefixes and steps. Then we sample $n_{mc}$ additional rollouts, conditioned on each prefix in the seed rollout to compute the Monte-Carlo estimate of $Q^\pi$ at each prefix. In Fig. 13 we plot the beam search performance of PAVs trained with different ratios of $n_{mc}/n_{cov}$, as we scale the total dataset size. Here, the beam size is fixed to 128 and the base policy is the Gemma 9B SFT policy and prover is *Bo*4 policy. We find that under low sampling budgets, optimizing for coverage ($n_{cov} > n_{mc}$) is better for performance, and when budget is higher, reducing label noise in $Q_{mc}^\pi$ by setting $n_{mc} > n_{cov}$ gets us more improvements. In addition, we also spend some initial sampling budget is spent to identify "high value" states where $Q^\pi$ is larger than a threshold, and identify the first step with low $Q^\pi$ on an incorrect partial rollout from this state. We found this strategy to scale better with dataset size.

In Fig. 13, our seed rollouts are *i.i.d.* sampled from $\pi$, but prior works (Luo et al., 2024; Setlur et al., 2024; Hwang et al., 2024) employed a "first pit" strategy for coverage. Here, some initial sampling budget is spent to identify "high value" states where $Q^\pi$ is larger than a threshold. Then, for any incorrect sample from these high value states greater budget is spent to estimate the first step (first pit) with $Q^\pi = 0$. All prefixes (and their estimated $Q$-values) until the first pit are then added to the training data. In Fig. 14, we compare beam search using PAVs trained using data from the first pit strategy, and the random sampling strategy. Both of them use the best value of $n_{mc}/n_{cov}$ from Fig. 13 for every dataset size. We find the first pit strategy to be better than random, especially when the number of seed rollouts are limited. Once we get coverage over such pits, we sample a large number of partial rollouts conditioned on each prefix until the first pit. This is used to compute the Monte Carlo estimate of $Q$ values more accurately on the path to the first pit. Each prefix and estimated $Q$ value pair is then added to the dataset used to train PAVs.

**Training details.** All PAVs used in this work are trained by taking the Gemma 9B pretrained checkpoint and finetuning it on the data collected from the above strategy. The data collection uses first pit strategy for better coverage over pits in the seed rollouts. Based on findings in Fig. 13, we use a high value of $n_= 20$ to estimate the $Q$-values accurately for each step in the seed rollout. For each base policy, in total, we collect a dataset of over $300,000$ prefix, $\hat{Q}^\pi$-value pairs (where $\hat{Q}^\pi$ is the Monte Carlo estimate) on which we finetune the Gemma 9B model with cross-entropy loss. Since the distribution of values for $\hat{Q}^\pi$ can be skewed, we split the range of $\hat{Q}^\pi$-values into two buckets, based on which we also partition the training data. The first bucket is the set of all prefixes with $\hat{Q}^\pi < 0.5$ and the second is the set of all prefixes with $\hat{Q}^\pi \geq 0.5$. Then, we use class-balanced sampling over these buckets to train the finetune the pretrained model for 20000 training iterations, using a batch size of 32. We use an Adam optimizer with a maximum learning rate of $5e - 7$. We use a linear warm up (till 2000 steps), and a cosine decay learning rate schedule to train the models. Since a pretrained LLM would output a matrix of logits (vocabulary size × sequence length) we fix a token

as the "scoring token" to be the end of the sequence / prefix that needs to be scored. The logits of this scoring token are then used to determine the prediction for the LLM being trained.

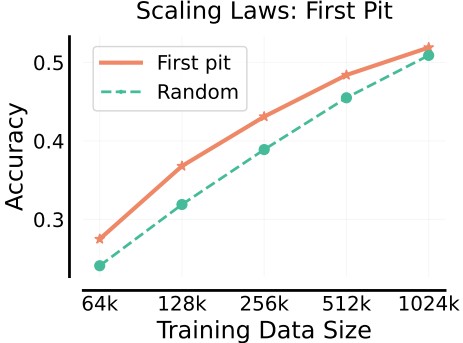

**Figure 14:** *First pit strategy from Luo et al. (2024); Setlur et al. (2024)*: We compare the beam search performance (with beam size 128) for a PAV trained on data collected using two types of seed rollouts. For the seed rollouts, we either randomly sample from the distribution of state action pairs induced by the base policy. Or we improve coverage particularly using the "first pit" strategy of identifying the first state with low $Q$ values on a partial rollout that starts from a high $Q$-value state and ends with an outcome reward of 0, i.e., $Q$ value is 0.

**Using trained PAVs:** For test-time search over responses from the base policy $\pi$, and fixing a prover policy $\mu$, the following is how we use PAVs for beam search at test-time. At any given time step, the beam holds partially generated responses (states) to a math problem. We score each of the states using our effective reward $Q^\pi + \alpha A^\mu$. Since $A^\mu$ is computed using $Q^\mu$ (equation 2), we need only train verifiers to predict $Q^\pi$ and $Q^\mu$. Finally, at the end of beam search, when only we have complete responses in the beam, we pick the best response using an ORM trained only to predict the correctness of final answers. In practice, we find that using a trained ORM to rate complete responses at the end of beam search is not critical, and using the PRM that predicts $Q^\pi$ to rank the full responses also performs similarly. So, the training of an ORM model can be avoided.

## E ADDITIONAL DETAILS: EXPERIMENTS ON RL TRAINING WITH PAVS

**Experiment: RL training of Gemma 9B SFT policy with PAV-RL.** Similar to Figure 7)(c), where we compared the sample efficiency and raw performance of PAV-RL with ORM-RL for the Gemma 2B SFT base policy, in Figure 15, we compare the same for the Gemma 9B SFT base policy. Here, the prover policy for the effective reward in PAV-RL is the Gemma 2B SFT policy, which is worse than the Gemma 9B SFT policy, let alone the policy iterates during RL training. We find PAV-RL to boost sample efficiency by 5×, while being 6% better in raw accuracy.

**Experiment: Discounting strong provers.** In Section 5 we find that the better prover policy to train the Gemma 2B base policy is the Gemma 2B prover policy, and the stronger Gemma 9B RL policy. We attribute this finding to our discussion in Section 3.2, where we explain how very strong provers end up with advantages $A^\mu$ which are close to 0, since they maybe able to complete the solution from most states generated by the base policy. But what happens if we account for the steps taken by the stronger prover policy to complete the solution? We test this with the following experiment

We train PAVs to predict the advantages of discounted rewards from strong prover policies. Here, for the problem $x$, and the state, action pair $s, a$, the process rewards are given by the effective reward from equation 5: $Q^\pi + \alpha A^\mu$, except that the advantage $A^\mu$ is the difference in discounted rewards, i.e.:

$$A^\mu(s, a) = \mathbb{E}_{y \sim \mu(\cdot|s,a)} \left[ \lambda^{\text{len}(y)-\text{len}(s)-1} \text{Rex}(y, y_x^\star) \right] - \mathbb{E}_{y \sim \mu(\cdot|s)} \left[ \lambda^{\text{len}(y)-\text{len}(s)} \text{Rex}(y, y_x^\star) \right],$$

where the prover policy samples solution $y$ with $\text{len}(y)$ steps to complete the solution, from a state $s$, which already has $\text{len}(s)$ steps in it.

For this setting, we train a verifier to predict discounted rewards for the Gemma 9B prover policy. We find that the discounted process rewards from the stronger 9B prover policy performs worse than

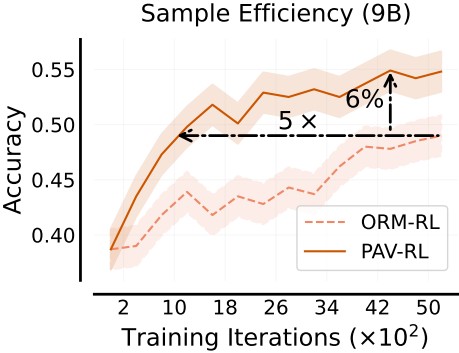

**Figure 15:** *Sample efficiency and performance gains for PAV-RL on Gemma 9B SFT policy:* We plot the test accuracy across PAV-RL and ORM-RL training iterations, where the base policy is the Gemma 9B SFT policy. The effective rewards in PAV-RL boost the sample efficiency by 5× over outcome rewards in ORM-RL. Additionally, PAV-RL trained policy is 6% more accurate over than ORM-RL trained policy, when compared over the course of 5000 training iterations.

undiscounted rewards from the weaker Gemma 2B prover policy, when using either to train the 2B base policy with online RL (Figure 16).

The main reason for the discounted rewards to not enable the use of strong provers is because strong prover policies tend to disregard states generated by the base policy (as illustrated in Figure 2b). This means, that irrespective of whether the weak prover policy generates a partially correct or incorrect solution, when we rollout the strong prover policy from this state generated by the base policy, the strong prover directly attempts to answer the math problem with its own solution trace. Thus, from any state the strong prover is expected to complete the solution with roughly the same number of steps. This means that $A^\mu \approx 0$ even in the discounted case, which reduces the ability of the strong prover policy to distinguish steps taken by the base policy.

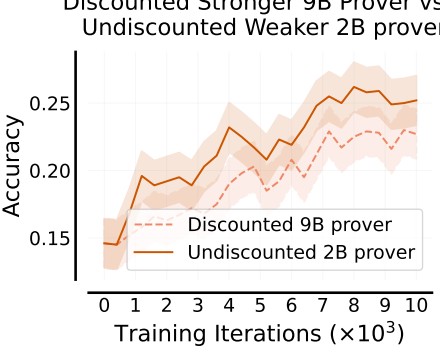

**Figure 16:** *Comparing weaker, undiscounted 2B prover policy with discounted, stronger 9B prover:* We plot the test accuracy across PAV-RL training iterations, where the base policy $\pi$ is the Gemma 2B SFT policy, and two different options of the prover policy $\mu$ used in the computation of the effective rewards in equation 5. The discounted stronger 9B prover policy still under performs the weaker Gemma 2B prover policy.

**Experiment: Using PRMs from prior works to assign step-level rewards during online RL.** We also experiment with using the PRMs proposed in prior works Snell et al. (2024); Wang et al. (2024); Luo et al. (2024) as step-lever rewards in online RL. For this, we use $Q_{\text{base}}^\pi$ as the step-level score during RL training initialized with the base policy $\pi_{\text{base}}$. This step-level reward is used to reward trajectories sampled during online RL training, in addition to the outcome reward. We plot our results in Figure 17 and find that the test accuracy drops quite quickly, even though the train rewards keep improving, since the policy just learns to hack the step-level Q-values, for the reasons we explain in Section 3.2. We see a similar collapse when we use $Q^\mu$ instead of $A^\mu$ (see Appendix G for qualitative examples).

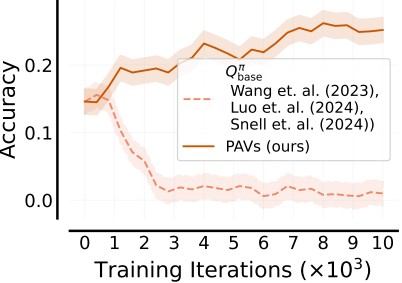

**Figure 17:** *Using PRMs from prior works to assign step-level rewards during online RL:* . We take the PRM that predicts $Q$-values of the base policy, *i.e.*, $Q_{\text{base}}^{\pi}$, and use it to assign step-level rewards during online RL training, in addition to the outcome reward. Note that, Snell et al. (2024); Wang et al. (2024); Luo et al. (2024) used the same automated PRM $Q_{\text{base}}^{\pi}$ to run test-time beam search or best-of-$N$ search over $\pi_{\text{base}}$. We find that using the same for online RL leads to the step-level rewards being hacked for reasons we explain in Section 3.2.

**Training details.** As discussed in Section 5, the initialization for RL training is the RFT (rejection finetuned) checkpoint for the corresponding base policies. More specifically, we consider two base policies Gemma 2B SFT, and Gemma 9B SFT, where the RL training is initialized with the policy obtained by further optimizing the base policy via rejection finetuning (RFT) Yuan et al. (2023). This is done to improve coverage over states and actions during the initial iterations of RL. For rejection finetuning we train the SFT model (base policy) on all correct trajectories sampled when collecting seed rollouts for training PAVs (that predict the advantage of the same base policy). The training details for RFT remain same as SFT and are detailed in Appendix C. The RL training is run for 10000 iterations for the 2B model, and for 5000 iterations for the 9B model. For both we use the Adam optimizer with a learning rate of 1e-7, and a batchsize of 32. The maximum response length is set to 512. For both we used a learning rate schedule with a linear warm up for 10% of the total training iterations. The implementation of our policy gradient algorithm also uses a value function that is initialized to be the base policy itself. The value function is only used as a baseline during RL training (REINFORCE).

Most importantly, we use a validation set to identify a good choice of $\alpha$ in the effective reward (in equation 5). For Gemma 2B this value is 5.0 and for the 9B policy $\alpha = 3.0$. Similar to the choice of $\alpha$ for test-time search, we find that most values of $\alpha$ in the range of 0.5 to 6.0 improved performance over ORM-RL, to different degrees. Both policies are also optimized with KL regularization, against the initial iterate of the RL policy, where the strength of the KL penalty is set to 0.001 for both.

**Confidence intervals:** We compute a 95% confidence interval over the true mean of the test accuracy, at each iterate of the RL training in Figure 7, Figure 15, and for each value of $N$ in Figure 8. This mean is computed over the 500 examples in the MATH500 test set. For improved confidence intervals over our results with PAV-RL training in Figure 7(a), we run RL training with three random seeds and find the performance across these runs to be 25.85, 25.77, 25.73 for the Gemma 2B SFT base policy and 55.72, 55.89, 55.61 for the Gemma 9B SFT base policy.

**Tuning $\alpha$ in practice.** In practice, we arrive at the ranges for searching over the hyper-parameter $\alpha$ through a systematic procedure on a hold-out validation set. This procedure can be potentially repeated for any new problem instance where PAVs need to be used. In particular, we tune $\alpha$ with a two layer search strategy where the outer layer search is coarse-grained and used to identify a good high-level range of $\alpha$ values such that the performance of PAVs is comparable to ORMs (i.e., PAVs don't yield degenerate solutions). The inner layer of search is more fine-grained and used to tune the performance of PAVs even more. Note that both these layers of search are carried out on a small hold out validation set. Additionally, as we state below, the outer level of coarse-grained search is already good enough for PAVs to outperform ORMs on the MATH benchmark, for both beam search and online RL. For both test-time search and RL we identified a ''good'' range of $\alpha$ by running binary search. Since $\alpha > 0$, we start with a high value of $\alpha = 10.$, and then keep reducing it by half $(10 \rightarrow 5 \rightarrow 2 \rightarrow 1)$ until we see a run of test-time beam search or online RL that yields a non-trivial performance improvement over only using outcome rewards. During this outer level binary search,

we stopped at $\alpha = 5.0$ for online RL and $\alpha = 1.0$ for test-time search. Once we identified the range, we run a second level of fine-grained search search, i.e., we search linearly between $\alpha \in [0.0, 1.0]$ in intervals of 0.1 for using PAVs at test-time, and $\alpha \in [0.5, 6.0]$ in intervals of 0.5 for using PAVs as process rewards in online RL. As we state in the paper, the choice of $\alpha$ within this range is quite robust. Any $\alpha \in (0.1, 0.7)$ for test-time search and $\alpha \in (1.5, 5.5)$ for online RL puts PAVs in a regime where the performance of PAVs for either is better than only using outcome rewards.

## F    THEORETICAL ANALYSIS: COMPLEMENTARY PROVERS AMPLIFY BASE POLICY IMPROVEMENT

In this section, we present the proofs for our theoretical results in the main paper (Sec. 3.4). We begin by describing some notation we use in our proofs, the natural policy gradient algorithm we use for the policy update, followed by the proofs for Theorem 3.1, and Proposition F.1. Our results in this section are in the tabular setting, with softmax parameterization of the policies. Note, that for the deterministic Markov decision process induced by the LLM, we are indeed in a tablular setting, where the set of states and actions is discrete, but large and grows exponentially in sequence length.

**Notation and preliminaries.** We use $d_h^\pi, d_h^\mu$ to denote the distribution over states $s_h$ at time step $h$, starting from the initial state distribution given by the empirical distribution over the questions in the dataset $\mathcal{D}$, and following the base policy $\pi$, or prover policy $\mu$ respectively. Mathematically,

$$d_h^\pi(s_h = (x, a_1, \ldots, a_{h-1})) = \sum_{x \in \mathcal{D}} \prod_{h'=1}^{h-1} \pi(a_{h'} \mid s_{h'-1} = (x, a_1, \ldots, a_{h'-1}))$$

The term $d_s^\pi$ denotes the distribution over future states, starting from state $s$, and following policy $\pi$. Here, $s$ can be a state at any time $h \in [0, \ldots, H]$. For convenience, we overload the notation $d_s^\pi$ (the distribution over future states induced by a policy starting from state $s$), and use $d_\rho^\pi$ to denote the distribution over states $d_s^\pi$ starting from a random state drawn from $\rho$, and following policy $\pi$.

The term $Q^\pi(s_h, a_h)$ refers to the value of state-action pair $s_h, a_h$, i.e., the expected return in the future, starting the policy from state $s_h$ and taking action $a_h$:

$$Q^\pi(s_h, a_h) \coloneqq \mathbb{E}_{a_{h+1}, \ldots, a_H \sim \pi(\cdot | s_h, a_h)} \left[ \mathrm{Rex}\left((a_1, \ldots, a_H), y_x^\star\right) \right]. \tag{7}$$

Note that $y_x^\star$ is known on the dataset $\mathcal{D}$, and state $s_h$ contains the question $x$ as part of it. Similarly, we can define the value function $V^\pi(s_h)$ of a state $s_h$ as:

$$V^\pi(s_h) \coloneqq \mathbb{E}_{a_{h-1} \sim \pi(\cdot | s_h)} Q^\pi(s_{h-1}, a_{h-1}). \tag{8}$$

The advantage function is then given by:

$$A^\pi(s_h, a_h) \coloneqq Q^\pi(s_h, a_h) - V^\pi(s_h). \tag{9}$$

The policy gradient algorithm we use to update the base policy iteratively updates the base policy, and we use $\pi_t$ to refer to the base policy iterate at time $t$. Finally, we use $\mathcal{S}$ to denote the set of all states and $\mathcal{A}$ for the set of all actions that the LLM can take at any state.

**Parameterization of the base policy.** We adopt the softmax parameterization for the base policy:

$$\pi_\theta(a \mid s_h) = \frac{\exp(\theta_{s_h, a})}{\sum_{a' \in \mathcal{A}} \exp(\theta_{s_h, a'})}. \tag{10}$$

Here $\theta_{s_h, a} \in \Theta$ is a class of parameters, that controls the probability of taking action $a$ at state $s_h$. The full set of parameters across all states and actions is denoted by $\theta \in \mathbb{R}^{d \times |\mathcal{S}| \times |\mathcal{A}|}$. Whenever clear from context, we overload the notation $\pi_t$ to denote both the policy at iterate $t$, i.e., $\pi_{\theta_t}$ and the parameter $\theta_t$ itself. Similarly, the operator $\nabla_{\pi_t}[\cdot]$, is referring to $\nabla_{\theta_t}[\cdot]$.

**Defining policy improvement.** Let $\rho$ be a distribution over all states $\{s_h : h \in [0, 1, \ldots, H]\}$, then $\mathbb{E}_{s \sim \rho} V^\pi(s)$, and $\mathbb{E}_{s \sim \rho} V^\mu(s)$ give us the expected value functions over states across time steps, measured under $\rho$, for policies $\pi$ and $\mu$ respectively. We assume that $d_h^\pi$ and $d_h^\mu$ are both absolutely continuous with respect to $\rho$, and use the expected value function over $\rho$ as the quantity we track before and after a policy update. A positive change in $\mathbb{E}_{s \sim \rho} V^\pi(s)$ implies a net positive improvement in the base policy. Thus, progress is made at each update of the policy when:

$$\mathbb{E}_{s \sim \rho} V^{\pi_{t+1}}(s) \; - \; \mathbb{E}_{s \sim \rho} V^{\pi_t}(s) > 0.$$

### F.1 Natural Policy Gradient

The natural policy gradient (NPG) algorithm (Kakade, 2001) defines a Fisher information matrix (induced by the policy), and performs gradient updates in the geometry induced by the following matrix:

$$F_\rho(\pi) = \mathbb{E}_{s \sim d_\rho^\pi} \mathbb{E}_{a \sim \pi(\cdot|s)} \left[ \nabla_\pi \log \pi(a \mid s) \left( \nabla_\pi \log \pi(a \mid s) \right)^\top \right] \tag{11}$$

Typically, the NPG update does gradient updates on the objective $\ell_{ORM-RL}$

$$\pi_{t+1} = \pi_t + \gamma \cdot F_\rho(\pi^t)^\dagger \left( \nabla_\pi \ell_{\text{PAV-RL}}(\pi) \Big|_{\pi=\pi_t} \right), \tag{12}$$

where $M^\dagger$ denotes the Moore-Penrose pseudoinverse of the matrix $M$. We restrict to using the initial state distribution $\rho$ in our update rule, *i.e.*, we restrict attention to states $s$ reachable from $\rho$, since $\rho$ governs the performance measure of interest when evaluating the expected value of a policy. Thus, without loss of generality, we can exclude states that are not reachable under $\rho$. Specifically, we restrict the MDP to the set of states $\{s_h \ : \ \exists \pi \text{ such that } d_\rho^\pi(s_h) > 0, h \in [0, \dots, H]\}$.

### F.2 Useful Lemmas

**Lemma F.1.** *[The performance difference lemma; (Kakade & Langford, 2002)] For all policies $\pi, \pi'$ and states $s_0$,*

$$V^\pi(s) - V^{\pi'}(s) \quad = \quad \mathbb{E}_{s_h \sim d_s^\pi} \mathbb{E}_{a_h \sim \pi(\cdot|s_h)} \left[ A^{\pi'}(s_h, a_h) \right].$$

*Proof.* See proof of Lemma 6.1 in Kakade & Langford (2002). □

**Lemma F.2** (Natural policy gradient update). *For the natural policy gradient in equation 12, the corresponding policy update is given by:*

$$\pi^{t+1}(a \mid s) = \pi^t(a \mid s) \cdot \frac{\exp\left(\gamma \cdot \left(A^t(s, a) + \alpha \cdot A^\mu(s, a) - \alpha \mathbb{E}_\pi A^\mu(s, a)\right)\right)}{Z^t(s)}, \tag{13}$$

$$Z^t(s) = \gamma \cdot \sum_{a \in \mathcal{A}} \left( A^t(s, a) + \alpha \cdot A^\mu(s, a) \right) \tag{14}$$

*Proof.* We use arguments similar to the proof of Lemma 15 in Agarwal et al. (2021), with the key difference of separately accounting for the term $A^\mu(s, a)$. For the sake of completeness we reproduce some of the derivation, accounting for the $A^\mu$ term in the process. We follow compatible function approximation in Sutton et al. (1999) and Kakade (2001). For a vector $w \in \mathbb{R}^{d \times |\mathcal{S}||\mathcal{A}|}$, we define the error function

$$L^\pi(w) = \mathbb{E}_{s \sim d_\rho^\pi}, \mathbb{E}_{a \sim \pi(\cdot|s)} \left( w^\top \nabla_\pi \log \pi(\cdot \mid s) - (A^\pi(s, a) + \alpha A^\mu(s, a)) \right)^2. \tag{15}$$

Let $w^\star$ be the minimizer of $L^\pi(w)$ with the smallest $\ell_2$ norm. Then by definition of Moore-Penrose pseudoinverse:

$$w^\star = F_\rho(\pi)^\dagger \mathbb{E}_{s \sim d_\rho^\pi, a \sim \pi(a|s)} [\nabla_\pi \log \pi(a|s) (A^\pi(s, a) + \alpha A^\mu(s, a))]$$
$$= F_\rho(\pi)^\dagger \nabla_\pi \ell_{\text{PAV-RL}}(\pi). \tag{16}$$

In other words, $w^\star$ is precisely proportional to the NPG update direction. Note further that for the Softmax policy parameterization, we have:

$$w^\top \nabla \log \pi(a|s) = w_{s,a} - \sum_{a' \in \mathcal{A}} w_{s,a'} \pi(a'|s).$$

Since $\sum_{a \in \mathcal{A}} \pi(a|s) A^\pi(s, a) = 0$, this immediately yields that $L^\pi(A^\pi + \alpha A^\mu) = 0$. However, this might not be the unique minimizer of $L^\pi$, which is problematic since $w^\star(\pi)$ as defined in terms of the Moore-Penrose pseudoinverse is formally the smallest norm solution to the least-squares problem, which $A^\pi + \alpha A^\mu$ may not be. However, given any vector $v \in \mathbb{R}^{|\mathcal{S}||\mathcal{A}|}$, let us consider solutions of the

form $A^\pi + \alpha A^\mu + v$. Due to the form of the derivatives of the policy for the softmax parameterization, we have for any state $s, a$ such that $s$ is reachable under $\rho$,

$$v^\top \nabla_\pi \log \pi(a|s) = \sum_{a' \in \mathcal{A}} (v_{s,a'} \mathbf{1}[a = a'] - v_{s,a'} \pi(a'|s)) = v_{s,a} - \sum_{a' \in \mathcal{A}} v_{s,a'} \pi(a'|s).$$

This is because $\pi$ is a stochastic policy with $\pi(a|s) > 0$ for all actions $a$ in each state $s$, so that if a state is reachable under $\rho$, it will also be reachable using $\pi$, and hence the zero derivative conditions apply at each reachable state. For $A^\pi + \alpha A^\mu - \mathbb{E}_\pi A^\mu + v$ to minimize $L^\pi$, we would like $v^\top \nabla_\pi \log \pi(a|s) = 0$ for all $s, a$ so that $v_{s,a}$ is independent of the action and can be written as a constant $c_s$ for each $s$ by the above equality. Hence, the minimizer of $L^\pi(w)$ is determined up to a state-dependent offset, and

$$F_\rho(\theta)^\dagger \nabla_\pi \ell_{PAV-RL} \pi = A^\pi + \alpha A^\mu + v,$$

where $v_{s,a} = c_s$ for some $c_s \in \mathbb{R}$ for each state $s$ and action $a$. Finally, we observe that this yields the updates

$$\pi_{t+1} = \pi^t + \gamma(A^\pi + \alpha A^\mu + v) \quad \text{and} \quad \pi_{t+1}(a|s) = \pi_t(a|s) \frac{\exp(\gamma A^t(s, a) + \gamma \alpha A^\mu(s, a))}{Z^t(s)}.$$

Owing to the normalization factor $Z^t(s)$, the state dependent offsets cancel in the updates for $\pi$, which yields the statement of the lemma. □

### F.3 PROOF OF THEOREM 3.1

*Proof.* For some notational convenience, we use $V^t, V^{t+1}$, to denote the value functions $V^{\pi_t}, V^{\pi_{t+1}}$ for policies at $\pi_t, \pi_{t+1}$ at time $t$ and $t+1$ respectively. Similarly, we use $A^t, A^{t+1}$ for $A^{\pi_t}, A^{\pi_{t+1}}$ respectively. For the distribution over states $d_\rho^{\pi_{t+1}}$ induced by the policy $\pi_{t+1}$, starting from an initial distribution of states given by $\rho$, we simplify the notation and use $d_\rho^{t+1}$. Similarly we use $d_\rho^t$ for $d_\rho^{\pi_t}$.

Next, for simplicity we set $\alpha = 1$ in the natural policy gradient update in Lemma F.2. It is easy to see that the lower bound result we show holds for any value of $\alpha > 0$, and the term in the lower bound scales linearly with $\alpha$. Note, that this is not a free variable, and $\alpha$ has to be $O(1)$, since as we increase the value of $\alpha$ we would have to correspondingly reduce $\gamma$ for our result to hold. For now, we fix $\alpha = 1$.

From the policy difference Lemma F.1, we can write:

$$\mathbb{E}_{s \sim \rho} V^{t+1}(s) - \mathbb{E}_{s \sim \rho} V^t(s) = \mathbb{E}_{s \sim d_\rho^{t+1}} \mathbb{E}_{a \sim \pi^{t+1}(a|s)} \left[ A^t(s, a) \right] \tag{17}$$

Next, from the natural policy gradient update in Lemma F.2, we can write $A^t(s, a)$ as:

$$A^t(s, a) = \frac{1}{\gamma} \cdot \log \left( \frac{\pi^{t+1}(a \mid s) \cdot Z^t(s, a)}{\pi^t(a \mid s)} \right) - A^\mu(s, a) \tag{18}$$

Substituting equation 18 in equation 17 we get:

$$\mathbb{E}_{s \sim \rho} V^{t+1}(s) - \mathbb{E}_{s \sim \rho} V^t(s) = \frac{1}{\gamma} \mathbb{E}_{s \sim d_\rho^{t+1}} \left[ \text{KL}(\pi^{t+1}(\cdot \mid s) \| \pi^t(\cdot \mid s)) \right]$$

$$+ \frac{1}{\gamma} \log Z^t(s) - \mathbb{E}_{s \sim d_\rho^{t+1}} \mathbb{E}_{a \sim \pi^{t+1}(a|s)} A^\mu(s, a). \tag{19}$$

Recall that for $\alpha = 1$,

$$\log Z^t(s) = \log \mathbb{E}_{a \sim \pi^t(\cdot|s)} \exp \left( \gamma \cdot (A^t(s, a) + A^\mu(s, a)) \right) \tag{20}$$

Applying Jensen's inequality we get:

$$\log Z^t(s) \geq \gamma \cdot \mathbb{E}_{a \sim \pi^t(\cdot|s)} \left[ A^t(s, a) + A^\mu(s, a) \right] \tag{21}$$

$$= \gamma \cdot \mathbb{E}_{a \sim \pi^t(\cdot|s)} \left[ A^\mu(s, a) \right], \tag{22}$$

since $\mathbb{E}_{\pi^t}\left[A^t(s,a)\right] = 0$.

Note that in equation 19 the KL term is always non-negative. Thus, we can lower bound our policy improvement as:

$$\mathbb{E}_{s\sim\rho}V^{t+1}(s) - \mathbb{E}_{s\sim\rho}V^t(s) \geq \mathbb{E}_{s\sim d_\rho^{t+1}}\langle\pi^{t+1}(\cdot\mid s) - \pi^t(\cdot\mid s), A^\mu(s,\cdot)\rangle, \tag{23}$$

where the inner product is the standard euclidean product as our actions space $\mathcal{A}$ is discrete.

In the following we will treat the distribution $\pi^{t+1}(\cdot\mid s)$ as a vector denoted by $\pi$ Next, from the NPG update we know that:

$$\pi^{t+1}(a\mid s) - \pi^t(a\mid s) = \pi^t(a\mid s)\left(\frac{\exp\left(\gamma A^t(s,a) + \gamma A^\mu(s,a)\right)}{Z^t(s)} - 1\right) \tag{24}$$

We note that for $\gamma \ll 1$, $\exp\gamma(A^t(s,a)) + \gamma(A^\mu(s,a)) = \Theta(1 + \gamma(A^t(s,a) + A^\mu(s,a)))$, where the terms that grow as $\omega(\gamma)$ are being ignored. Based on this, for $\gamma \ll 1$, $\exists$ constants $0 < C_1 < C_2$ such that:

$$\exp(\gamma A^t(s,a) + \gamma A^\mu(s,a)) - 1 \in [C_1\gamma(A^t(s,a) + A^\mu(s,a)), C_2\gamma(A^t(s,a) + A^\mu(s,a))]$$

$$\pi^{t+1}(a\mid s) - \pi^t(a\mid s) \geq \pi^t(a\mid s)\left(\frac{1 + C_1\gamma(A^t(s,a) + A^\mu(s,a))}{1 + C_2\gamma\mathbb{E}_{a\mid\pi^t(\cdot\mid s)}\left[A^t(s,a) + A^\mu(s,a)\right]} - 1\right)$$

$$\geq C_3\gamma\frac{\left(\pi^t(a\mid s)\left(A^t(s,a) + A^\mu(s,a)\right) - \pi_t(a\mid s)\mathbb{E}_{a\sim\pi_t(a\mid s)}\left[A^t(s,a) + A^\mu(s,a)\right]\right)}{1 + C_2\gamma\mathbb{E}_{a\sim\pi_t(a\mid s)}\left[A^t(s,a) + A^\mu(s,a)\right]} \tag{25}$$

$$= C_3\gamma\frac{\left(\pi^t(a\mid s)\left(A^t(s,a) + A^\mu(s,a)\right) - \pi_t(a\mid s)\mathbb{E}_{a\sim\pi_t(a\mid s)}\left[A^\mu(s,a)\right]\right)}{1 + \gamma C_2\mathbb{E}_{a\sim\pi_t(a\mid s)}\left[A^t(s,a) + A^\mu(s,a)\right]}, \tag{26}$$

where we reused: $\mathbb{E}_{\pi^t}\left[A^t(s,a)\right] = 0$. Here, $C_3 > 0$ is a constant.

We now plug in the above lower bound into equation 23 to get the final lower bound on the policy improvement in Theorem 3.1. For this, we will once again use the assumption that the learning rate $\gamma \ll 1$, which allows us to use $1 + \gamma\mathbb{E}_{a\sim\pi_t(a\mid s)}\left[A^t(s,a) + A^\mu(s,a)\right] \geq C_4$ for some constant $C_4 > 0$. This is because, in our setting the range of the advantages is $[-1,1]$. Since, advantages are bounded in $[-1,1]$, we know that $1 + \gamma C_2\mathbb{E}_{a\sim\pi_t(a\mid s)}\left[A^t(s,a) + A^\mu(s,a)\right] \leq 1 - \gamma$, where $\gamma \ll 1$. Thus,

$$\mathbb{E}_{s\sim\rho}\left[V^{t+1}(s) - V^t(s)\right] \geq \frac{C_3}{1-\gamma}\left(\gamma\mathbb{E}_{s\sim d_\rho^{t+1}}\left[\mathbb{E}_{a\sim\pi^t(a\mid s)}\left[A^\mu(s,a)A^t(s,a)\right]\right]\right.$$

$$+\gamma\mathbb{E}_{s\sim d_\rho^{t+1}}\left[\mathbb{E}_{a\sim\pi^t(a\mid s)}\left[(A^\mu(s,a))^2\right]\right]$$

$$\left.-\gamma\mathbb{E}_{s\sim d_\rho^{t+1}}\left[\left(\mathbb{E}_{a\sim\pi^t(a\mid s)}\left[A^\mu(s,a)\right]\right)^2\right]\right)$$

Setting $C$ from Theorem 3.1 as $C_3/1-\gamma$,

$$\mathbb{E}_{s\sim\rho}\left[V^{t+1}(s) - V^t(s)\right] \geq \gamma\mathbb{E}_{s\sim d_\rho^{t+1}}\left[\mathbb{E}_{a\sim\pi^t(a\mid s)}\left[A^\mu(s,a)A^t(s,a)\right]\right]$$

$$+C\gamma\mathbb{E}_{s\sim d_\rho^{t+1}}\left[\mathbb{E}_{a\sim\pi^t(a\mid s)}\left[(A^\mu(s,a))^2\right]\right]$$

$$-C\gamma\mathbb{E}_{s\sim d_\rho^{t+1}}\left[\left(\mathbb{E}_{a\sim\pi^t(a\mid s)}\left[A^\mu(s,a)\right]\right)^2\right] \tag{27}$$

This gives us,

$$\mathbb{E}_{s\sim\rho}\left[V^{t+1}(s) - V^t(s)\right] \geq C\gamma\mathbb{E}_{s\sim d_\rho^{t+1}}\left[\mathbb{V}_{a\sim\pi_t(a\mid s)}\left[A^\mu(s,a)\right] - \mathbb{E}_{a\sim\pi_t(a\mid s)}\left[A^\mu(s,a)A^t(s,a)\right]\right] \tag{28}$$

Now, for the last step we note that $d_\rho^{t+1}$ is component wise larger than $\rho$, and this gives us the final result:

$$\mathbb{E}_{s \sim \rho}\left[V^{t+1}(s) - V^t(s)\right] \geq C\gamma \mathbb{E}_{s \sim \rho}\left[\mathbb{V}_{a \sim \pi_t(a|s)}\left[A^\mu(s, a)\right] - \mathbb{E}_{a \sim \pi_t(a|s)}\left[A^\mu(s, a)A^t(s, a)\right]\right] \quad (29)$$

$\square$

### F.4 Discussion on Remark 3.1

First, we note that if the $Q$ value of a base policy $\pi$ at state, action pair $(s, a)$ is $Q^\pi(s, a)$, then for the prover $\mu$ set to the best-of-K policy $\text{BoK}(\pi)$, the $Q$ value at the same state, action pair is:

$$Q^\mu(s, a) = 1 - (1 - Q^\pi(s, a))^K \quad (30)$$

This is because, in our setup the final outcome of correctness given by Rex is a random variable taking values in $\{0, 1\}$, with expectation $Q^\pi(s, a)$, when completing a rollout starting from prefix $(s, a)$. Thus, we can treat the outcome as a Bernoulli. Now, we can simply compute the probability of sampling a single correct answer out of $K$ attempts, which is also a Bernoulli random variable with mean given by $Q^\mu(s, a)$ in equation 30.

Next, we observe that whenever $Q^\pi(s, a) \ll 1$, e.g., when $Q^\pi(s, a) = O(1/K)$, for all values of $(s, a)$, we can do the taylor approximation of $Q^\mu(s, a)$ around 0, and note that $Q^\mu(s, a) = \Theta(K \cdot Q^\pi(s, a))$. Next note that:

$$\begin{aligned}
\mathbb{V}_\pi A^\mu(s, a) &= \mathbb{V}_\pi Q^\mu(s, a) \\
&= \mathbb{V}_\pi A^\mu(s, a) \\
&= \mathbb{V}_\pi\left[1 - (1 - Q^\pi(s, a))^K\right] \\
&= \mathbb{V}_\pi\left[(1 - Q^\pi(s, a))^K\right]
\end{aligned}$$

This means that the first term in Theorem 3.1 which measures "distinguishability" now increases by a factor of $K^2$. Similarly, we can see that the the term which measures "misalignment" can change in magnitude by a factor of $O(K)$, since the misalignment term is linear in $Q^\mu$. These two observations combined lead us to the final remark in Remark 3.1,

### F.5 Improving a Stronger Base policy with a Weaker Prover Policy

In Proposition F.1, we consider the case where the $\pi$ and $\mu$ differ in performance, as measured under the distribution of states $\rho$ in the following way:

$$\mathbb{E}_{s \sim \rho}[|V^\mu(s) - V^{\pi_t}(s)|] = \eta.$$

Next, whenever the prover's preference over actions is complementary to the base policy, by a factor of $\eta$, i.e.,

$$\mathbb{E}_{s \sim \rho}\mathbb{E}_{a \sim \pi}[|Q^\mu(s, a) - Q^{\pi_t}(s, a)|] = \Omega(\eta),$$

then the variance of $A^\mu$ or $A^{\pi_t}$ under $\pi_t$ should scale as $\eta^2$.

Thus, we see that when $\pi_t$ fails to distinguish actions (*i.e.*, $\mathbb{E}_{s \sim \rho}\mathbb{V}_{\pi_t}[A^{\pi_t}(s, a)]$ is small) regardless of the strength of prover policy $\mu$, as long as it is sufficiently complementary to $\pi_t$, the prover policy induces an improvement in base policy, that scales as $\eta^2$.

**Proposition F.1** (Complementary $\mu$ boosts improvements in $\pi$). *Under the distribution over states given by $\rho$, let prover $\mu$ and base policy at iterate $t$, $\pi_t$, differ in absolute performance, i.e., $\mathbb{E}_{s \sim \rho}[|V^\mu(s) - V^{\pi_t}(s)|]$. When $\mathbb{E}_{s \sim \rho}\mathbb{V}_{a \sim \pi_t}[A^{\pi_t}(s, a)] < \mathbb{E}_{s \sim \rho}\mathbb{V}_{a \sim \pi_t}[A^\mu(s, a)]$, and $\mu$ is complementary to $\pi_t$, i.e., $\mathbb{E}_{s \sim \rho}\mathbb{E}_{a \sim \pi_t}|Q^{\pi_t}(s, a) - Q^\mu(s_h, a)| = \Omega(\eta)$, then $\mathbb{E}_{s \sim \rho}[V^{\pi_{t+1}}(s) - V^{\pi_t}(s)] \gtrsim \eta^2$.*

*Proof.* We begin by proving an upper bound on the disagreement between prover and base policy:

$$\mathbb{E}_{s \sim \rho} \mathbb{E}_{a \sim \pi_t} |Q^\mu(s,a) - Q^{\pi_t}(s,a)|$$

$$\leq \mathbb{E}_{s \sim \rho} \mathbb{E}_{a \sim \pi_t} |Q^\mu(s,a) - V^\mu(s,a)| + \mathbb{E}_{s \sim \rho} \mathbb{E}_{a \sim \pi_t} |Q^{\pi_t}(s,a) - V^\mu(s,a)|$$

$$\leq \mathbb{E}_{s \sim \rho} \mathbb{E}_{a \sim \pi_t} |Q^\mu(s,a) - V^\mu(s,a)| + \mathbb{E}_{s \sim \rho} [|V^\mu(s) - V^{\pi_t}(s)|] + \mathbb{E}_{s \sim \rho} \mathbb{E}_{a \sim \pi_t} |Q^{\pi_t}(s,a) - V^\mu(s,a)|$$

$$\leq \eta + \mathbb{E}_{s \sim \rho} \sqrt{\mathbb{V}_{\pi_t} [A^\mu(s,a)]} + \mathbb{E}_{s \sim \rho} \sqrt{\mathbb{V}_{\pi_t} [A^{\pi_t}(s,a)]},$$

where the last inequality uses Cauchy-Schwartz. Next we apply Jensen's inequality on the terms in the square root, and the conditions that $\mathbb{E}_{s \sim \rho} \mathbb{E}_{a \sim \pi_t} |Q^{\pi_t}(s,a) - Q^\mu(s_h,a)| = \Omega(\eta)$ to conclude:

$$\sqrt{\mathbb{E}_{s \sim \rho} \mathbb{V}_{\pi_t} [A^\mu(s,a)]} \gtrsim \eta,$$

whenever $\mathbb{E}_{s \sim \rho} \mathbb{V}_{a \sim \pi_t} [A^{\pi_t}(s,a)] < \mathbb{E}_{s \sim \rho} \mathbb{V}_{a \sim \pi_t} [A^\mu(s,a)]$. Invoking the result in Theorem 3.1 yields:

$$E_{s \sim \rho}[V^{\pi_{t+1}}(s) - V^{\pi_t}(s)] \gtrsim \eta^2$$

$\square$

## G   EXAMPLES GENERATED BY BASE POLICY TRAINED ON $Q^\pi + \alpha Q^\mu$.

When we train the base policy with reinforcement learning where the reward is $Q^\pi + \alpha Q^\mu$, instead of the effective reward $Q^\pi + \alpha A^\mu$, we find that the policy training converges to contrived solutions with degenerate failures. For example, we see in multiple examples (like the four below), that for every question, the model learns to output "REPHRASE THE PROBLEM" as the first step. This is because under $Q^\mu$ this trivial step achieves a positive reward. While there exists a better solution where the policy outputs the correct answer and achieves higher rewards, it fails to recover from this local optima and find a better solution. Across multiple training runs, we observed similar trends when training with $Q^\pi + \alpha Q^\mu$. Note that in the idealized update, where we observe advantages $A^\mu$ or value function $Q^m u$ on all actions at a state (*e.g.*, in the NPG update in Sec. 3.4), using $Q^\pi + \alpha Q^\mu$ is equivalent to $Q^\pi + \alpha A^\mu$. But, in practice, $A^\mu$ offers significant benefits in the form of variance reduction in policy gradient iterates. It does that by not assigning reward to steps like "REPHRASE THE PROBLEM", since the likelihood of a prover solving before or after this step is expected to be roughly the same (*i.e.*, the prover policy makes no progress).

---

**Example 1**

```
Q: My club has 25 members.  In how many ways can I choose members to form a 4-person
executive committee?

A:

>>>
REPHRASE THE PROBLEM
WE NEED TO FIND THE NUMBER OF WAYS TO CHOOSE 4 MEMBERS OUT OF 25 FOR THE EXECUTIVE
COMMITTEE SO WE CAN USE THE COMBINATION FORMULA WHICH IS
N C R = N! / (R! * (N - R)!) WHERE N IS THE TOTAL NUMBER OF MEMBERS AND R IS
THE NUMBER OF MEMBERS WE CHOOSE SO IN THIS CASE N = 25 AND R = 4 SO WE PLUG THEM IN AND
SIMPLIFY 25 C 4 = 25! / (4! * (25 - 4)!) = (25 * 24 * 23 * 22) / (4 * 3 * 2 * 1) = 104005
# Answer

104005
```

---

**Example 2**

```
Q: We have a triangle $\triangle ABC$ and a point $K$ on $BC$ such that $AK$ is
an altitude of $\triangle ABC$. If $AC = 10,$ $BK = 7$, and $BC = 13,$ then
what is the area of $\triangle ABC$?

A:

>>>
REPHRASE THE PROBLEM
WE NEED TO FIND THE AREA OF $\triangle ABC$ SO WE CAN USE THE FORMULA $A = \frac{1}{2}bh$
WHERE $b$ IS THE BASE AND $h$ IS THE HEIGHT SO SINCE $AK$ IS AN ALTITUDE, THAT MEANS
$AK \perp BC$ SO WE CAN USE $BC$ AS THE BASE AND $AK$ AS THE HEIGHT SO NOW WE NEED
```

```
TO FIND $AK$ SO WE CAN USE THE PYTHAGORE THEOREM ON $\triangle AKC$ SINCE $AC = 10$
# Answer

48
```

**Example 3**

```
Q: Suppose that $n, n+1, n+2, n+3, n+4$ are five consecutive integers.

Determine a simplified expression for the sum of these five consecutive integers.

A:

>>>
REPHRASE THE PROBLEM
WE NEED TO FIND THE SUM OF THESE FIVE CONSECUTIVE INTEGERS SO WE CAN USE THE FORMULA
$S_n = \frac{n}{2}(a_1 + a_n)$ WHERE $S_n$ IS THE SUM, $n$ IS THE NUMBER OF TERMS,
$a_1$ IS THE FIRST TERM, AND $a_n$ IS THE LAST TERM SO IN THIS CASE $n = 5, a_1 = n,
a_5 = n + 4$
# Answer

5n + 10
```

**Example 4**

```
Q: What is the value of $x^2+y^2-z^2+2xy$ when $x=-3$, $y=5$, and $z=-4$?

A:

>>>
REPHRASE THE PROBLEM
WE NEED TO FIND THE VALUE OF $x^2+y^2-z^2+2xy$ WHEN $x=-3$, $y=5$, AND $z=-4$
SO WE JUST PLUG IN THE VALUES AND SIMPLIFY
# Answer

-2
```

## H   COMPUTATIONAL COST OF TRAINING AND USING PAVS OVER ORMS

The training data for PAVs is collected by running rollouts to estimate the Q-values for each step. Since learning to predict expected future success at earlier steps is a statistically more challenging task than predicting the final score at the end of the complete sequence (as in ORM) , the training data for PAVs is larger and scales with the average number of steps in the LLM's output ($\approx 10$ in our case), in order to be able to achieve the same level of prediction accuracy on all steps in a generation. With a 10x larger dataset, naturally the computational cost of training PAVs (compared to ORMs which only predict final outcomes) also scales by roughly the same factor.

Despite a larger cost for training PAVs, once trained, PAVs are much more compute-efficient than ORMs on test questions. In other words, while we do incur a larger training cost, this cost is amortized over rounds of deployment. Concretely, as we show in Section 4.1, for test-time search, PAVs are 1.5-5x more compute efficient than ORMs, at achieving the same level of performance (Figure 4 in submission). This means that if we were to use a verifier at least twice upon deployment, PAV is already more compute-efficient than an ORM, accounting for both the training and test-time compute cost of PAVs.

For online RL, as we show in Section 4 (Figure 7c), PAVs achieve the same level performance in $1/6^{\text{th}}$ of the RL training iterations that ORMs require. But in our implementation, for each iteration, PAVs score each step by feeding each prefix in a generation separately through the trained model, and ORMs only score the full trajectory. Thus, in this implementation PAVs consume 10x more compute per batch to score the generations in a batch (since we have 10 steps on average per generation). A more efficient implementation would score all prefixes in a generation in a single forward pass through the trained PAV. Nevertheless, the reduction in RL iteration complexity with PAVs is big enough that the overall computational cost is less, compared to ORMs. For example, using the

formula for training and inference FLOPs in Hoffmann et al. (2022), during online RL, to train the Gemma 2B base policy with PAVs, we need to spend about $2.5 \times 10^{18}$ FLOPs, but to achieve the same performance with ORMs, we need about $5.9 \times 10^{18}$ FLOPs, resulting in at least half the total computational FLOPs.

## I  CONNECTING THE EFFECTIVE REWARDS TO REWARD-SHAPING POTENTIAL FUNCTIONS

The effective reward we define in equation 5 matches the definition of a potential-based reward shaping term, as defined in Ng et al. (1999). Before we explain why this is the case, we provide some background below.

**Background on potential based reward-shaping functions**: In Ng et al. (1999), instead of learning a policy for a reward function $R$, the policy is trained to optimize the transformed reward $R + F$. Here, $F(s_h, a_h, s_{h+1})$ is a reward shaping function, that takes as input the the current state $s_h$, action taken by the policy $a_h$, and next state $s_{h+1}$, and maps this to a scalar reward. In particular, they show that when the shaping function $F$ is potential-based, i.e., it is of the form:

$$F(s_h, a_h, s_{h+1}) = \Phi(s_{h+1}) - \Phi(s_h),$$

for some state-dependent potential function $\Phi$, then the policy that optimizes the transformed reward $R + F$, also optimizes the original reward $F$.

In Equation 5, our effective reward function is: $Q^\pi(s_h, a_h) + \alpha A^\mu(s_h, a_h)$. This reward function matches the functional form of the transformed reward function $R + F$ defined in Ng. et. al. (1999). Here, $Q^\pi$ corresponds to the reward $R$, and $\alpha A^\mu(s_h, a_h)$ is the potential-based reward function $F$. To see why $F$ satisfies this definition we use the definition of advantages from Equation 2. We can write $\alpha A^\mu(s_h, a_h)$ as:

$$\alpha(Q^\mu(s_h, a_h) - V^\mu(s_h)) = \alpha(V^\mu(s_{h+1}) - V^\mu(s_h)).$$

Following the theoretical result on potential-based reward shaping functions in Ng et al. (1999), the optimal policy under our effective reward $Q^\pi + \alpha A^\mu$ also lies in the set of optimal policies which only optimize the outcome reward (final answer correctness, or $Q^\pi$).

## J  DISCUSSION, LIMITATIONS AND FUTURE WORK

We began our exposition with the following question: how to define process rewards such that optimizing the base policy against process rewards ultimately improves the outcome level correctness of the final answer? Our key finding is that process rewards defined as advantages of a prover policy, distinct from the base policy improve the efficiency of exploration for steps sampled from the base policy during test-time search and online RL. This improved exploration in turn leads to discovery of better solutions, resulting in a higher accuracy on the math reasoning task. We also formally characterized the set of good prover policies as policies with step-level advantages that meaningfully contrast steps generated by the base policy, while still producing step-level advantages that are aligned with the base policy. Having trained process advantage verifiers (PAVs) to predict advantages under the prover policy, we empirically observed that test-time search against the trained PAVs improve the compute-efficiency of search by $1.5 - 5\times$, and accuracy of search by over $8\%$ compared to running best-of-$N$ against an ORM. Next, we present one of the significant results that validate the use of dense supervision when optimizing the base policy with online RL. Specifically, we show that dense online RL with rewards from our trained PAVs, improves sample efficiency of online RL by $5 - 6\times$, and results in an accuracy gain of over $6\%$.

**Limitations.** Despite the promise of our results, there are several limitations to our work that present important avenues for future research. First, while we can easily compute the right hand side of our result in Theorem 3.1 to understand whether a given prover policy will improve a fixed base policy, it is unclear how to automatically design a flexible class of optimal (or very good) prover policies for a sequence of base policy iterates. Perhaps simultaneously optimizing the prover and the base policy (in a two-player game) might provide for an approach to obtain the best prover during RL, but this is largely an open question. Second, since inevitably learning a process advantage verifier

(PAV) will incur fitting errors and this upper bounds peformance of our method. Fitting errors can be circumvented if our approach if we can simply run rollouts from prover policies during online RL or search to estimate advantages without training verifiers, and extending our approach to this setup is a good avenue for future work.

# K COMPARING PAVs WITH METHODS DESIGNED TO IMPROVE TEST-TIME EXPLORATION

In this section, we aim to explicitly evaluate the exploration capabilities of PAV. Note that our goal is not to claim that PAVs are the best approach to exploration, but instead to compare PAVs to an existing exploration method to understand if building on our approach of connecting process rewards with exploration and potential functions, and designing novel forms of process rewards could be a fruitful endeavor for future research on exploration in LLMs.

Specifically, we compare test-time beam search guided by process supervision from PAVs with the importance weighted search approach outlined in AlphaLLM (Tian et al., 2024), an approach that runs MCTS for search. We use the heuristic from AlphaLLM and our preliminary results show that PAVs are 8x more compute efficient at test-time beam search, as outlined below.

**Background on Alpha LLM**: For the importance weighted search approach of AlphaLLM, we implement beam search in the following way. At any given point, the beam consists of $N$ states $s_1, s_2, s_3, \ldots, s_N$. These are partially unrolled solutions from the base policy $\pi$, up until state $s_i$ (prefix). We then expand each node in the beam 3 times, by conditionally sampling from $\pi$ (conditioned on each of the states in the beam). We get $N \times 3$ states, which we rank with the following scoring function and then select the top $N$ state. Each of the new expanded states is of the form $s, a$, where $s$ is the previous state in the beam and $a \sim \pi(\cdot \mid s)$ is the new sampled action (step). Following, Section 4.3 from Tian et al. (2024), the score for the new state $(s, a)$ is $Q^\pi(s, a) + C \cdot U(s)$ where $U(s)$ is the uncertainty bonus for the state $s$, which is computed as $U(s) = \sqrt{n(s)/\sum_{i=1}^{N} n(s_i)}$. We use $C = 0.25$, which we identified by tuning performance over a hold out validation set we use for PAVs as well. This resembles UCB or UCT-style exploration. Concretely, $n(s)$ is the effective number of children for node $s$ (Section 4.3.2 in Tian et al. (2024)). The term $n(s) = C' \cdot I(s)$ is computed by linearly scaling the importance $I(s)$ defined as $I(s) = \max_a |V^\pi(s) - Q^\pi(s, a)|$, where $a$ is one of the 3 actions sampled from state $s$ when expanding the beam. We tune and set $C' = 2.0$. Intuitively, AlphaLLM chooses to explore states that can change the $Q$-values by a lot, when we continue to sample from them. When the $Q$-values deviate by a lot, the $I(s)$ term increases. Consequently, so does $N(s)$ (effective children count) and $U(s)$ (uncertainty bonus).

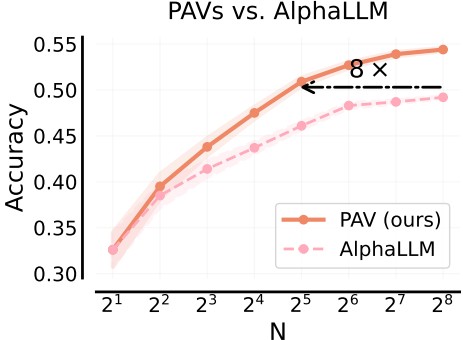

**Figure 18**: *Comparing the exploration capabilities of PAVs with AlphaLLM (Tian et al., 2024):* We find that PAVs are 8× more compute efficient. This is likely because the exploration metric in AlphaLLM (Tian et al., 2024) uses the absolute magnitude of the advantage under the base policy, vs. PAVs which use the signed advantage under the prover policy. Thus, our exploration metric prefers steps that increase the likelihood of a complementary prover to discover the correct solution, as opposed to preferring steps that simply change the previous state's value function (under the base policy) by the largest magnitude (which can also be negative).

## L WEAKER POLICIES ARE BETTER PROVERS FOR GEMMA 9B RL TRAINING

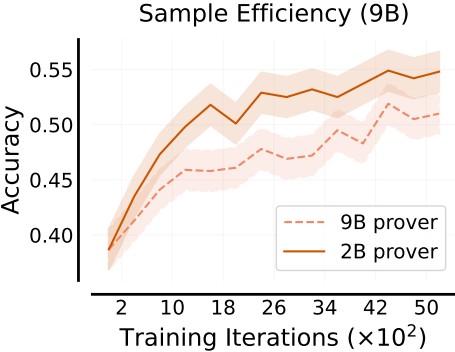

**Figure 19:** *Weaker Gemma 2B Policy is a better prover for Gemma 9B RL Training:* We train the Gemma 9B SFT policy further via online RL where the dense rewards are given by the effective rewards in equation 5. When we use the weaker Gemma 2B SFT policy as the prover, we are able improve the Gemma 9B policy much more than the effective rewards that use the stronger Gemma 9B SFT policy as the prover.

In Figure 19, we provide supporting empirical evidence for our claims in Section 5, where we stated that we found the weaker Gemma 2B SFT policy to be a better prover for updating the stronger Gemma 9B base policy with dense rewards during online RL, as opposed to using the Gemma 9B SFT policy as the prover. Furthermore, the Gemma 2B SFT prover rapidly becomes even weaker than the initial base policy (Gemma 9B SFT policy) within a few gradient steps of RL, but a fixed PAV that uses the weak Gemma 2B SFT prover is able to still sustain performance gains in RL.

