# OpenReview forum: "Rewarding Progress: Scaling Automated Process Verifiers for LLM Reasoning"
_ICLR.cc/2025/Conference — ICLR 2025 Spotlight_

### Official Review · Reviewer_UVHt · 2024-10-24

**Soundness:** 3
**Presentation:** 4
**Contribution:** 3
**Rating:** 8
**Confidence:** 1

**Summary:**

The paper proposes a method for designing rewards for Process Reward Models, used to improve reasoning in LLMs with step-by-step feedback rather than only outcome-based feedback. The authors relate the reward to a measure of how much a step changes the likelihood of producing a correct response in the future. They discuss their method for obtaining this measure and compare it to alternatives. They provide theoretical and empirical results to support their claims that their approach improves accuracy and efficiency.

**Strengths:**

I am not an expert in this specific area so I cannot confirm the originality or significance of the paper, but the authors do discuss related works and compare the paper with them. The paper is generally clearly written, though I suspect it is even easier to follow if you are more familiar with the topic than I am. I really like the ‘takeaways’ at the end of each section for improving intelligibility. The paper includes both theory and empirical results. I like that it discusses potential alternative solutions and why these were not pursued or would not work as well as the proposed approach.

**Weaknesses:**

Can you please point me to where you formally characterise what it means for the prover policy to be “too” misaligned with the base policy?

Does your work help improve the speed of reasoning by decreasing the number of steps required, or does it just improve the chances of finding the right answer eventually? I presume this relates to how long a rollout is. Related to this, you say in Line 82 that ‘advantages under an overly capable prover, that can succeed from any step, fail to distinguish between good and bad steps.’ Why do you not then consider the number of steps required to get to the solution from that point, as a way of quantifying ‘improvement’?

Do you formally quantify the exploration-exploitation trade-off that feeds into Result 3, since this seems important for your findings that you can improve accuracy?

Line 73 / 74 - repeated ‘the the combinatorial’.

**Questions:**

See weaknesses.

---

> ### Author Response · Authors · 2024-11-23
> **Response to Reviewer UVHt (Part I)**
>
> Thank you for the review and for a positive assessment of our paper! To address your concerns we add a new experiment that defines step-level process rewards using discounting, i.e. it takes into account the number of steps taken by the prover policy to complete the solution from the last step generated by the base policy. We also provide a pointwise responses to each of your questions below. **Please let us know if your concerns are addressed, and if so, we would be grateful if you are willing to raise your score.**
>
> >> **What does it mean for prover policy to be too misaligned with the base policy?**
>
> In the right hand side of Equation 6 (Theorem 3.1), we define the alignment between prover policy $\mu$ and current base policy $\pi$ as the expected inner product between $A^\mu(s, a)$ and $A^\pi(s, a)$, where the expectation is taken over actions $a \sim \pi(\cdot \mid s )$ sampled from the base policy $\pi$, and the distribution over states $\rho$. Mathematically, the alignment is denoted as $E_{s\sim \rho} E_{a\sim \pi(a \mid s)} A^\mu(s, a) A^\pi(s, a)$. In our theoretical result in Section 3.4, we show that process rewards defined as advantages of prover policies with higher variance (under actions sampled from the base policy) guarantee a stronger improvement in the base policy (this is the distinguishability term in Equation 6). At the same time, we cannot have the prover to be simply a high variance (entropy) policy which takes high values of $A^\mu$, without being correlated with the outcome rewards at all. Thus, we want prover policies such that the process rewards $A^\mu$ are correlated with the base policy advantages $A^\pi$, where $A^\pi$ prefers actions that achieve a high outcome reward under the base policy. This is exactly what is measured by our alignment term in Equation 6.
>
>
> >> **Accounting for steps taken by the strong prover policy to arrive at the correct answer**
>
> This is a great point and we add a new experiment that defines process rewards in a way that takes into account the number of steps taken by the prover policy to complete the solution.
>
> **We add a new result (Figure 16) and discussion on using discounted rewards from provers to Appendix E of the submission**. Here, we train PAVs to predict the advantages of discounted rewards from strong prover policies. Here, for the problem $\mathbf{x}$, and the state, action pair $s, a$, the process rewards are given by the effective reward from Equation 5: $Q^\pi + \alpha A^\mu$, except that the advantage $A^\mu$ is the difference in discounted rewards, i.e.:
>
> $A^\mu(s, a)  =  E_{y \sim \mu(\cdot \mid s, a)} \left[ \lambda^{\mathrm{len}(y)-\mathrm{len}(s)-1} \mathrm{Rex}(y, y^\star_{x})\right]  - E_{y \sim \mu(\cdot \mid s)} \left[  \lambda^{\mathrm{len}(y)-\mathrm{len}(s)} \mathrm{Rex}(y, y^\star_{x})\right],$
>
> where the prover policy samples solution $y$ with $\mathrm{len}(y)$ steps to complete the solution, from a state $s$, which already has $\mathrm{len}(s)$ steps in it.
>
> For this setting, we train a verifier to predict discounted rewards for the Gemma 9B prover policy. We find that the discounted process rewards from the stronger 9B prover policy performs worse than undiscounted rewards from the weaker Gemma 2B prover policy, when using either to train the 2B base policy with online RL.
>
> The main reason for the discounted rewards to not enable the use of strong provers is because strong prover policies tend to disregard states generated by the base policy (as illustrated in Figure 2b). This means, that irrespective of whether the weak prover policy generates a partially correct or incorrect solution, when we rollout the strong prover policy from this state generated by the base policy, the strong prover directly attempts to answer the math problem with its own solution trace. Thus, from any state the strong prover is expected to complete the solution with roughly the same number of steps. This means that $A^\mu \approx 0$ even in the discounted case, which reduces the ability of the strong prover policy to distinguish steps taken by the base policy.

---

> > ### Author Response · Authors · 2024-11-23
> > **Response to Reviewer UVHt (Part II)**
> >
> > >> **Exploration-exploitation tradeoff – how does it feed into result 3?**
> >
> > Thank you for this question. We have added the following discussion to Appendix C.
> >
> > For process rewards defined as our effective reward $Q^\pi + \alpha A^\mu$, as $\alpha \rightarrow 0$, the process rewards are purely exploitative, i.e. only upweight steps that are already likely to each the correct solution, under the current base policy $\pi$. On the other hand, as we increase $\alpha > 0$, then we also upweight steps that are preferred under the prover policy $\mu$. When the prover policy and the base policy are not too misaligned i.e., $E_\pi A^\mu A^\pi$ is sufficiently high, then in expectation, we expect for the the effective reward to explore new solution traces that might help the discovery of the correct solution, when sampling solutions from the base policy $\pi$. This is because, it is possible that some solution trace (set of steps) has low probability under $\pi$, but because it is preferred (high $A^\mu$ for each step in the trace) by a complementary prover policy $\mu$, the trace ends up getting a positive reward, and gets up-weighted. This aids in the discovery of the correct solution. We explain this with an illustrative example in Figure 2 (Section 3.1, L177-188). We support this hypothesis with empirical results in Result 3 of Section 4 and Result 3 of Section 5. In particular, the Result 3 of Section 5 which shows how the policy trained with our effective rewards is better at discovering solutions to hard problems, compared to the policy only trained on outcome rewards.
> >
> >
> >
> > >> **Do our process rewards improve speed of reasoning or improve discovery of correct solutions?**
> >
> > We do not incentivize shorter solutions beyond only rewarding solutions that reach a final answer in 1024 tokens or less. When we explicitly tried to account for the solution length (in terms of the number of steps) when computing process rewards (see answer above), we did not observe any improvement in performance over the process rewards that do not penalize longer solutions. Having said that, we believe that accounting for step length is a promising direction of designing process rewards to improve the efficiency of test-time search. Currently, we are mainly focused on coverage (discovery of correct solutions), but as models get better at solving problems via test-time search, training models by rewarding shorter solutions over longer ones can prove to be an effective way to optimize test-time compute.
> >
> > >> **Typo Line 73 / 74**
> >
> > Thank you for pointing out the typo. We have fixed this in the submission.

---

> ### Comment · Reviewer_UVHt · 2024-11-25
>
> Thank you very much for addressing my questions. Since I am not very familiar with this area, I will wait to see if other reviewers raise their score above an 8 before considering changing my own, as I cannot speak confidently about its significance or contribution.

---

### Official Review · Reviewer_7XQD · 2024-10-31

**Soundness:** 3
**Presentation:** 4
**Contribution:** 3
**Rating:** 8
**Confidence:** 1

**Summary:**

This paper mainly discusses how to design process rewards when using process reward models (PRMs) to improve reasoning in large language models. The authors believe that the per-step process rewards should measure progress, or advantage, instead of absolute Q-values for a better explore-exploit tradeoff  during beam search and online RL. The advantages should be computed using a prover policy different from the base policy. To boost improvement of the base policy, the prover policy should be able to distinguish actions taken by the base policy but are not too misaligned from the base. Based on this insight, the authors introduce process advantage verifiers (PAVs) and show that PAVs could largely scale test-time compute and RL sample efficiency compared to outcome reward models (ORMs).

**Strengths:**

1. This paper is well-organized and well-written. The concepts and ideas are clear and could be easily understood.
2. The results are promising. The proposed method (PAVs) are > 8% more accurate and 1.5-5x more compute-efficient than ORMs.
3. The insight to define process rewards as progress (or advantage) is inspiring. The method proposed (PAV) is novel.
4. The paper provides comprehensive analysis and guidance on how to choose prover policies and how to collect data to train PAVs, which is very beneficial to the community.

**Weaknesses:**

1. In the experiments, only ORM is presented as a baseline, without any inclusion of previous PRM methods. It would be more helpful if you include baseline methods such as [1] Wang et al. (2024), [2] Shao et al. (2024) or [3] Luo et al. (2024) to demonstrate how they perform poorly compare to your method. Alternatively, could you clarify if there were any particular challenges in implementing or comparing to these previous PRM approaches?
2. Although the theoretical analysis is solid, could you provide a small, concrete example to illustrate why advantages are more effective than value functions as process rewards? This could help readers to understand your statement in Section 3.1 (Process rewards should be advantages not value functions) better.

[1] Wang, P., Li, L., Shao, Z., Xu, R. X., Dai, D., Li, Y., Chen, D., Wu, Y., & Sui, Z. (2024). Math-Shepherd: Verify and Reinforce LLMs Step-by-step without Human Annotations. https://arxiv.org/abs/2312.08935

[2] Shao, Z., Wang, P., Zhu, Q., Xu, R., Song, J., Bi, X., Zhang, H., Zhang, M., Li, Y. K., Wu, Y., & Guo, D. (2024). DeepSeekMath: Pushing the Limits of Mathematical Reasoning in Open Language Models. https://arxiv.org/abs/2402.03300

[3] Luo, L., Liu, Y., Liu, R., Phatale, S., Lara, H., Li, Y., Shu, L., Zhu, Y., Meng, L., Sun, J., & Rastogi, A. (2024). Improve Mathematical Reasoning in Language Models by Automated Process Supervision. https://arxiv.org/abs/2406.06592

**Questions:**

See weaknesses.

---

> ### Author Response · Authors · 2024-11-23
> **Response to Reviewer 7XQD (Part I)**
>
> Thank you for the review and for a positive assessment of our paper! To address your concerns, we have made the comparison with other PRM models from prior works more explicit, and have added a new experiment where we compare RL training with PAVs and RL training with PRMs from prior works. We also respond to your other question on advantages vs. value functions. **Please let us know if your concerns are addressed, and if so, we would be grateful if you are willing to raise your score.**
>
> >> **Comparing with baselines that use PRMs proposed in prior works**
>
> **PRM baselines for test-time search**
> For test-time search, our results in Fig 5a of our submission already include both baselines: 1) beam search; and 2) best-of-N search using PRMs from prior works. **We make this more explicit with a new plot (Figure 13) we add in Appendix C**. We find that beam search with PAVs is 1.5x-5x more compute efficient than both beam search with PRMs from prior works, and best-of-N search with PRMs from prior works. We do not attempt to compare beam search with PAVs against weaker search algorithms used in conjunction with prior PRMs. When evaluating prior work (Snell et. al. (2024)) that uses $Q^\pi$ as their proposed PRM, we run beam search where the states in the beam are ranked only using $Q^\pi$, as opposed to the effective reward $Q^\pi + \alpha A^\mu$ in our case. Thus, the search procedure in the prior work and ours is identical, and we only change the re-ranking mechanism.
>
> Other works on PRMs (Wang et. al. (2024), Luo et. al. (2024)) that also propose to use $Q^\pi$, use PRMs for best-of-N search, where they only rank the full sequences using the trained PRM (by taking the minimum over the $Q^\pi$ values at each step in the generation sampled from the base policy $\pi$). For completeness, we also compare PAVs with these approaches (PRMs-as-ORMs), which performs similarly to using PAVs-as-ORMs in Figure 5a.
>
> **PRM baselines for online RL training**
> We add a new experiment where we use the PRMs proposed in prior works on test-time search, (Wang et. al. (2024), Luo et. al. (2024), Snell et. al. (2024)) as step-level rewards for training a base policy with RL. Here, the PRMs are trained offline and fixed, and then used to assign scores to intermediate steps in trajectories sampled during online RL.
> For this, we add a new experimental result where we use $Q^\pi_{\mathrm{base}}$ as the step-level score during RL training initialized with the base policy $\pi_{\mathrm{base}}$ since $Q^\pi_{\mathrm{base}}$ is the PRM proposed in prior works. This step-level reward is used to reward trajectories sampled during online RL training,  in addition to the outcome reward, similar to our effective reward  in Equation 5. We find that the test accuracy drops quite quickly, even though the train rewards keep improving, since the policy just learns to hack the step-level Q-values, for the reasons we explain in L240-245. We see a similar collapse when we use $Q^\mu$ instead of $A^\mu$ (see Appendix G for qualitative examples).
> On the other hand, if we were to use the Q-values or advantages from the current policy iterate $\pi_t$ as the step-level reward, then that is equivalent to only optimizing the outcome reward, and the only benefit of using the step rewards would be to reduce variance during online RL iterations. Thus, for online RL as well, our proposed step-level rewards (PAVs) which uses advantages of the prover policy $A^\mu$ outperforms baselines that plugin PRMs proposed in prior works on test-time search, **We have added this new experiment on online RL where step-level rewards are given by PRM  $Q^\pi_{\mathrm{base}}$ proposed in prior works  to Appendix E, Figure 17**.

---

> > ### Author Response · Authors · 2024-11-23
> > **Response to Reviewer 7XQD (Part II)**
> >
> > >> **Illustrative example of Advantages over Value Functions**
> >
> > In Figure 2a we provide an illustration of why PRMs that use a combination of advantages and value functions improve exploration over PRMs that only use value functions. We show this in the context of test-time beam search. In particular, we show that advantages can promote more diversity over steps in solution traces by decoupling the evaluation of the action (step) from that of the previous state (prefix). We provide the following explanation in L178-188 of the submission.
> >
> > From the $2$ states in the beam, we sample $3$ actions. If we pick next states purely based on highest values of $Q^\pi$, we would be comparing steps sampled from different states ($a_{1,1}$ vs. $a_{2,1}$) against each other. Clearly, a reduction in expected final outcome, i.e.,  $Q^\pi(s_1, a_{1,1}) - V^\pi(s_1)$, means that $a_{1, 1}$  *by itself* has a negative effect of $-0.05$ on the probability of success from $s_1$, whereas $a_{2,1}$ has a positive effect of $+0.20$ from $s_2$. However, expanding the beam based on *absolute* values of $Q^\pi$ retains the action that makes negative progress, and removes state $s_2$ from the beam (as beam size is 2). In other words, $Q^\pi$ fails to decouple the ''evaluation'' of an action (step), from the ''promise'' shown by the previous state. This will not be an issue for every problem, and particularly not when the beam capacity is unbounded, but under finite computational and sampling constraints, using $Q^\pi$ might retain states with potentially unfavorable steps that hurt the overall likelihood of success. If we could also also utilize the progress made by the previous step along with the likelihood of success $Q^\pi$ when deciding what to retain in the beam, then we can address this tradeoff.

---

> > > ### Comment · Reviewer_7XQD · 2024-11-27
> > >
> > > I appreciate the authors for addressing my concerns! However, I am not very familiar with the area and cannot confidently assess the contribution and in-depth quality of this paper. After reading the discussions between the authors and other reviewers, I decide to keep my score at 8.

---

### Official Review · Reviewer_jZuo · 2024-11-01

**Soundness:** 3
**Presentation:** 3
**Contribution:** 3
**Rating:** 6
**Confidence:** 3

**Summary:**

The paper proposes process advantage verifiers (PAVs) that provide process rewards improve the efficiency of test-time search leading to better accuracy. Outcome reward models (ORMs) and process reward models (PRMs) assign reward to the final outcome/per-step respectively. The paper proposes the use of PRMs as a dense reward generator but changing the reward allocation to provide advantage values instead so that better exploration may be encouraged. The key intuition is that reinforcing promising steps that make progress towards the goal improves exploration of possible answers and is crucial when employing search-based strategies such as beam-search.

The authors utilize a toy example to show that Q-values may be retaining unfavorable steps reducing the overall likelihood of success. Thus, the authors resort to the advantage framework (Sutton et al) and show that by considering advantages (relative increase/decreases in likelihoods of success) we can mitigate the drawbacks of only using Q-values.

The authors then introduce PAVs and introduce the optimization objective. The  authors argue that using advantage values derived from the same base policy would not be informative. The authors then propose to use a different prover policy (this lightly connects the optimization objective with off-policy learning). The paper then shows that a good process reward measure is a prover policy that provides a notion of progress. The authors analyze this hypothesis using a toy example. The authors then showcase a theoretical intuition that states that a prover can be expected to improve a base policy when it is able to distinguish different actions taken by the base policy. The authors then showcase that Best-of-K policies (sampling of K policies from $\pi$ and then using the best one) contains complementary provers.

Finally, the authors propose an empirical evaluation to showcase the advantage of PAVs over ORMs.

**Strengths:**

S1. The paper tackles an interesting concept and the notion of using advantages is quite intuitive. The use of advantages is well known in the AI community and its application to neural verifiers is a novel contribution.

S2. The analysis of showing that weak provers may amplify stronger base policies is convincing.

**Weaknesses:**

W1. The paper is generally well-written but is a bit confusing to follow at times. There seems to be a lot of \vspace manipulation that conflicts with ICLR paper guidelines (example sec 3.1, ICLR guidelines stipulate that there must be 1 line of space after figure caption). This made the paper a bit hard to read. The related work is very light and the conclusion is non-existent with no future work or limitations discussed. I believe that the paper could have reduced some of the analysis and expanded on this further.

W2. The paper focuses its empirical evaluation on ORMs and states that there are major advantages w.r.t. them but I believe that a fair comparison would be to use PRMs since they are the closest possible baseline. The authors do prempt this by stating that PRMs have only demonstrated 1-2% improvement w.r.t. ORMs but that is in the context of best-of-N search. There are no comparisons with PRMs except for Fig 5a. However, this improvement is not discussed nor is the setting adequately mentioned making it hard to evaluate. Are the PRM results in Fig 5a computed using beam-search or best-of-N search.

W3. There are recent results of utilizing beam search or some tree-based search with PRMs [1, 2]. This would likely be more compute efficient than ORMs or PRMs utilizing best-of-N search. As a result, the compute efficiency would be best to be compared to them. Is there any reason this was not considered.

W4. The paper claims to have significant improvements but lacks a comprehensive comparison of PRMs w.r.t. standard benchmarks as reported in other papers. I am unable to distill where the claims of massive improvement stem from esp considering that two different search strategies are employed and in the case where PAVs-as-ORMs are used, the improvement drops to 4%. Similar comments for results in Sec 5. PRMs are a natural fit for RL as well I would assume.

Could the authors please justify their empirical evaluation. There is a lack of evaluations on GSM8k and other standard datasets and some baseline combinations that utilize beam search are not expressed. I fully appreciate the authors extensive ablations and analysis but I feel that to truly understand the utility of PAVs as neural verifiers/reward models, one would need to compare them with the same search strategy but just a different ranking scheme (PRMs vs PAVs). Could the authors please provide additional details here?

**Questions:**

Please address my concerns in the weaknesses.

Also, could you please elaborate on what you mean by lines 369-374? If I interpret the writing correctly, does it mean that PAVs cannot be used directly in beam-search without having access to PRMs and ORMs? It would be great to clarify my concerns and perhaps improve the clarity of the paper to make it more accessible.

Overall it is a good paper. I hope the authors can resolve my concerns.

---

> ### Author Response · Authors · 2024-11-19
> **Response to Reviewer jZuo (Part I)**
>
> Thank you for the review! To address your concerns, we clarify that for experiments in Section 4 (test-time search), we already compare with several prior works that use PRMs for beam search and best-of-N search. If there is a comparison that we are missing, please let us know and we will be happy to add it. We also include a new experiment for RL training with PRM baselines that we used for test-time search. We also respond to your other questions and concerns below. **Please let us know if your concerns are addressed, and if so, we would be grateful if you are willing to raise your score.**  We are happy to discuss further!
>
>
> >> **Comparing beam search with PAVs and search with PRMs from prior works**
>
> For test-time search, our results in Fig 5a of our submission already include both baselines: 1) beam search; and 2) best-of-N search using PRMs from prior works. **We make this more explicit with a new plot (Figure 13) we add in Appendix C**. We find that beam search with PAVs is 1.5x-5x more compute efficient than both beam search with PRMs from prior works, and best-of-N search with PRMs from prior works. We do not attempt to compare beam search with PAVs against weaker search algorithms used in conjunction with prior PRMs.
>
> When evaluating prior work (Snell et. al. (2024)) that uses $Q^\pi$ as their proposed PRM, we run beam search where the states in the beam are ranked only using $Q^\pi$, as opposed to the effective reward $Q^\pi + \alpha A^\mu$ in our case. Thus, the search procedure in the prior work and ours is identical, and we only change the re-ranking mechanism.
>
> Other works on PRMs (Wang et. al. (2024), Luo et. al. (2024)) that also propose to use $Q^\pi$, use PRMs for best-of-N search, where they only rank the full sequences using the trained PRM (by taking the minimum over the $Q^\pi$ values at each step in the generation sampled from the base policy $\pi$). For completeness, we also compare PAVs with these approaches (PRMs-as-ORMs), which performs similarly to using PAVs-as-ORMs in Figure 5a.
>
> >> **PRM baselines for online RL training**
>
> We add a new experiment where we use the PRMs proposed in prior works on test-time search, (Wang et. al. (2024), Luo et. al. (2024), Snell et. al. (2024)) as step-level rewards for training a base policy with RL. Here, the PRMs are trained offline and fixed, and then used to assign scores to intermediate steps in trajectories sampled during online RL.
> For this, we add a new experimental result where we use $Q^\pi_{\mathrm{base}}$ as the step-level score during RL training initialized with the base policy $\pi_{\mathrm{base}}$ since $Q^\pi_{\mathrm{base}}$ is the PRM proposed in prior works. This step-level reward is used to reward trajectories sampled during online RL training,  in addition to the outcome reward, similar to our effective reward  in Equation 5. We find that the test accuracy drops quite quickly, even though the train rewards keep improving, since the policy just learns to hack the step-level Q-values, for the reasons we explain in L240-245. We see a similar collapse when we use $Q^\mu$ instead of $A^\mu$ (see Appendix G for qualitative examples).
> On the other hand, if we were to use the Q-values or advantages from the current policy iterate $\pi_t$ as the step-level reward, then that is equivalent to only optimizing the outcome reward, and the only benefit of using the step rewards would be to reduce variance during online RL iterations. Thus, for online RL as well, our proposed step-level rewards (PAVs) which uses advantages of the prover policy $A^\mu$ outperforms baselines that plugin PRMs proposed in prior works on test-time search, **We have added this new experiment on online RL where step-level rewards are given by PRM  $Q^\pi_{\mathrm{base}}$ proposed in prior works  to Appendix E, Figure 17**.
>
> >> **Expanding on our explanation in L369-374 on how to use PAVs during test-time.**
>
> For test-time search over responses from the base policy $\pi$, and fixing a prover policy $\mu$, the following is how we use PAVs for beam search at test-time. At any given time step, the beam holds partially generated responses (states) to a math problem. We score each of the states using our effective reward $Q^\pi + \alpha A^\mu$. Since $A^\mu$ is computed using $Q^\mu$ (Equation 2 in submission), we need only train verifiers to predict $Q^\pi$ and $Q^\mu$. Finally, at the end of beam search, when only we have complete responses in the beam, we pick the best response using an ORM trained only to predict the correctness of final answers. In practice, we find that using a trained ORM to rate complete responses at the end of beam search is not critical, and using the PRM that predicts $Q^\pi$  to rank the full responses also performs similarly. So, the training of an ORM model can be avoided. **We have added this discussion to Appendix D.** Please let us know if it is still unclear and we would be happy to expand on this further.

---

> > ### Author Response · Authors · 2024-11-19
> > **Response to Reviewer jZuo (Part II)**
> >
> > >> **Expanding on related works, adding a conclusion with limitations, improving spacing before Section 3.1.**
> >
> > In Appendix A, we expanded on all related works in detail. In Appendix K, we have now expanded on our short conclusion in L537-539, and also discuss some limitations of our current work and possible lines of future work. We have also added more white space between Figure 2 and Section 3.1 to improve readability. We agree with you that it might be better to move some discussion on related work and conclusion from the Appendices into the main submission, in place of some of the analysis. **If you could point us to some of the analysis that you felt was most appropriate to be relegated to the Appendix, we would be happy to do that, so that we can make some space in the main submission for related works and conclusion.**
> >
> >
> > >> **Empirical evaluation on GSM8k dataset**
> >
> > We choose the harder MATH benchmark (Hendrycks et. al. (2021)) for our empirical evaluation for two reasons. First, the performance on some other reasoning datasets like GSM8K is already saturated (for example, the performance of some of the base LLMs we consider like Gemma2-9B and Gemma2-27B is itself $>85%$ on GSM8K). Second, the MATH benchmark is common across all prior works that study process and outcome reward models (Snell et. al. (2024), Wang et. al. (2024), Lightman et. al. (2023), Shao et. al. (2024), Cobbe et. al. (2021)), whereas only a subset of these works also evaluate on GSM8k. This enables us to perform a direct comparison with all prior works.
> >
> > At the same time, we agree that expanding our results to GSM8k can only help to further strengthen our work. We are trying to add this result but are unsure if it will complete during the span of the rebuttal period, since we need to train base models, collect data to train PAVs, train the PAVs, and then use it for search/RL. That said, we will try to add it for the final version of the paper.

---

> > > ### Comment · Reviewer_jZuo · 2024-11-20
> > >
> > > Thank you for providing Fig. 13. That addresses my primary concern with the paper.
> > >
> > > Re: How to make space for the related work and the conclusion?
> > > I leave this to you. IMO Sec 4.2 might be a good candidate to mention in a sentence and move the details. Sec 1 could be shortened. Ultimately, I'd prefer if the authors can pitch their strongest points in the main paper while providing a fair comparison with related work and adequate conclusions without defering the reader to the appendix.

---

> > > > ### Author Response · Authors · 2024-11-20
> > > >
> > > > We are glad to note that **Fig 13 address your primary concern**. As per your suggestion, we have now cut down Section 4.2, moving details on the data collection strategy for PAVs to Appendix D. We use this space to now include all the relevant works in Section 6 of the main paper and add a conclusion with limitations, future work in Section 7 of the main paper.
> > > >
> > > > **We hope that this addresses all your outstanding concerns, and if so, we would be grateful if you are willing to raise your score.**

---

> > > > > ### Comment · Reviewer_jZuo · 2024-11-20
> > > > >
> > > > > Thanks for your response. I will improve my score by one point.
> > > > >
> > > > > I thank the authors for improving the related work and conclusion. I still think that its a bit slim and still violates the ICLR paper guidelines (no line spacing between section 7 and the preceding text). If the paper gets accepted I hope the authors are able to really distill and present the key points and make the paper a bit more "breathable".

---

> > > > > > ### Author Response · Authors · 2024-11-23
> > > > > > **Thank You!**
> > > > > >
> > > > > > Thank you for the response and raising the score. We have also made more updates to the writing and the paper to make it more breathable -- in particular, we believe spacing issues should be resolved and we have added a bigger discussion of related work in the main paper (all edits shown in the teal color). With these edits, we believe all primary related works discussed in Appendix A should now appear in Section 6.
> > > > > >
> > > > > > **Please let us know if this addresses your concern regarding spacing and formatting, and related works. If so, we would be grateful if you would be willing to further upgrade your score in the light of these revisions, thanks so much!**

---

> > > > > > > ### Comment · Reviewer_jZuo · 2024-11-25
> > > > > > >
> > > > > > > Thank you for your response.
> > > > > > >
> > > > > > > I will improve the presentation score of the paper with the added changes.
> > > > > > >
> > > > > > > I will maintain my overall score however. I believe testing this approach with more benchmarks like GSM8k as well as a different family of LLMs would have improved its impact.
> > > > > > >
> > > > > > > If the paper gets rejected, I'd encourage the authors to fix some of the issues pointed during this review process and resubmit. All the best!

---

### Official Review · Reviewer_Ys2B · 2024-11-02

**Soundness:** 4
**Presentation:** 4
**Contribution:** 4
**Rating:** 8
**Confidence:** 3

**Summary:**

The paper looks at the the problem of training LLMs for reasoning - which in this context means, given a problem, going through multiple steps to arrive at an answer. Authors argue that training and inference in reasoning models can be improved through extending simple outcome-based reward models with a dense, advantage-based reward models. The key insight is that the advantage should be computed under a separate ("verifier") policy, which is neither too strong (since any action under the weak base policy would be just ignored and verifier would succeed anyway) neither too weak (the verifier policy would fail anyway).
The paper first motivates and elaborates on those insights with some toy experiments and theoretical formalisation, and then applies the conceptual understanding to fine-tune Gemma LLMs (2B, 9B, 27B) on the MATH dataset, obtaining significantly better results (in sample efficiency, accuracy and compute efficiency) than models fine-tuned with outcome-based rewards investigated in the prior literature.

**Strengths:**

- The paper is very clearly written. It is full of intuition, it guides the reader through conceptual, toy-experimental, theoretical and empirical results, every step seems motivated, there are very helpful "Takeaway" summaries at the end of each section, the figures are clear and aid the understanding. There is a clean, coherent narrative. I enjoyed reading it a lot.
 - Formal results are not extremely complex, but they correspond to the empirical treatment well, and provide additional tools for the empirical part (such as motivating the use of best-of-k policies). The proofs (in the appendix) are formal and clearly written, and do not take shortcuts. Theoretical development includes the policy improvement framework and beam-search analysis for the best-of-k case.
 - Prior work, although in large part delegated to the attachment, is referenced extensively.
 - Experiments are convincing and done on a relatively large scale.
 - Overall, the method described in the paper seems clearly useful and promising.

**Weaknesses:**

- The (final) experimental section seems a bit too narrow. Although authors reference the "conventional belief" of using mathematical correctness or relevance of steps in the introduction, they only compare to the baseline of ORM reward. It is difficult to judge how much of an improvement we should expect in other domains, as other SOTA MATH models are only briefly referenced (and not compared to) in the appendix.
 - There are multiple places where the paper claims that the major benefit of using a correlated, high-variance verifier is to encourage exploration. But there are many ways to encourage exploration: epsilon-greedy policies, UCB, max-ent regularisation etc. It seems that the paper advocates for using the verifier only because it takes unnecessarily strong position on only using greedy beam search. This makes sense as the choice within the framework, but it again makes it difficult to judge how much of an improvement the new method really is, compared to those other techniques.
 - The insight of "it's bad to use extremely good expert policy to judge moves" seems a to apply less in a context where we use a discount factor $\neq 1$. A strong move would still help even a strong expert, if it saves it time to arrive at the solution. It is not clear to me whether a "need to generate simpler, repetitive, and even incorrect steps to explore and discover the final answer" really applies in general.
- I had trouble understanding when (or whether) introducing the sub-optimal verifier can result in a worse behavior, or when does $\mathbb{E}[V^{t+1} - V^t] < 0$. Some confusion arose because the verifier is initially assumed to be just a fixed $\mu$, while it is actually $\mu(\pi_t)$ (e.g. best-of-K($\pi_t$)).

**Questions:**

- How do the results change if we introduce non-unit discount factor?
- How to situate using PAV to encourage exploration among the alternative approaches already studied in the RL literature?
- Theorem 3.1 seems to have a typo, the second term should appear with a negative sign. In general, adding an advantage function is a form of potential shaping, as you note in the introduction, which means that the set of optimal policies should be preserved under any verifier. But the bound in Theorem 3.1 can be negative for very misaligned $\mu$ and $\pi$, can it? Does that mean that the update step can be negative? If yes - when can it happen?
 - In section 3.3, your reward $r$ is binary - what do you mean by "the maximum reward $r$ across $N$ samples" in Fig 3c)?
 - You grid-search for a good $\alpha$. Do you think that the value you found (3.0 - 5.0) will generalise across tasks? What is the cost of the search?
 - You want $\mu$ primarily to be able to distinguish between actions of $\pi$. Would it help if you ran multiple verifiers at the same time?
 - Remark 3.1 did not felt sufficiently motivated by the preceding paragraph.
 - Notation in the Appendix F: it might be worth spelling out that the distribution $d^\pi_s$ is marginalised over all future time steps (or just writing the definition). It's not clear to me why is $\theta_{s, a}$ assumed to be in $\mathbb{R}^d$. Typo in eq 8: it should read $a_{h+1}$. Typo above eq 18, should read $A^{t}$. Typo in line 1124, should read "equation 23". In general, it's not clear to me what happens in the step of moving to eq 27 from eq 26. What happens to $C_1, C_2, C_3, C_4$? What exactly is the meaning of $\lesssim$?

---

> ### Author Response · Authors · 2024-11-22
> **Response to Reviewer Ys2B (Part I)**
>
> Thank you for the review and for a positive assessment of our paper! To address your concerns, we have made the comparison with other PRM models from prior works more explicit, and have added a new experiment where we compare RL training with PAVs and RL training with PRMs (and not just ORMs) from prior works. We have also added another new experiment on using discounted rewards from strong provers, and provide answers to other questions below. **Please let us know if your concerns are addressed, and if so, we would be grateful if you are willing to raise your score.**
>
> >> **Comparing with baselines that use PRMs proposed in prior works**
>
> **PRM baselines for test-time search**
> For test-time search, our results in Fig 5a of our submission already include both baselines: 1) beam search; and 2) best-of-N search using PRMs from prior works. **We make this more explicit with a new plot (Figure 13) we add in Appendix C**. We find that beam search with PAVs is 1.5x-5x more compute efficient than both beam search with PRMs from prior works, and best-of-N search with PRMs from prior works. We do not attempt to compare beam search with PAVs against weaker search algorithms used in conjunction with prior PRMs. When evaluating prior work (Snell et. al. (2024)) that uses $Q^\pi$ as their proposed PRM, we run beam search where the states in the beam are ranked only using $Q^\pi$, as opposed to the effective reward $Q^\pi + \alpha A^\mu$ in our case. Thus, the search procedure in the prior work and ours is identical, and we only change the re-ranking mechanism.
>
> Other works on PRMs (Wang et. al. (2024), Luo et. al. (2024)) that also propose to use $Q^\pi$, use PRMs for best-of-N search, where they only rank the full sequences using the trained PRM (by taking the minimum over the $Q^\pi$ values at each step in the generation sampled from the base policy $\pi$). For completeness, we also compare PAVs with these approaches (PRMs-as-ORMs), which performs similarly to using PAVs-as-ORMs in Figure 5a.
>
> **PRM baselines for online RL training**
> We add a new experiment where we use the PRMs proposed in prior works on test-time search, (Wang et. al. (2024), Luo et. al. (2024), Snell et. al. (2024)) as step-level rewards for training a base policy with RL. Here, the PRMs are trained offline and fixed, and then used to assign scores to intermediate steps in trajectories sampled during online RL.
> For this, we add a new experimental result where we use $Q^\pi_{\mathrm{base}}$ as the step-level score during RL training initialized with the base policy $\pi_{\mathrm{base}}$ since $Q^\pi_{\mathrm{base}}$ is the PRM proposed in prior works. This step-level reward is used to reward trajectories sampled during online RL training,  in addition to the outcome reward, similar to our effective reward  in Equation 5. We find that the test accuracy drops quite quickly, even though the train rewards keep improving, since the policy just learns to hack the step-level Q-values, for the reasons we explain in L240-245. We see a similar collapse when we use $Q^\mu$ instead of $A^\mu$ (see Appendix G for qualitative examples).
> On the other hand, if we were to use the Q-values or advantages from the current policy iterate $\pi_t$ as the step-level reward, then that is equivalent to only optimizing the outcome reward, and the only benefit of using the step rewards would be to reduce variance during online RL iterations. Thus, for online RL as well, our proposed step-level rewards (PAVs) which uses advantages of the prover policy $A^\mu$ outperforms baselines that plugin PRMs proposed in prior works on test-time search, **We have added this new experiment on online RL where step-level rewards are given by PRM  $Q^\pi_{\mathrm{base}}$ proposed in prior works  to Appendix E, Figure 17**.

---

> > ### Author Response · Authors · 2024-11-22
> > **Response to Reviewer Ys2B (Part II)**
> >
> > >> **Discounting rewards from strong provers**
> >
> > This is a great suggestion! **We add a new result (Figure 16) and discussion on using discounted rewards from provers to Appendix E of the submission**. Here, we train PAVs to predict the advantages of discounted rewards from strong prover policies. Here, for the problem $\mathbf{x}$, and the state, action pair $s, a$, the process rewards are given by the effective reward from Equation 5: $Q^\pi + \alpha A^\mu$, except that the advantage $A^\mu$ is the difference in discounted rewards, i.e.:
> >
> > $A^\mu(s, a)  =  E_{y \sim \mu(\cdot \mid s, a)} \left[ \lambda^{\mathrm{len}(y)-\mathrm{len}(s)-1} \mathrm{Rex}(y, y^\star_{x})\right]  - E_{y \sim \mu(\cdot \mid s)} \left[  \lambda^{\mathrm{len}(y)-\mathrm{len}(s)} \mathrm{Rex}(y, y^\star_{x})\right],$
> >
> > where the prover policy samples solution $y$ with $\mathrm{len}(y)$ steps to complete the solution, from a state $s$, which already has $\mathrm{len}(s)$ steps in it.
> >
> > For this setting, we train a verifier to predict discounted rewards for the Gemma 9B prover policy. We find that the discounted process rewards from the stronger 9B prover policy performs worse than undiscounted rewards from the weaker Gemma 2B prover policy, when using either to train the 2B base policy with online RL.
> >
> > The main reason for the discounted rewards to not enable the use of strong provers is because strong prover policies tend to disregard states generated by the base policy (as illustrated in Figure 2b). This means, that irrespective of whether the weak prover policy generates a partially correct or incorrect solution, when we rollout the strong prover policy from this state generated by the base policy, the strong prover directly attempts to answer the math problem with its own solution trace. Thus, from any state the strong prover is expected to complete the solution with roughly the same number of steps. This means that $A^\mu \approx 0$ even in the discounted case, which reduces the ability of the strong prover policy to distinguish steps taken by the base policy.
> >
> > >> **Other strategies of incentivizing exploration: UCB, $\epsilon$-greedy, max-entropy**
> >
> > This is a great question, but in practice we hypothesize that it might be computationally infeasible to maximize the entropy or run $\epsilon$-greedy search over the space "steps", which are equivalent to actions in our setting.  This is because the steps are high-dimensional sequences of tokens (a single step generated by the LLM consists of 100s of tokens). Consequently, it is computationally infeasible to compute the probability distribution over steps (for count-based exploration) or enable exploration through algorithms like maximum entropy regularization over steps. Running these procedures at token-level would make learning statistically harder since the horizon (maximum number of steps generated for an input problem) of the response now blows up to 1000s of tokens (instead of 10 steps in our case or around 100 steps in RL settings when discount is set to 0.99). Finally, exploration by reranking multiple samples from the base model is also not enough because it may not sample diverse enough solutions.
> >
> > At the same time, we agree that computationally feasible relaxations of maximum-entropy or $\epsilon$-greedy strategies can enable efficient training/test time exploration. We will also add this as a future action item. That said, we note that the objective of our work is to understand how to define automated process rewards and what they can enable. We find that process rewards defined as advantages of an appropriately chosen prover policy can enable efficient training/test time exploration, but of course, there may be other ways of enabling exploration without defining process rewards. We are not claiming that this is the best way of performing exploration in LLMs. We would be happy to edit specific parts of the submission where this is unclear. We are not claiming that this is the best way of performing exploration in LLMs. We would be happy to edit specific parts of the submission where this is unclear.

---

> ### Author Response · Authors · 2024-11-22
> **Response to Reviewer Ys2B (Part III)**
>
> >> **Is there a need to generate simpler, repetitive and incorrect steps to explore and discover the correct answer?**
>
> We agree with you that it is unclear if there is a need to generate repetitive or incorrect steps to discover the correct answer. But, these steps enable the model to spend more training time compute to discover answers to hard problems, as we describe below.
>
> On hard questions (that were unsolved by the SFT model with pass@256), we clearly do not have coverage over the correct solution trace. Note that these are samples from the SFT model that was only trained to predict the correct final answer in least number of steps, and without any incorrect ste[s in between. So, for these hard questions, we hypothesize that the it is possible for the policy to spend more training time compute, by sampling simpler, repetitive or incorrect steps, that makes it easier to discover the answers for hard math questions. At the same time, simply generating repetitive steps is also bad, which is why we need process advantage verifiers to evaluate a step exactly based on its ability to improve the likelihood of arriving at the correct solution, as opposed to its mathematical relevance and correctness, as judged by a human. This is precisely what we set out to do in Section 3, where we choose process rewards to be potential functions that enable the optimization of outcome rewards (final answer correctness).
>
> >> **Confusion on fixed vs. varying prover, i.e., is $\mu  = \mathrm{BoK}(\pi_{t})$**?
>
> The prover policy is always fixed. When we say that the class of ''best-of-K'' policies serves as a class that contains good, ``complementary’’ prover policies, we always mean the best-of-K over the original base policy. Thus, we use $\mathrm{BoK}(\pi_{\mathrm{base}})$ to update the current policy $\pi_t$ during online RL. Here, $\pi_t$ is the policy at the $t^\mathrm{th}$ RL training iteration, and $\pi_0$ (initialization of RL training) is set to be the base policy $\pi_{\mathrm{base}}$.
>
> >> **We have matching optimal policies with potential functions, but the right hand side of the bound in Theorem 3.1 can be negative?**
>
> Yes, you are correct! Using the theoretical result in Ng et. al. (1999), we can claim that the set of optimal policies that optimize the outcome reward alone, should match the set of optimal policies for our effective reward $Q^\pi + \alpha A^\mu$, since the $\alpha A^\mu$ term is a potential-based reward shaping function (**we have added discussion on this in Appendix I**).
>
> The above argument does not break, even when $\mu$ and $\pi$ are highly misaligned. This is because, when $\mu$ and $\pi$ are misaligned, $E_\pi [A^\mu A^\pi]$ ends up being negative, which can make the right hand side of the lower bound in Theorem 3.1 negative, as you correctly note. But, this just makes the guarantee from policy improvement weaker. In other words, even though $E[V^\pi_{t+1}]$ is better than $E[V^\pi_t]$, we cannot guarantee the improvement using Theorem 3.1. On the other hand, whenever $\mu$ and $\pi$ are aligned, i.e., $E_\pi [A^\mu A^\pi] > 0$ and $Var_\pi [A^\mu]$ is large, we are guaranteed a stronger improvement in the policy iterates. **We hope this clarifies your confusion about Theorem 3.1, and we are happy to explain further if there is still any confusion**.
>
> >> **Clarifying our Remark 3.1 on the Best-of-K(base policy) set containing a good set of provers**
>
> We have updated the submission to explain the main motivation behind considering the class of best-of-K (over base policy). It is mainly to study a class of policies of increasing strengths (test performance), compared to the base policy. This class is conveniently parameterized (with $K$) for us to run a search over and identify a good prover policy. In Appendix F.4, (to which we add a reference in main submission) we explain what choice of $K$ can give us the best policy improvement lower bound from Theorem 3.1 .
>
> >> **Grid search on $\alpha$**
>
> For our RL experiments, we ran a grid search over $\alpha \in [0.5, 6.0]$, using intervals of $0.5$, and found optimal values of $3.0$ for the 9B model and $5.0$ for the 2B model. While this search was computationally expensive, we noted that all values of $5.5 > \alpha > 1.5$ significantly improved the performance of both 2B and 9B models trained with PAVs, over models trained with only ORMs. In fact tuning $\alpha$ on the smaller model (2B) is enough, and using the same for the 9B model already improves performance over ORM. This means that the choice of $\alpha$ is not a very sensitive one and in practice, we expect it to transfer across related problem instances.

---

> > ### Author Response · Authors · 2024-11-22
> > **Response to Reviewer Ys2B (Part IV)**
> >
> > >> **You want μ primarily to be able to distinguish between actions of π. Would it help if you ran multiple verifiers at the same time**
> >
> > Yes this is a great idea and a promising direction of future work. We can potentially use advantages of multiple prover policies to design potential functions that are most helpful to improve the base policy.
> >
> > >> **Maximum reward in Section 3.3 (didactic setting)**
> >
> > Yes, you are correct. The reward is binary, so the maximum would simply be $1$. We have made this clear in the submission.
> >
> > >>**Notation in Appendix F**
> >
> > Thank you for pointing out the typos. We have made the following updates to the submission. 1) We have added a mathematical definition of $d^\pi_h$; 2) We have added more steps between Equation 26 and 27 to explain the arguments in the proof more clearly; and 3) we have fixed the other typos. Thank you for the suggestion!

---

> > > ### Comment · Reviewer_Ys2B · 2024-11-22
> > > **Response**
> > >
> > > I thank the authors for their detailed response to the review, and addressing most of my concerns in a satisfactory way.
> > >
> > > Regarding some specific points remaining:
> > >  - I am still wary of the importance of the optimisation over $\alpha$. Even though this specific experiment showed some stability over its values, the range is still quite small (half an order of magnitude), and the experiment itself is quite limited (only two base model versions from the same family, one task etc.). In comparison, you also mention tuning $\alpha$ in test-time experiment, where you use a much smaller range (which is also a choice to make!), and consequently, an order-of-magnitude smaller $\alpha$.
> > >  - Since you only explain your notation for $\gtrsim$ in the appendix (which is, I think, non-standard - the closest is possibly the big-O notation), Theorem 3.1 could be more clearly stated by explicitly pointing an existence of the constant $c$. It could also use a sentence of explanation for what happens when the bound is negative (basically what you wrote in the response above).
> > >  - I agree with your characterisation of the paper as the exploration of using PAV, and not necessarily studying the best way of performing exploration - which limits its impact somewhat (as I pointed out above, this possibly stems from taking an unnecessarily strong view on only using greedy beam search). I liked the paper overall, but I think, based on this, all the other points raised in this discussion with me and with other reviewers, that my current (positive) rating fairly reflects the paper's contribution.

---

> ### Author Response · Authors · 2024-11-26
>
> Thank you for your response. To address the questions above, we add a new experiment where we aim to explicitly evaluate the exploration capabilities of PAV. Note that our goal is not to claim that PAVs are the best approach to exploration, but instead to compare PAVs to an existing exploration method to understand if building on our approach of connecting process rewards with exploration and potential functions, and designing novel forms of process rewards could be a fruitful endeavor for future research on exploration in LLMs.
>
> Specifically, we compare test-time beam search guided by process supervision from PAVs with the importance weighted search approach outlined in AlphaLLM [1], an approach that runs MCTS for search. We use the heuristic from AlphaLLM and our preliminary results show that PAVs are 8x more compute efficient at test-time beam search, as outlined below. We also respond to your other questions below.  **Please let us know if this addresses your remaining concerns, and if so, we would be grateful if you are willing to raise your score.**
>
> >> **Are PAVs comparable with other strategies for exploration?**
>
> **Background on Alpha LLM**: For the importance weighted search approach of AlphaLLM, we implement beam search in the following way. At any given point, the beam consists of $N$ states $s_1, s_2, s_3, \ldots, s_N$. These are partially unrolled solutions from the base policy $\pi$, up until state $s_i$ (prefix). We then expand each node in the beam 3 times, by conditionally sampling from $\pi$ (conditioned on each of the states in the beam). We get $N \times 3$ states, which we rank with the following scoring function and then select the top $N$ state. Each of the new expanded states is of the form $s, a$, where $s$ is the previous state in the beam and $a \sim \pi(\cdot \mid s)$ is the new sampled action (step). Following, Section 4.3 from Ye Tian et. al. (2024), the score for the new state $(s, a)$ is $Q^\pi(s, a)  + C \cdot U(s)$ where $U(s)$ is the uncertainty bonus for the state $s$, which is computed as $U(s) = \sqrt{ \frac{n(s)}{\sum_{i=1}^{N} n(s_i)}}$. We use $C=0.25$, which we identified by tuning performance over a hold out validation set we use for PAVs as well. This resembles UCB or UCT-style exploration. Concretely, $n(s)$ is the effective number of children for node $s$ (Section 4.3.2 in Ye Tian et. al. (2024)). The term $n(s) = C^\prime \cdot I(s)$ is computed by linearly scaling the importance $I(s)$ defined as $I(s) = \max_a |V^\pi(s) - Q^\pi(s,a)|$, where $a$ is one of the $3$ actions sampled from state $s$ when expanding the beam. We tune and set $C^\prime=2.0$. Intuitively, AlphaLLM chooses to explore states that can change the $Q$-values by a lot, when we continue to sample from them. When the $Q$-values deviate by a lot, the $I(s)$ term increases. Consequently, so does $N(s)$ (effective children count) and $U(s)$ (uncertainty bonus).
>
>
> **In Appendix K (Figure 18), we show our results for the experiment of test-time beam search with PAVs, vs. the UCB style metric for exploration in Alpha LLMs. Since we only had two days to implement and run experiments for AlphaLLM (with no code base available), our findings are preliminary**. Nevertheless, we find that PAVs are 8x more compute efficient than AlphaLLM at test-time exploration for the discovery of the correct solution trace. This is likely because the exploration metric in AlphaLLM uses the absolute magnitude of the advantage under the base policy, vs. PAVs which use the signed advantage under the prover policy. Thus, our exploration metric prefers steps that increase the likelihood of a complementary prover to discover the correct solution, as opposed to preferring steps that simply change the previous state’s value function  (under the base policy) by the largest magnitude (which can also be negative).
>
> [1] Tian, Y., Peng, B., Song, L., Jin, L., Yu, D., Mi, H., & Yu, D. (2024). Toward Self-Improvement of LLMs via Imagination, Searching, and Criticizing. arXiv preprint arXiv:2404.12253.

---

> > ### Author Response · Authors · 2024-11-26
> >
> > >> **Optimization over $\alpha$**.
> >
> > **We have added the following discussion to Appendix E**, that explains how one can tune the hyper-parameter $\alpha$ by sweeping over a reasonable range of ``good’’ $\alpha$ values, identified through binary search.
> >
> > In practice, we arrive at the ranges for searching over the hyper-parameter $\alpha$ through a systematic procedure on a hold-out validation set. This procedure can be potentially repeated for any new problem instance where PAVs need to be used. In particular, we tune $\alpha$ with a two layer search strategy where the outer layer search is coarse-grained and used to identify a good high-level range of $\alpha$ values such that the performance of PAVs is comparable to ORMs (i.e., PAVs don’t yield degenerate solutions). The inner layer of search is more fine-grained and used to tune the performance of PAVs even more. Note that both these layers of search are carried out on a small hold out validation set. Additionally, as we state below, the outer level of coarse-grained search is already good enough for PAVs to outperform ORMs on the MATH benchmark, for both beam search and online RL.
> >
> > For both test-time search and RL we identified a ‘’good’’ range of $\alpha$ by running binary search. Since $\alpha > 0$, we start with a high value of $\alpha = 10.$, and then keep reducing it by half ($10 \rightarrow 5 \rightarrow 2 \rightarrow 1$) until we see a run of test-time beam search or online RL that yields a non-trivial performance improvement over only using outcome rewards. During this outer level binary search, we stopped at $\alpha=5.0$ for online RL and $\alpha=1.0$ for test-time search. Once we identified the range, we run a second level of fine-grained search search, i.e., we search linearly between $\alpha \in [0.0, 1.0]$ in intervals of $0.1$ for using PAVs at test-time, and  $\alpha \in [0.5, 6.0]$ in intervals of $0.5$ for using PAVs as process rewards in online RL. As we state in the paper, the choice of $\alpha$ within this range is quite robust. Any $\alpha \in (0.1, 0.7)$ for test-time search and $\alpha \in (1.5, 5.5)$ for online RL puts PAVs in a regime where the performance of PAVs for either is better than only using outcome rewards.
> >
> > >> **Fixing the ‘≳’ notation used in Theorem 3.1 and pointing to the existence of a constant more explicitly**.
> >
> > Thank you for this suggestion! We have updated the submission to avoid using the notations ‘≳’, $\Omega(\cdot)$, and $\mathcal{O}(\cdot)$ in Theorem 3.1 (Section 4.2). Instead, we state the lower bound in Theorem 3.1 in terms of a universal positive constant $C > 0$, and prove that such a $C$ exists in the proof of Theorem 3.1 in Appendix F.

---

> > > ### Author Response · Authors · 2024-11-30
> > >
> > > Dear Reviewer,
> > >
> > > We apologize for bothering you, but since there are only two days left in the extended discussion period, we wanted to check in with you to see if our rebuttal addresses all outstanding concerns and have a chance to address any new ones.
> > >
> > > Authors

---

> > > > ### Comment · Reviewer_Ys2B · 2024-12-02
> > > > **Response**
> > > >
> > > > I appreciate authors' continued effort and engagement. Indeed, it does look like the preliminary results vs AlphaLLM seem promising, although more evaluation is certainly needed to assert that PAV method is generally superior in encouraging exploration.   I think this is possibly the most interesting future direction. However, I still see the impact and novelty of the paper (as well as the relatively narrow experimental suite, as pointed above) as a bit too low to raise my score to 10, so I leave my current rating.

---

### Official Review · Reviewer_11y4 · 2024-11-04

**Soundness:** 3
**Presentation:** 3
**Contribution:** 3
**Rating:** 8
**Confidence:** 1

**Summary:**

This work addresses the design of process reward models (PRMs) in the context of online reinforcement learning with an LLM base policy. The key problem the authors presents is that existing PRM approaches rewards only Q-score - the expected accuracy given a current state and a next action. They identify that absolute Q-score in beam search favors actions that might have high Q-score but might have little or negative improvement from the current state. Instead, they propose to use Process Advantage Verifiers (PAV) using the difference or process advantages between Q-values of a next action and the previous action to identify, using a separate prover policy to compute these process advantages rather than a base policy. They provide intuition and theory that PAV performs well with a prover policy that is not too misaligned with the base policy but can discriminate well between different actions. Their experimental results show significant accuracy and efficiency gains over both Outcome Reward Models (ORM) and other PRM baselines. The work additionally demonstrates accuracy and efficiency gains in the online RL setting.

**Strengths:**

* Methodology is mostly clear and well-elaborated on motivations by walking through didactic example in section 3.3 and intuition behind prover policy selection in 3.4.

* Results in experiments is well-outlined in sections 4 and 5, and shows significant improvement over ORM and PRM baselines.

* Idea of using a separate "prover" policy to compute advantage is interesting and novel, including the discussion on the choice of complementary policy.

**Weaknesses:**

* This work has some issues in communicating and emphasizing important aspects of using a separate prover policy, which is a key part to distinguish from Shao et al. 2024.

* I'm concerned by the lack of comparison with other PRM baselines. The work notes there are several competing PRM approaches (Appendix A), but section 4 seems to only compare with only one.

**Questions:**

* As the use of a separate prover is a key part of the work's novelty, it does not seem appropriate that a key result Proposition F.1 is relegated to the Appendix, along with the explicit characterizations of what is a "complementary" policy.

* Authors show many negative qualitative examples of inappropriate prover policies in Figure 2 and Appendix G, but do not seem to provide any positive examples of good prover policy results.

* The introduction of a prover policy $\mu$ in equation (2) of section 3.1 seems slightly confusing. The initial construction of $A$ uses the base policy $\pi$ throughout and only later notes that $A$ can be computed under any policy. This section would flow better if these were differently ordered or reworded.

* Why was (Snell et al. 2024) chosen for a representative PRM baseline in Figure 5a)?

Minor things:
* There appears to be a typo and inconsistent notation in Appendix B as the 15-word vocabulary set only has 14 elements. The example in Figure 10 shows this set should be 0-indexed but the set in Section 3.3 and Appendix B are 1-indexed.

* Figure 5b is difficult to read due to similar colors used (Similar shades of red/orange).

---

> ### Author Response · Authors · 2024-11-19
> **Response to Reviewer 11y4 (Part I)**
>
> Thank you for the review and for a positive assessment of our paper! To address your concerns, we include experiments for RL training with PRM baselines that we used for test-time search, and make the comparison with prior works on automated PRMs (Wang et. al. (2024), Luo et. al. (2024), Snell et. al. (2024)) for test-time search more explicit. We also provide qualitative examples for process rewards from good vs. bad prover policies, and respond to your other questions and concerns below. **Please let us know if your concerns are addressed, and if so, we would be grateful if you are willing to raise your score.**
>
>
> >> **Discussion on “Why do we use a separate prover policy?”**.
>
> As you correctly note, the fact that we use a separate prover policy is indeed a key point of distinction with prior work Shao et. al. (2024).  In L233-249 of our submission (Section 3.2), we explain in detail why we need a prover policy $\mu$, different from the base policy $\pi$ we optimize. In  a nutshell, there are two main reasons for this choice: 1) as we note in L244 of our submission, training with process rewards where $\pi =  \mu$ during RL ***mathematically*** leads to gradients that are equivalent to those observed when purely optimizing the outcome reward ($\ell_{ORM-RL}$); and 2) when the outcome rewards under a poor base policy are very sparse, $Q^\pi \approx 0$, on most states, and consequently $A^\pi \approx 0$. Thus, we need a separate prover that can distinguish steps generated by even a poor base policy. In particular, we find “complementary” provers that both distinguish actions taken by the base policy, and are not too misaligned with it, lead to largest improvements in the base policy. We show this theoretically in Theorem 3.1. We have updated the discussion in this section to more clearly signpost this argument, early on in the paper.
>
> >>  **Introduction of separate prover policies in Section 3.2.**
>
> Thanks for the question! We have updated the paper to now clarify that the goal of Section 3.1 is primarily to motivate per-step advantages over Q-functions, with the search example in mind. The very same arguments in this section also apply to a different prover policy, but we stuck with the same policy in Section 3.1 due to the clarity of explaining one concept (Q-values vs advantages) before the next idea of separate prover policies comes through.
>
> **Flow of Section 3**: In the beginning of Section 3, we define process rewards as potential-shaped step-level rewards that, when optimized during RL or test-time search, should yield better performance as measured by the outcome rewards. With this intuition, we explored a number of choices for the design of process rewards.  We begin with Q-values, and explain with an illustrative example in Figure 2, why Q-value serves as a poor choice of potential functions, compared to advantages.
> Now, given this choice, the next question is "what policy should advantage be measured under?". To answer this question, we come back to the notion of potential functions and demonstrate a certain set of prover policies $\mu$, distinct from the base policy $\pi$ is better for this use case. In Section 3.2, to expand the set of possible potential functions that conform to the definition of potential functions in Ng et. al. (1999), we allow the advantages to be computed under a broader choice of prover policies. Following this section, in the rest of the submission, we also allow the prover policy $\mu$ to be different from the base policy $\pi$.
>
> If there is a particular part of the submission where this choice is particularly unclear, please let us know and we would be happy to edit the submission to make this more clear.

---

> > ### Author Response · Authors · 2024-11-19
> > **Response to Reviewer 11y4 (Part II)**
> >
> > >> **Comparison with other PRM baselines and comparisons with Snell et al. in Figure 5a.**
> >
> > Several works on automated PRMs, use the Q-function as the step-level reward (Wang et. al. (2024), Luo et. al. (2024), Snell et. al. (2024)). Each of these use the step-level rewards in different ways at test-time, and we compare PAVs with all of them in Section 4. In Figure 5a, we compare the performance of PAVs with $Q^\pi$ for test-time beam search (see the line for “PRM $Q^\pi$”), as done in Snell et. al. (2024). In the same figure, the line for “PAV-as-ORM” corresponds to using the trained PAVs as outcome reward models, i.e., to score the full sequence. The sequence is scored by computing the minimum over the step-level scores from the PAV model. This is similar to how Wang et. al. (2024) and Luo et al. (2024) use their trained PRMs during test-time search. For a more direct comparison, we run exactly the same procedure for test-time best-of-$N$ search with PRM $Q^\pi$ as done in  Wang et. al. (2024) and Luo et al. (2024). **We have explained these comparisons with a new plot (Figure 13) in Appendix C.**
> >
> >
> > **We also add a new experiment (Figure 17) where we use the PRMs proposed in prior works**  (Wang et. al. (2024), Luo et. al. (2024), Snell et. al. (2024)) as step-level rewards in online RL.** Here, the PRMs are trained offline and fixed, and then used to assign scores to intermediate steps in trajectories sampled during online RL. For this, we use $Q^\pi_{\mathrm{base}}$ as the step-level score during RL training initialized with the base policy $\pi_{\mathrm{base}}$ since $Q^\pi_{\mathrm{base}}$ is the PRM proposed in prior works. This step-level reward is used to reward trajectories sampled during online RL training,  in addition to the outcome reward, similar to our effective reward  in Equation 5. We find that the test accuracy drops quite quickly, even though the train rewards keep improving, since the policy just learns to hack the step-level Q-values, for the reasons we explain in L240-245. We see a similar collapse when we use $Q^\mu$ instead of $A^\mu$ (see Appendix G for qualitative examples).
> > On the other hand, if we were to use the Q-values or advantages from the current policy iterate $\pi_t$ as the step-level reward, then that is equivalent to only optimizing the outcome reward, and the only benefit of using the step rewards would be to reduce variance during online RL iterations. Thus, for online RL as well, our proposed step-level rewards (PAVs) which uses advantages of the prover policy $A^\mu$ outperforms baselines that plugin PRMs proposed in prior works on test-time search, **We have added this new experiment on online RL where step-level rewards are given by PRM  $Q^\pi_{\mathrm{base}}$ proposed in prior works  to Appendix E, Figure 17**.
> >
> >
> > Finally, Lightman et. al. (2023) collected human labeled training data for generations from GPT-4, in our initial experiments, we found that the PRM trained on this data performs poorly on rollouts from the Gemma model family. This finding is consistent given the results from Snell et al. 2024 which also finds training on PRM800K dataset from Lightman et al, to be primarily ineffective in the context of PALM-2-S* models. Given this distribution shift, we only compare PAVs, with other works on automated process verifiers like Wang et. al. (2024), Luo et. al. (2024),  Snell et. al. (2024) and Shao et. al. (2024) as discussed above.
> >
> >
> > >> **Result in Proposition F.1 vs. the result in Theorem 3.1**
> >
> > We moved Proposition F.1 to the Appendix mainly because it is almost directly implied by our main result in Theorem 3.1, that we choose to highlight over Proposition F.1. This is because Theorem 3.1 directly characterizes the set of ''complementary prover'' policies. These are the policies that have a high positive value for the right hand side of Equation 6, i.e. are able to distinguish steps taken by the base policy (high $Var_{\pi} A^{\mu}(s, a) $), without being too misaligned with it (high $E_{\pi} [A^{\mu}(s, a) A^{\pi}(s, a)]$). As noted by our policy improvement result in Theorem 3.1, when we update the base policy using the natural policy gradient where the step-level rewards are from complementary prover policies, we are guaranteed a greater improvement in the base policy. On the other hand, Proposition F.1 simply connects a lower bound on the absolute performance difference between base and prover policies $|V^{\pi_t} - V^\mu|$ to a proportional improvement in the base policy $V^{\pi_{t+1}} - V^{\pi_t}$, whenever the prover policy is sufficiently complementary.
> >
> > If you think that it is still better to move Proposition F.1 to the main paper, and have some suggestions on other parts we can move to the Appendix in order to make space, we would be happy to do that.

---

> > > ### Author Response · Authors · 2024-11-19
> > > **Response to Reviewer 11y4 (Part III)**
> > >
> > > >> **Qualitative examples of good prover policies**.
> > >
> > > In Section 3.4, we formally characterize the set of good prover policies that are complementary, i.e., whose advantages strongly distinguish states sampled by the base policy $\pi$ without being misaligned with the base. We discuss how some provers in the class of best-of-K policies over $\pi$, i.e., $\mathrm{BoK}(\pi))$ can be good complementary provers (Remark 3.1). We verify this claim empirically in Section 4. In particular, we see in Figure 5b that when we use the  $\mathrm{Bo}4(\pi)$ policy as a prover policy in PAVs, we observe the best performance during test-time beam search.
> > >
> > > **In Figure 11 of Appendix B, we add heatmaps of step-level rewards** obtained by using prover policies of different strengths in our didactic setup. We see that for the weaker prover, the advantage magnitudes are much higher, since the weaker prover policy is more likely to complete a partially correct generation over an incorrect one, i.e., a generation where the last set of tokens in the prefix partially matches the planted sub-sequence, vs. one where it does not. On the other hand, when the strength of the prover policy increases, the magnitude of token-level rewards reduces ($A^\mu \approx 0$) since it can complete the solution with nearly equal likelihood from both correct and incorrect prefixes sampled from the base policy. Thus, RL training with the effective rewards  (Equation 5) computed using strong provers performs similarly to RL training with only outcome rewards, as noted in Figure 3b.
> > >
> > >
> > > >> **Vocabulary set for the didactic example.**
> > >
> > > Yes, you are correct! In the didactic example, the vocabulary set is 0-indexed, and there are 15 elements in total. We have updated this typo in the paper.
> > >
> > > >> **Color coding in Figure 5b.**
> > >
> > > Thank you for the suggestion. We have updated the figure Figure 5b with a different color coding for 5b.

---

> > > > ### Author Response · Authors · 2024-11-24
> > > >
> > > > Dear Reviewer,
> > > >
> > > > We apologize for bothering you, but since there are only two days left in the discussion period, we wanted to check in with you to see if our rebuttal addresses all outstanding concerns and have a chance to address any new ones.
> > > >
> > > > Thanks,
> > > > Authors

---

> ### Comment · Reviewer_11y4 · 2024-11-25
>
> I thank the authors for their detailed response to my review, and for addressing or clarifying most of my concerns satisfactorily. I have raised my overall score as well as presentation score in response.

---

### Official Review · Reviewer_CgRb · 2024-11-04

**Soundness:** 3
**Presentation:** 3
**Contribution:** 3
**Rating:** 6
**Confidence:** 3

**Summary:**

The paper proposes a method for improving reasoning abilities of large language models on math problems. Given a math problem, the model generates an answer autoregressively. The reasoning steps can be considered actions in a search space. The method is based on the advantage function as a (shaped) reward, and then uses RL to further train the base policy.
The results are demonstrated on the MATH dataset where the final answer is a numerical one, which makes the validation of the outcome relatively straightforward.

**Strengths:**

The idea is clear, it has appropriate theoretical foundations, and the authors empirically demonstrate the advantage of the proposed approach.

**Weaknesses:**

It is not clear whether they propose this as training or test time improvement. More precisely, I assume that both, but at certain places it is not evident which one the paper is talking about.
I am not convinced by the argument that the proposed effective reward scheme improves exploration. I do not think this is properly motivated either intuitively, theoretically or empirically. Also, the suggestion of the improved exploration is not related to checking correctness and relevance.
In the second paragraph of Section 3, around lines 157-159, the authors reference the potential functions from Ng et al. (1999), but the description is vague, and it does not seem to match the original definition.
The language should be reviewed, including repeated words.

**Questions:**

In Equation (5), the effective reward is added to the return in the formula for the gradient. What is the reason of this apparent contradiction?
The authors argue that a strong model cannot be used as a prover because a strong policy can reach the answer from any state. I think this is only true if the value function does not take the answer length into account, i.e., it is not discounted and there is no per-step cost. An interesting follow-up would be to re-evaluate, for example, Figure 3b with value functions including a per-step cost.
Which example does line 197 ("based on the example") refer to?
Line 267 states that RL with effective reward achieves a tenfold improvement in sample efficiency, but I cannot see this is Figure 3a. What is this improvement based on?
In some of the figures, there is a shaded area. It should be specified what they mean (std deviation? confidence intervals?) and what sample size they are based on.

---

> ### Author Response · Authors · 2024-11-19
> **Response to Reviewer CgRb (Part I)**
>
> Thank you for the review and for a positive assessment of our work! To address your concerns, we make several clarifications in the paper regarding why PAVs improve exploration; we add new  experiments with discounted process rewards from strong provers; and clarify the relation to potential functions. We also respond to your other questions and concerns below, and have modified the paper to include discussion on potential functions.  **Please let us know if your concerns are addressed, and if so, we would be grateful if you are willing to raise your score.** We would be very happy to discuss further.
>
> >> **Are PAVs used for test-time or training time improvement?**
>
> Once we train PAVs, we use them in two different ways: 1) to rank intermediate generations during test-time beam search, where we see that PAVs improve the compute-efficiency of test-time search (Section 4); and 2) as step-level process rewards when training the base policies with online RL, where we find that PAVs improve the sample efficiency of online RL (Section 5). We have made this more clear in the section headings to avoid misunderstanding as to which section refers to train usage vs test usage.
>
> >> **Why does the proposed effective reward scheme improve exploration?**
>
> This is a great question! We view ''exploration'' as a scheme that enables the discovery of high-outcome reward solutions. Under this definition, exploration bonuses are not required strictly to be count-based quantities like entropy or pseudocounts, but rather in our case they correspond to advantages under some prover policy. To empirically show that PAVs improve exploration (i.e., speed up discovery of high-reward solutions), in Figure 6, we provide empirical evidence for why our process rewards (PAVs) enable exploration during online RL training, with discussion in L421-448 of our submission. Specifically, we define the set of hard questions as those that remain unsolved even after running a best-of-256 search over 256 independent samples from the SFT model, and using the most accurate outcome verifier (i.e., we check if the final answer matches). We then check how many of these are solved by the policy trained with PAVs, if we run the best-of-N search using N<256 samples. Compared to a policy trained with ORMs, the PAV trained policy is able to solve 4-5x more fraction of the hard questions.
>
> Essentially, the additive $A^\mu$ term in the effective reward serves as a bonus to increase the likelihood of steps that induce ''progress'' under the prover policy,. i.e., improve the chances of the prover policy to discover the correct answer. Thus, even though a trajectory sampled from the base policy fails to reach the correct answer, some steps in the trajectory still end up being up-weighted, which would not be the case without the additive term $A^\mu$ (when we only use outcome rewards). This results in improved coverage over solutions at test-time and aids the policy in finding a good sequence without committing to a myopic set of prefixes early on in beam search or RL. Corroborating this intuition is the better best-of-N performance for the policy trained with our process rewards (PAVs), compared to ORMs (Figure 6). For test-time beam search, advantages under the prover policy promote the coverage of solution traces in the beam which aligns better with the canonical definition of exploration; please see the discussion in L177-187 of our submission which explains this with an illustration (Figure 2). Hence, we say that PAVs improve exploration – if you think a different terminology will be better here, please do let us know and we would be happy to make a change in the paper.

---

> > ### Author Response · Authors · 2024-11-19
> > **Response to Reviewer CgRb (Part II)**
> >
> > >> **Connecting process rewards to potential functions in L157-159**.
> >
> > We might be missing something but to our understanding, the effective reward we define in Equation 5 **matches** the definition of a potential-based shaped reward, as defined in Ng et. al. (1999). **We have added the above discussion to Appendix I of the submission. Please let us know if this addresses your concerns.**
> >
> > Before we explain why this is the case, we provide some background below.
> >
> > **Background on potential functions**: In Ng et. al. (1999), instead of learning a policy for a reward function $R$, the policy is trained to optimize the transformed reward $R + F$. Here, $F(s_h, a_h, s_{h+1})$ is a reward shaping function, that takes as input the the current state $s_h$, action taken by the policy $a_h$, and next state $s_{h+1}$, and maps this to a scalar reward. In particular, they show that when the shaping function $F$ is potential-based, i.e., it is of the form $F(s_h, a_h, s_{h+1}) = \Phi(s_{h+1}) - \Phi(s_h)$ for some state-dependent potential function $\Phi$, then the policy that optimizes the transformed reward $R + F$, also optimizes the original reward $F$.
> >
> > In Equation 5, our effective reward function is: $Q^\pi(s_h, a_h) + \alpha A^\mu(s_h, a_h)$. This reward function matches the functional form of the transformed reward function $R + F$ defined in Ng. et. al. (1999). Here, $Q^\pi$ corresponds to the reward $R$, and $\alpha A^\mu(s_h, a_h)$ is the potential-based reward function $F$. To see why $F$ satisfies this definition we use the definition of advantages from Equation 2. We can write $\alpha A^\mu(s_h, a_h)$ as $\alpha (Q^\mu(s_h, a_h) - V^\mu(s_h)) = \alpha (V^\mu(s_{h+1}) - V^\mu(s_h)) $. Following the theoretical result on potential-based reward shaping functions in Ng. et. al. (1999), the optimal policy under our effective reward $Q^\pi + \alpha A^\mu$ also lies in the set of optimal policies which only optimize the outcome reward (final answer correctness, or $Q^\pi$).
> >
> >
> > >> **Using discounting rewards from strong provers**
> >
> > This is a great suggestion! **We add a new result (Figure 16) and discussion on using discounted rewards from provers to Appendix E of the submission**. Here, we train PAVs to predict the advantages of discounted rewards from strong prover policies. Here, for the problem $\mathbf{x}$, and the state, action pair $s, a$, the process rewards are given by the effective reward from Equation 5: $Q^\pi + \alpha A^\mu$, except that the advantage $A^\mu$ is the difference in discounted rewards, i.e.:
> >
> > $A^\mu(s, a)  =  E_{y \sim \mu(\cdot \mid s, a)} \left[ \lambda^{\mathrm{len}(y)-\mathrm{len}(s)-1} \mathrm{Rex}(y, y^\star_{x})\right]  - E_{y \sim \mu(\cdot \mid s)} \left[  \lambda^{\mathrm{len}(y)-\mathrm{len}(s)} \mathrm{Rex}(y, y^\star_{x})\right],$
> >
> > where the prover policy samples solution $y$ with $\mathrm{len}(y)$ steps to complete the solution, from a state $s$, which already has $\mathrm{len}(s)$ steps in it.
> >
> > For this setting, we train a verifier to predict discounted rewards for the Gemma 9B prover policy. We find that the discounted process rewards from the stronger 9B prover policy performs worse than undiscounted rewards from the weaker Gemma 2B prover policy, when using either to train the 2B base policy with online RL.
> >
> > The main reason for the discounted rewards to not enable the use of strong provers is because strong prover policies tend to disregard states generated by the base policy (as illustrated in Figure 2b). This means, that irrespective of whether the weak prover policy generates a partially correct or incorrect solution, when we rollout the strong prover policy from this state generated by the base policy, the strong prover directly attempts to answer the math problem with its own solution trace. Thus, from any state the strong prover is expected to complete the solution with roughly the same number of steps. This means that $A^\mu \approx 0$ even in the discounted case, which reduces the ability of the strong prover policy to distinguish steps taken by the base policy.

---

> > > ### Author Response · Authors · 2024-11-19
> > > **Response to Reviewer CgRb (Part III)**
> > >
> > > >> **Example referred to in L197 of our submission**
> > >
> > > The example in L197 refers to Figure 2a of our paper, which illustrates why advantages enable more explorations when used as process rewards, compared to Q-values (see discussion in L177-182 of our submission). We have fixed this reference.
> > >
> > > >> **L267: 10x sample-efficiency in the didactic setup.**
> > >
> > > We apologize for not clarifying this in the original paper and have updated the paper to add this discussion now. When we run RL training for 10k iterations with only the outcome rewards, the policy is able to learn the planted sub-sequence and achieve a reward of 1.0. The policy trained with effective rewards is able to do this in $<$1k iterations. Thus, we concluded a 10x sample efficiency gain. We have added this point to Section 3.3.
> > >
> > > >> **Shaded area and confidence intervals in the figures.**
> > >
> > > This refers to a 95% confidence interval over the true mean of the reported metric. For Figures 8,9, and 15 this is computed using 500 IID examples. In Figure 5, we also use $5$ independent runs of the search algorithms to compute confidence intervals that additionally account for the randomness in the search procedures. We have clarified this in Appendix C, E.

---

> > > > ### Author Response · Authors · 2024-11-24
> > > >
> > > > Dear Reviewer,
> > > >
> > > > We apologize for bothering you, but since there are only two days left in the discussion period, we wanted to check in with you to see if our rebuttal addresses all outstanding concerns and have a chance to address any new ones.
> > > >
> > > > Thanks,
> > > > Authors

---

> ### Comment · Reviewer_CgRb · 2024-11-27
>
> I thank the authors for their detailed response to the review. I have increased the presentation score.
>
> Some further comments:
>
> **On the question of exploration**
>
> I don't think their definition of "exploration" is the one commonly used. This is an issue of intuition: their definition is essentially the general RL goal (finding high outcome reward solutions), so saying that "their method works well because it explores well" is essentially saying "it works well because it works well".
>
> This distinction is important (only) in the explanation of the intuition behind the proposed method. For example, a question is: Why does having a better model act as a prover / reward helper better than simply training the prover model further? I think this also makes the discussion in Section 5, Result 3 a bit imprecise.
>
> (The authors argue in line 346 that since a weak prover can also improve the base policy, so their method is not something similar to knowledge distillation. I can accept this argument.)
>
> Based on the beginning of Section 3.1, I would say what the authors really do during beam-search is exploration in action space. More precisely, using advantages and not Q values results in searching for actions that are of high value, independently of the states.
>
> This is a unique feature of this environment: the action space is really large, and states are used to a large degree as a sort of stepping stone for the next action. (This is in some ways the opposite of planning: instead of thinking ahead, try to come up with a thought that will help form the next thought.)
>
> *Minor comments:*
> * Takeaways in lines 452-453: The Best-of-K approach is highlighted here, but it is not mentioned with regards to exploration in the section of the main text.
> * Takeaways in lines 251-252: "better explore-exploit tradeoff ... and online RL": in the preceding text, exploration is only mentioned in the context of beam search, not that of training (and I presume online RL refers to the training phase).
> * Does this method improve exploration in general, or does it make the base policy explore in the directions the prover considers good?
>
> **On the questions of potential functions**
>
> Let me rephrase my original question. In line 158, the authors say that in Section 3.1, they evaluate two different choices of potential functions. This leads the reader to believe that, since the two choices in Section 3.1 are $Q$ vs $A$, they mean that their *reward scheme* is analogous to potential functions. (See also lines 196-197: "we view process rewards as potential functions".)
>
> My understanding, however, is that $Q$ is a potential function, and $A$ is (or can be) a *reward scheme based on a potential function* ($Q$).
>
> **On the question of discounting**
>
> What I understand from their reasoning is that with strong provers, they usually throw the base policy partial solution out of the window as a first step, regardless of discounting. I can accept this.
>
> **On the question of "example" in L197**
>
> Minor comment: "based on the example *in* Figure 2a" (in is missing)
>
> **On the question of confidence intervals**
>
> Thanks for the clarification. It seems as though that no multiple training seeds were used for the figures.

---

> ### Author Response · Authors · 2024-11-27
>
> Thank you for responding to us! We have updated the submission to address the minor concerns (and are happy to adjust any wording if there are remaining concerns even after the paper update deadline passes). We clarify the latest comments below; for the remainder of the points that you accept in the response above – they sound great to us!  **Please let us know if your concerns are addressed, and if so, we would be grateful if you are willing to raise your score.**
>
>
> >> **Clarification on the definition of exploration**
>
> You are right that there are several definitions [1, 2, 3, 4] of exploration in RL. For instance, in more theoretical work on bandits [3, 4], exploration is defined as the algorithm that helps discovery of optimal actions that maximize a given reward function, and its efficacy is measured in terms of cumulative regret or sample efficiency of learning; in unsupervised RL  [1], exploration is typically defined as maximizing coverage or diversity of states since a target reward function is not known; in meta RL [2], exploration often refers to identifying the underlying MDP so that optimal policies can then be found.
>
> In this paper, we used the term “exploration” to refer to the first definition – a procedure that helps discover the optimal action as quickly as possible. With this in mind, we justify the phrase “PAVs enable better exploration” by showing that PAVs improve sample efficiency (i.e., reduces cumulative regret) for RL, and enables finding optimal actions via test-time search. PAV advantages are akin to exploration bonuses that one could add on top of a standard greedy search that is run with respect to the policy’s Q-function. We are happy to use a different term for exploration if you would suggest, but we hope that the above explanation clarifies the meaning we referred to, for avoiding any misunderstanding.
>
> [1] Jin, C., Krishnamurthy, A., Simchowitz, M., & Yu, T. (2020, November). Reward-free exploration for reinforcement learning. In International Conference on Machine Learning (pp. 4870-4879). PMLR.
>
> [2] Gupta, A., Mendonca, R., Liu, Y., Abbeel, P., & Levine, S. (2018). Meta-reinforcement learning of structured exploration strategies. Advances in neural information processing systems, 31.
>
> [3] Auer, P. "Finite-time Analysis of the Multiarmed Bandit Problem." (2002).
>
> [4] Abbasi-Yadkori, Y., Pál, D., & Szepesvári, C. (2011). Improved algorithms for linear stochastic bandits. Advances in neural information processing systems, 24.
>
>
> >> **Why does having a better model act as a prover / reward helper better than simply training the prover model further?**
>
> While it is correct that if the prover policy is really good, we could have it serve as the base policy and improve it further via some sort of RL (either via outcome reward maximization or by using some process reward model), our conceptual insights and practical results show that improving a strong policy further would benefit from PAV rewards in turn computed using a new weak prover policy. Specifically, on one hand, we see that using Gemma 9B as the prover policy for test-time beam search over samples from the Gemma 2B base policy worked best (Figure 5c). At the same time, note that to train the stronger Gemma 9B policy with dense rewards during online RL, we used the weaker Gemma 2B prover policy (Figure 7c, Figure 19) to get good results – implying that weak policies can still serve as good provers for improving strong policies further. Does this address the question?
>
> >> **Based on the beginning of Section 3.1, I would say what the authors really do during beam-search is exploration in action space. More precisely, using advantages and not Q values results in searching for actions that are of high value, independently of the states. This is a unique feature of this environment.**
>
> Yes, this is correct! We do not claim that our approach is a general method for exploration, but that using advantages as exploration bonuses helps ``explore’’ actions that are of high value under the prover policy, and as you correctly note, independent of the previous states. We empirically find that this form of exploration is helpful for LLM math reasoning problems but some other forms of process rewards / exploration bonuses might work for other settings.

---

> ### Author Response · Authors · 2024-11-27
>
> >> **My understanding, however, is that Q is a potential function, and A is (or can be) a reward scheme based on a potential function (Q).**
>
> Thanks for pointing this out! Our language here was indeed a bit confusing: we have now edited the paper to indicate that the potential function is $Q^\mu$, and that $A^\mu$ function we study is the effective reward shaping term (i.e., $\Psi(s_{h+1}) - \Psi(s_h)$ in Ng et al. 1999), where $\Psi(s_h) = Q^\mu(s_{h-1}, a_h)$ is the potential function. We have also clarified that we do not evaluate two potential functions, but compare two types of process rewards / reward shaping terms. We have made this clear in Section 3 and in Appendix I where we discuss the connection to potential-based reward-shaping terms in detail.
>
> >> **Does this method improve exploration in general, or does it make the base policy explore in the directions the prover considers good?**
>
> Note that our goal is not to claim that PAVs are the best approach for exploration, but instead to compare PAVs to an existing exploration method and understand if building on our approach of connecting process rewards with exploration and potential functions, and designing novel forms of process rewards could be a fruitful endeavor for future research on exploration in LLMs.
>
> At the same time, to answer your concern, we would like to point to a new experiment we run to address Reviewer Ys2B’s comments. Here, we compare test-time beam search guided by process supervision from PAVs with the importance weighted search approach outlined in AlphaLLM [1], an approach that runs MCTS for search. We use the heuristic from AlphaLLM and our **preliminary results** show that PAVs are 8x more compute efficient at test-time beam search.
>
> **Background on Alpha LLM**: For the importance weighted search approach of AlphaLLM, we implement beam search in the following way. At any given point, the beam consists of $N$ states $s_1, s_2, s_3, \ldots, s_N$. These are partially unrolled solutions from the base policy $\pi$, up until state $s_i$ (prefix). We then expand each node in the beam 3 times, by conditionally sampling from $\pi$ (conditioned on each of the states in the beam). We get $N \times 3$ states, which we rank with the following scoring function and then select the top $N$ state. Each of the new expanded states is of the form $s, a$, where $s$ is the previous state in the beam and $a \sim \pi(\cdot \mid s)$ is the new sampled action (step). Following, Section 4.3 from Ye Tian et. al. (2024), the score for the new state $(s, a)$ is $Q^\pi(s, a)  + C \cdot U(s)$ where $U(s)$ is the uncertainty bonus for the state $s$, which is computed as $U(s) = \sqrt{ \frac{n(s)}{\sum_{i=1}^{N} n(s_i)}}$. We use $C=0.25$, which we identified by tuning performance over a hold out validation set we use for PAVs as well. This resembles UCB or UCT-style exploration. Concretely, $n(s)$ is the effective number of children for node $s$ (Section 4.3.2 in Ye Tian et. al. (2024)). The term $n(s) = C^\prime \cdot I(s)$ is computed by linearly scaling the importance $I(s)$ defined as $I(s) = \max_a |V^\pi(s) - Q^\pi(s,a)|$, where $a$ is one of the $3$ actions sampled from state $s$ when expanding the beam. We tune and set $C^\prime=2.0$. Intuitively, AlphaLLM chooses to explore states that can change the $Q$-values by a lot, when we continue to sample from them. When the $Q$-values deviate by a lot, the $I(s)$ term increases. Consequently, so does $N(s)$ (effective children count) and $U(s)$ (uncertainty bonus).
>
> **In Appendix K (Figure 18), we show our results for the experiment of test-time beam search with PAVs, vs. the UCB style metric for exploration in Alpha LLMs**. Since we only had two days to implement and run experiments for AlphaLLM (with no code base available), our findings are preliminary. Nevertheless, we find that PAVs are 8x more compute efficient than AlphaLLM at test-time exploration for the discovery of the correct solution trace. This is likely because the exploration metric in AlphaLLM uses the absolute magnitude of the advantage under the base policy, vs. PAVs which use the signed advantage under the prover policy. Thus, our exploration metric prefers steps that increase the likelihood of a complementary prover to discover the correct solution, as opposed to preferring steps that simply change the previous state’s value function  (under the base policy) by the largest magnitude (which can also be negative).

---

> > ### Author Response · Authors · 2024-11-27
> >
> > >> **Takeaways in lines 452-453**
> >
> > Thanks for pointing this out! In the takeaway box, we have updated the submission to avoid the term “exploration” and clearly state that using the Best-of-K policies as provers improves performance of test-time beam search over samples from the base policy. Note that this result is highlighted in Result 2, Figure 5b (Section 4.1). We hope that this addresses the confusion.
> >
> > >> **Takeaways in lines 251-252**
> >
> > Thanks for pointing this out! We have updated both paragraphs: on beam search in Section 3.1 (L178-188); and on online RL with effective rewards in Section 3.2 (L232, L240), to make the exploration-exploitation tradeoff in PAVs more clear. We hope that this makes the takeaway ``Process rewards should correspond to progress, or advantage, as opposed to absolute values, for a better explore-exploit tradeoff during beam search and online RL’’ more clear.
> >
> > >> **Confidence Intervals**
> >
> > We did use multiple seeds (5 seeds) for the plots in Figure 4(a),(b),(c), Figure 5(a),(b),(c) and Figure 6 (as we note in Appendix C). These are our results on test-time beam search. For our RL training experiments (PAVs in Figure 7a), we have updated Appendix E to include our latest results on 3 independent runs of RL training (with 3 different random seeds), for both Gemma 2B SFT and Gemma 9B SFT base policies.

---

> > > ### Author Response · Authors · 2024-11-30
> > >
> > > Dear Reviewer,
> > >
> > > We apologize for bothering you, but since there are only two days left in the extended discussion period, we wanted to check in with you to see if our rebuttal addresses all outstanding concerns and have a chance to address any new ones.
> > >
> > > Authors

---

### Official Review · Reviewer_pfy5 · 2024-11-10

**Soundness:** 3
**Presentation:** 3
**Contribution:** 3
**Rating:** 6
**Confidence:** 3

**Summary:**

This paper introduces Process Advantage Verifiers - a novel approach to training and using process reward models for improving LLM reasoning. The key contribution is showing that process rewards should measure progress, which defined as advantages under a prover policy rather than just step correctness. The authors demonstrate that PAV can lead to significant improvements in both test-time search efficiency and online rl sample efficiency.

**Strengths:**

1. Novel insight about measuring progress through advantages rather than Q-values
2. Theoretical analysis characterizing good prover policies as those "complementary" to base policy
3. Clear empirical validation showing significant improvements over baselines

**Weaknesses:**

1. Evaluation only on mathematical reasoning tasks
2. Could benefit from testing on other structured reasoning domains
3. All experiments on a single model family (Gemma)

**Questions:**

1. Have you explored applying PAVs to other structured reasoning domains beyond mathematics?
2. Could you clarify the computational overhead of training and using PAVs compared to traditional ORMs?

---

> ### Author Response · Authors · 2024-11-19
> **Response to Reviewer pfy5 (Part I)**
>
> Thank you for the review and for a positive assessment of our work! To address your concerns, we have added discussion on the computational overhead of PAVs, compared to ORMs in Appendix H. **Please let us know if your concerns are addressed, and if so, we would be grateful if you are willing to raise your score.** We are very happy to discuss further!
>
>
> >> **Computational cost of training and using PAVs vs. ORMs**
>
> Thank you for the question. **We have added a detailed discussion on comparison of computational cost of training and using PAVs to Appendix H of the paper.** In summary, while training PAVs takes more compute than ORMs, when we use PAVs for test-time search or during online RL training they are more compute-efficient than ORMs for attaining the  same level of test performance. We detail our calculations below.
>
> **Training PAVs**: The training data for PAVs is collected by running rollouts to estimate the Q-values for each step. Since learning to predict expected future success at earlier steps is a statistically more challenging task than predicting the final score at the end of the complete sequence (as in ORM) , the training data for PAVs is larger and scales with the average number of steps in the LLM’s output ($\approx 10$ in our case), in order to be able to achieve the same level of prediction accuracy on all steps in a generation. With a 10x larger dataset, naturally the computational cost of training PAVs (compared to ORMs which only predict final outcomes) also scales by roughly the same factor.
>
> **Using PAVs**: Despite a larger cost for training PAVs, once trained, PAVs are much more compute-efficient than ORMs on test questions. In other words, while we do incur a larger training cost, this cost is amortized over rounds of deployment. Concretely, as we show in Section 4.1, for test-time search, PAVs are 1.5-5x more compute efficient than ORMs, at achieving the same level of performance (Figure 4 in submission). This means that if we were to use a verifier at least twice upon deployment, PAV is already more compute-efficient than an ORM, accounting for both the training and test-time compute cost of PAVs.
>
> For online RL, as we show in Section 5 (Figure 8c), PAVs achieve the same level performance in 1/6th of the RL training iterations that ORMs require. But in our implementation, for each iteration, PAVs score each step by feeding each prefix in a generation separately through the trained model, and ORMs only score the full trajectory. Thus, in this implementation PAVs consume 10x more compute per batch to score the generations in a batch (since we have 10 steps on average per generation). A more efficient implementation would score all prefixes in a  generation in a single forward pass through the trained PAV. Nevertheless, the reduction in RL iteration complexity with PAVs is big enough that even with our naïve implementation, the overall computational cost is less, compared to ORMs. For example, using the formula for training and inference FLOPs in [1], during online RL, to train the Gemma 2B base policy with PAVs, we need to spend about $2.5 \times 10^{18}$ FLOPs, but to achieve the same performance with ORMs, we need about $5.9 \times 10^{18}$ FLOPs, resulting in at least half the total computational FLOPs.
>
> [1] Hoffmann, Jordan, et al. "Training compute-optimal large language models." arXiv preprint arXiv:2203.15556 (2022).

---

> ### Author Response · Authors · 2024-11-19
> **Response to Reviewer pfy5 (Part II)**
>
> >> **Expanding PAVs to other tasks and model families.**
>
> We chose math reasoning domains in order to be able to perform a direct comparison  with prior works (Snell et. al. (2024), Wang et. al. (2024), Lightman et. al. (2023), Shao et. al. (2024), Cobbe et. al. (2021)) that study process and outcome reward models. Within this domain, most of these prior works study the harder MATH benchmark (Hendrycks et. al. (2021)), since performance on some other reasoning datasets like GSM8K is already saturated. For example, the performance of some of the base LLMs we consider (Gemma2-9B and Gemma2-27B) is itself $>85$% on GSM8K. Please note that the scope of tasks in our paper is comparable, if not larger than several of these prior works.
>
> At the same time, the conceptual framework we present for process reward models, and our approach PAV, is broad and not specific to math reasoning. It only requires access to a base policy and an accurate outcome reward model (this is the regular expression matcher $\mathrm{Rex}$ in our case) that can be queried without much cost on all generations sampled from the base policy, at least on a fixed set of input prompts . While extending our results to other settings like coding is definitely possible, we defer this direction of study to future work since it is unclear how to define a ''step'' for a code output. Additionally, while prior work [2] studies an initial design of steps, the codecontests [3] dataset is challenging for our models.  Thus to draw meaningful conclusions on coding, we will have to choose a different base model capable enough to generate code in a format with a natural partition of steps.
>
> On the MATH benchmark, we are trying to add results where we use PAVs to improve the performance of models from other families (like Mistral/Llama), but are unsure if it will complete during the span of the rebuttal period, since we need to train base models (which requires a different infrastructure), collect data to train PAVs, train the PAVs, and then use it for search/RL.
>
>
>
> [2] Zheng, Qinkai, et al. "Codegeex: A pre-trained model for code generation with multilingual benchmarking on humaneval-x." Proceedings of the 29th ACM SIGKDD Conference on Knowledge Discovery and Data Mining. 2023.
>
> [3] Li, Yujia, et al. "Competition-level code generation with alphacode." Science 378.6624 (2022): 1092-1097.

---

> > ### Author Response · Authors · 2024-11-24
> >
> > Dear Reviewer,
> >
> > We apologize for bothering you, but since there are only two days left in the discussion period, we wanted to check in with you to see if our rebuttal addresses all outstanding concerns and have a chance to address any new ones.
> >
> > Thanks,
> > Authors

---

> > > ### Author Response · Authors · 2024-11-30
> > >
> > > Dear Reviewer,
> > >
> > > We apologize for bothering you, but since there are only two days left in the extended discussion period, we wanted to check in with you to see if our rebuttal addresses all outstanding concerns and have a chance to address any new ones.
> > >
> > > Authors

---

### Meta-Review · Area_Chair_qFhR · 2024-12-17

**Metareview:**

The reviewers unanimously appreciate the reward-densifying transformation proposed here for test time LLM evaluation, both in theory and practice. I concur with this assessment and therefore recommend this paper be accepted.

**Additional Comments On Reviewer Discussion:**

NA

---

### Decision · Program_Chairs · 2025-01-22

Accept (Spotlight)